# Quantifying & Modeling Multimodal Interactions: An Information Decomposition Framework

**Paul Pu Liang**[1], **Yun Cheng**[1,2], **Xiang Fan**[1,3], **Chun Kai Ling**[1,8], **Suzanne Nie**[1],
**Richard J. Chen**[4,5], **Zihao Deng**[6], **Nicholas Allen**[7], **Randy Auerbach**[8], **Faisal Mahmood**[4,5],
**Ruslan Salakhutdinov**[1], **Louis-Philippe Morency**[1]
[1]CMU, [2]Princeton University, [3]UW, [4]Harvard Medical School, [5]Brigham and Women's Hospital,
[6]University of Pennsylvania, [7]University of Oregon, [8]Columbia University
`pliang@cs.cmu.edu, yc6206@cs.princeton.edu`

## Abstract

The recent explosion of interest in multimodal applications has resulted in a wide selection of datasets and methods for representing and integrating information from different modalities. Despite these empirical advances, there remain fundamental research questions: *How can we quantify the interactions that are necessary to solve a multimodal task? Subsequently, what are the most suitable multimodal models to capture these interactions?* To answer these questions, we propose an information-theoretic approach to quantify the degree of *redundancy*, *uniqueness*, and *synergy* relating input modalities with an output task. We term these three measures as the PID *statistics of a multimodal distribution* (or PID for short), and introduce two new estimators for these PID statistics that scale to high-dimensional distributions. To validate PID estimation, we conduct extensive experiments on both synthetic datasets where the PID is known and on large-scale multimodal benchmarks where PID estimations are compared with human annotations. Finally, we demonstrate their usefulness in (1) quantifying interactions within multimodal datasets, (2) quantifying interactions captured by multimodal models, (3) principled approaches for model selection, and (4) three real-world case studies engaging with domain experts in pathology, mood prediction, and robotic perception where our framework helps to recommend strong multimodal models for each application.

## 1 Introduction

A core challenge in machine learning lies in capturing the interactions between multiple input modalities. Learning different types of multimodal interactions is often quoted as motivation for many successful multimodal modeling paradigms, such as contrastive learning to capture redundancy [54, 82], modality-specific representations to retain unique information [100], as well as tensors and multiplicative interactions to learn higher-order interactions [52, 58, 115]. However, several fundamental research questions remain: *How can we quantify the interactions that are necessary to solve a multimodal task? Subsequently, what are the most suitable multimodal models to capture these interactions?* This paper aims to formalize these questions by proposing an approach to quantify the *nature* (i.e., which type) and *degree* (i.e., the amount) of modality interactions, a fundamental principle underpinning our understanding of multimodal datasets and models [62].

By bringing together two previously disjoint research fields of Partial Information Decomposition (PID) in information theory [13, 41, 108] and multimodal machine learning [9, 62], we provide precise definitions categorizing interactions into *redundancy*, *uniqueness*, and *synergy*. Redundancy quantifies information shared between modalities, uniqueness quantifies the information present in only one of the modalities, and synergy quantifies the emergence of new information not previously present in either modality. A key aspect of these four measures is that they not only quantify interactions between modalities, but also how they relate to a downstream task. Figure 1 shows a depiction of these four measures, which we refer to as PID statistics. Leveraging insights from

37th Conference on Neural Information Processing Systems (NeurIPS 2023).

neural representation learning, we propose two new estimators for PID statistics that can scale to high-dimensional multimodal datasets and models. The first estimator is exact, based on convex optimization, and is able to scale to features with discrete support, while the second estimator is an approximation based on sampling, which enables us to handle features with large discrete or even continuous supports. We validate our estimation of PID in 2 ways: (1) on synthetic datasets where PID statistics are known due to exact computation and from the nature of data generation, and (2) on real-world data where PID is compared with human annotation. Finally, we demonstrate that these estimated PID statistics can help in multimodal applications involving:

1. **Dataset quantification**: We apply PID to quantify large-scale multimodal datasets, showing that these estimates match common intuition for interpretable modalities (e.g., language, vision, and audio) and yield new insights in other domains (e.g, healthcare, HCI, and robotics).
2. **Model quantification**: Across a suite of models, we apply PID to interpret model predictions and find consistent patterns of interactions that different models capture.
3. **Model selection**: Given our findings from dataset and model quantification, a new research question naturally arises: *given a new multimodal task, can we quantify its* PID *values to infer (a priori) what type of models are most suitable?* Our experiments show success in model selection for both existing benchmarks and completely new case studies engaging with domain experts in computational pathology, mood prediction, and robotics to select the best multimodal model.

Finally, we make public a suite of trained models across 10 model families and 30 datasets to accelerate future analysis of multimodal interactions at `https://github.com/pliang279/PID`.

## 2 Background and Related Work

Let $\mathcal{X}_i$ and $\mathcal{Y}$ be sample spaces for features and labels. Define $\Delta$ to be the set of joint distributions over $(\mathcal{X}_1, \mathcal{X}_2, \mathcal{Y})$. We are concerned with features $X_1, X_2$ (with support $\mathcal{X}_i$) and labels $Y$ (with support $\mathcal{Y}$) drawn from some distribution $p \in \Delta$. We denote the probability mass (or density) function by $p(x_1, x_2, y)$, where omitted parameters imply marginalization. Key to our work is defining estimators that given $p$ or samples $\{(x_1, x_2, y) : \mathcal{X}_1 \times \mathcal{X}_2 \times \mathcal{Y}\}$ thereof (i.e., dataset or model predictions), returns estimates for the amount of redundant, unique, and synergistic interactions.

### 2.1 Partial Information Decomposition

Information theory formalizes the amount of information that one variable provides about another [86]. However, its extension to 3 variables is an open question [34, 70, 93, 106]. In particular, the natural three-way mutual information $I(X_1; X_2; Y) = I(X_1; X_2) - I(X_1; X_2|Y)$ [70, 93] can be both positive and negative, which makes it difficult to interpret. In response, Partial information decomposition (PID) [108] generalizes information theory to multiple variables by decomposing $I_p(X_1, X_2; Y)$, the total information 2 variables $X_1, X_2$ provide about a task $Y$ into 4 quantities (see Figure 1): redundancy $R$ between $X_1$ and

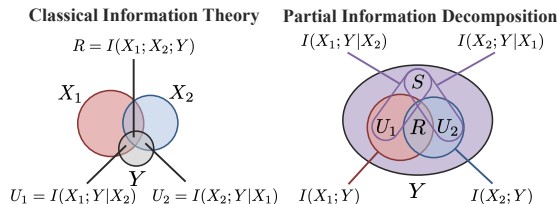

Figure 1: PID decomposes $I(X_1, X_2; Y)$ into redundancy $R$ between $X_1$ and $X_2$, uniqueness $U_1$ in $X_1$ and $U_2$ in $X_2$, and synergy $S$ in both $X_1$ and $X_2$.

$X_2$, uniqueness $U_1$ in $X_1$ and $U_2$ in $X_2$, and synergy $S$ that only emerges when both $X_1$ and $X_2$ are present. We adopt the PID definition proposed by Bertschinger et al. [13]:

$$R = \max_{q \in \Delta_p} I_q(X_1; X_2; Y), \tag{1}$$

$$U_1 = \min_{q \in \Delta_p} I_q(X_1; Y|X_2), \quad U_2 = \min_{q \in \Delta_p} I_q(X_2; Y|X_1), \tag{2}$$

$$S = I_p(X_1, X_2; Y) - \min_{q \in \Delta_p} I_q(X_1, X_2; Y), \tag{3}$$

where $\Delta_p = \{q \in \Delta : q(x_i, y) = p(x_i, y) \ \forall y \in \mathcal{Y}, x_i \in \mathcal{X}_i, i \in [2]\}$ and the notation $I_p(\cdot)$ and $I_q(\cdot)$ disambiguates mutual information under $p$ and $q$ respectively. The key lies in optimizing $q \in \Delta_p$ to satisfy the marginals $q(x_i, y) = p(x_i, y)$, but relaxing the coupling between $x_1$ and $x_2$: $q(x_1, x_2)$ need not be equal to $p(x_1, x_2)$. The intuition behind this is that one should be able to infer redundancy and uniqueness given only access to $p(x_1, y)$ and $p(x_2, y)$, and therefore they should only depend on $q \in \Delta_p$. Synergy is the only term that should depend on the coupling $p(x_1, x_2)$, and this is reflected

in (3) depending on the full $p$ distribution. This definition enjoys several useful properties in line with intuition, as we will see in comparison with related frameworks for interactions below [13].

## 2.2 Related Frameworks for Feature Interactions

**Information-theoretic definitions**: Perhaps the first measure of redundancy in machine learning is co-training [14, 8, 23], where 2 variables are redundant if they are conditionally independent given the task: $I(X_1; X_2|Y) = 0$. As a result, redundancy can be measured by $I(X_1; X_2; Y)$. The same definition of redundancy is used in multi-view learning [98, 94, 89] which further define $I(X_1; Y|X_2)$ and $I(X_2; Y|X_1)$ as unique information in $X_1, X_2$. However, $I(X_1; X_2; Y)$ can be both positive and negative [51]. PID resolves this by separating $R$ and $S$ such that $R - S = I(X_1; X_2; Y)$, identifying that prior measures confound redundancy and synergy. This crucially provides an explanation for the distinction between *mediation*, where one feature conveys the information already in another (i.e., $R > S$), versus *moderation*, where one feature affects the relationship of other features (i.e., $S > R$) [10, 36]. Furthermore, if $I(X_1; X_2; Y) = 0$ then existing frameworks are unable to distinguish between positive $R$ and $S$ canceling each other out.

**Statistical measures**: Other approaches have studied interaction measures via statistical measures, such as redundancy via distance between prediction logits using either feature [69], statistical distribution tests on input features [114, 7], or via human annotations [85]. However, it is unclear how to extend these definitions to uniqueness and synergy while remaining on the same standardized scale like PID provides. Also of interest are notions of redundant and synergistic interactions in human and animal communication [78, 79, 30, 85], which we aim to formalize.

**Model-based methods**: Prior research has formalized definitions of non-additive interactions [32] to quantify their presence [88, 101, 102, 46] in trained models, or used Shapley values on trained features to measure interactions [49]. Parallel research has also focused on qualitative visualizations of real-world multimodal datasets and models, such as DIME [68], M2Lens [105], and MultiViz [61].

# 3 Scalable Estimators for PID

**PID as a framework for multimodality**: Our core insight is that PID provides a formal framework to understand both the *nature* and *degree* of interactions involved when two features $X_1$ and $X_2$ are used for task $Y$. The nature of interactions is afforded by a precise decomposition into redundant, unique, and synergistic interactions, and the degree of interactions is afforded by a standardized unit of measure (bits). However, computing PID is a considerable challenge, since it involves optimization over $\Delta_p$ and estimating information-theoretic measures. Up to now, analytic approximations of these quantities were only possible for discrete and small support [13, 41, 109] or continuous but low-dimensional variables [76, 81, 110]. Leveraging ideas in representation learning, Sections 3.1 and 3.2 are our first technical contributions enabling scalable estimation of PID for high-dimensional distributions. The first, CVX, is exact, based on convex optimization, and is able to scale to problems where $|\mathcal{X}_i|$ and $|\mathcal{Y}|$ are around 100. The second, BATCH, is an approximation based on sampling, which enables us to handle large or even continuous supports for $X_i$ and $Y$. Applying these estimators in Section 4, we show that PID provides a path towards understanding the nature of interactions in datasets and those learned by different models, and principled approaches for model selection.

## 3.1 CVX: Dataset-level Optimization

Our first estimator, CVX, directly compute PID from its definitions using convex programming. Crucially, Bertschinger et al. [13] show that the solution to the max-entropy optimization problem: $q^* = \arg\max_{q \in \Delta_p} H_q(Y|X_1, X_2)$ equivalently solves (1)-(3). When $\mathcal{X}_i$ and $\mathcal{Y}$ are small and discrete, we can represent all valid distributions $q(x_1, x_2, y)$ as a set of tensors $Q$ of shape $|\mathcal{X}_1| \times |\mathcal{X}_2| \times |\mathcal{Y}|$ with each entry representing $Q[i, j, k] = p(X_1 = i, X_2 = j, Y = k)$. The problem then boils down to optimizing over valid tensors $Q \in \Delta_p$ that match the marginals $p(x_i, y)$.

Given a tensor $Q$ representing $q$, our objective is the concave function $H_q(Y|X_1, X_2)$. While Bertschinger et al. [13] report that direct optimization is numerically difficult as routines such as Mathematica's FINDMINIMUM do not exploit convexity, we overcome this by rewriting conditional entropy as a KL-divergence [38], $H_q(Y|X_1, X_2) = \log|\mathcal{Y}| - KL(q||\tilde{q})$, where $\tilde{q}$ is an auxiliary product density of $q(x_1, x_2) \cdot \frac{1}{|\mathcal{Y}|}$ enforced using linear constraints: $\tilde{q}(x_1, x_2, y) = q(x_1, x_2)/|\mathcal{Y}|$. Finally, optimizing over $Q \in \Delta_p$ that match the marginals can also be enforced through linear constraints: the 3D-tensor $Q$ summed over the second dimension gives $q(x_1, y)$ and summed over

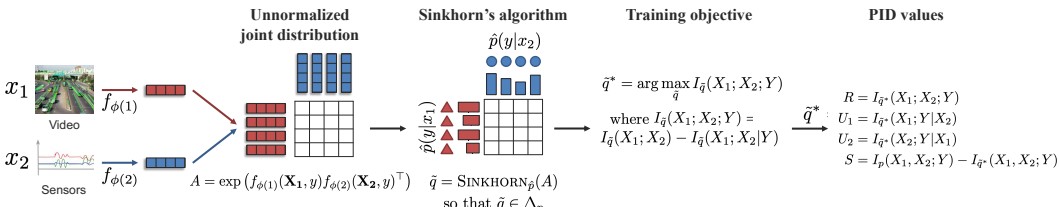

**Figure 2:** We propose BATCH, a scalable estimator for PID over high-dimensional continuous distributions. BATCH parameterizes $\tilde{q}$ using a matrix $A$ learned by neural networks such that mutual information objectives over $\tilde{q}$ can be optimized via gradient-based approaches over minibatches. Marginal constraints $\tilde{q} \in \Delta_p$ are enforced through a variant of the Sinkhorn-Knopp algorithm on $A$.

the first dimension gives $q(x_2, y)$, yielding the final optimization problem:

$$\underset{Q, \tilde{Q}}{\arg\max} \, KL(Q||\tilde{Q}), \quad \text{s.t.} \quad \tilde{Q}(x_1, x_2, y) = Q(x_1, x_2)/|\mathcal{Y}|, \tag{4}$$

$$\sum_{x_2} Q = p(x_1, y), \sum_{x_1} Q = p(x_2, y), Q \geq 0, \sum_{x_1, x_2, y} Q = 1. \tag{5}$$

The KL-divergence objective is recognized as convex, allowing the use of conic solvers such as SCS [75], ECOS [27], and MOSEK [6]. Plugging $q^*$ into (1)-(3) yields the desired PID.

**Pre-processing via feature binning**: In practice, $X_1$ and $X_2$ often take continuous rather than discrete values. We work around this by histogramming each $X_i$, thereby estimating the continuous joint density by discrete distributions with finite support. We provide more details in Appendix B.1.

### 3.2 BATCH: Batch-level Amortization

We now present BATCH, our next estimator that is suitable for large datasets where $\mathcal{X}_i$ is high-dimensional and continuous ($|\mathcal{Y}|$ remains finite). To estimate PID given a sampled dataset $\mathcal{D} = \{(x_1^{(j)}, x_2^{(j)}, y^{(j)})\}$ of size $n$, we propose an end-to-end model parameterizing marginal-matching joint distributions in $\Delta_p$ and a training objective whose solution returns approximate PID values.

**Simplified algorithm sketch**: Our goal, loosely speaking, is to optimize $\tilde{q} \in \Delta_p$ for objective (1) through an approximation using neural networks instead of exact optimization. We show an overview in Figure 2. To explain our approach, we first describe (1) how we parameterize $\tilde{q}$ using neural networks such that it can be learned via gradient-based approaches, (2) how we ensure the marginal constraints $\tilde{q} \in \Delta_p$ through a variant of the Sinkhorn-Knopp algorithm, and finally (3) how to scale this up over small subsampled batches from large multimodal datasets.

**Parameterization using neural networks**: The space of joint distributions $\Delta$ is often too large to explicitly specify. To tackle this, we implicitly parameterize each distribution $\tilde{q} \in \Delta$ using a neural network $f_\phi$ that takes in batches of modalities $\mathbf{X_1} \in \tilde{\mathcal{X}}_1^n, \mathbf{X_2} \in \tilde{\mathcal{X}}_2^n$ and the label $\mathbf{Y} \in \mathcal{Y}^n$ before returning a matrix $A \in \mathbb{R}^{n \times n \times |\mathcal{Y}|}$ representing an (unnormalized) joint distribution $\tilde{q}$, i.e., we want $A[i][j][y] = \tilde{q}(\mathbf{X_1}[i], \mathbf{X_2}[j], y)$ for each $y \in \mathcal{Y}$. In practice, $f_\phi$ is implemented via a pair of encoders $f_{\phi(1)}$ and $f_{\phi(2)}$ that learn modality representations, before an outer product to learn joint relationships $A_y = \exp(f_{\phi(1)}(\mathbf{X_1}, y) f_{\phi(2)}(\mathbf{X_2}, y)^\top)$ for each $y$, yielding the desired $n \times n \times |\mathcal{Y}|$ joint distribution. As a result, optimizing over $\tilde{q}$ can be performed via optimizing over parameters $\phi$.

**Respecting the marginal constraints**: How do we make sure the $\tilde{q}$'s learned by the network satisfies the marginal constraints (i.e., $\tilde{q} \in \Delta_p$)? We use an unrolled version of Sinkhorn's algorithm [24] which projects $A$ onto $\Delta_p$ by iteratively normalizing $A$'s rows and columns to sum to 1 and rescaling to satisfy the marginals $p(x_i, y)$. However, $p(x_i, y)$ is not easy to estimate for high-dimensional continuous $x_i$'s. In response, we first expand $p(x_i, y)$ into $p(y|x_i)$ and $p(x_i)$ using Bayes' rule. Since $A$ was constructed by samples $x_i$ from the dataset, the rows and columns of $A$ are already distributed according to $p(x_1)$ and $p(x_2)$ respectively. This means that it suffices to approximate $p(y|x_i)$ with unimodal classifiers $\hat{p}(y|x_i)$ parameterized by neural networks and trained separately, before using Sinkhorn's algorithm to normalize each row to $\hat{p}(y|x_1)$ and each column to $\hat{p}(y|x_2)$.

**Objective**: We choose the objective $I_q(X_1; X_2; Y)$, which equivalently solves the optimization problems in the other PID terms [13]. Given matrix $A$ representing $\tilde{q}(x_1, x_2, y)$, the objective can be computed in closed form through appropriate summation across dimensions in $A$ to obtain $\tilde{q}(x_i)$, $\tilde{q}(x_1, x_2)$, $\tilde{q}(x_i|y)$, and $\tilde{q}(x_1, x_2|y)$ and plugging into $I_{\tilde{q}}(X_1; X_2; Y) = I_{\tilde{q}}(X_1; X_2) - I_{\tilde{q}}(X_1; X_2|Y)$. We maximize $I_{\tilde{q}}(X_1; X_2; Y)$ by updating parameters $\phi$ via gradient-based methods. Overall, each

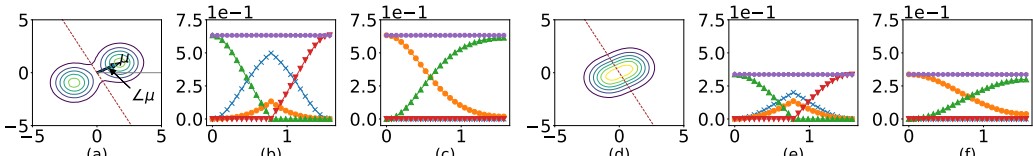

Figure 3: Left to right: (a) Contour plots of the GMM's density for $||\mu||_2 = 2.0$. Red line denotes the optimal linear classifier. (b) PID (Cartesian) computed for varying $\angle \mu$ with respect to the $x$ axis. (c) PID (Polar) for varying $\angle \mu$, with $U_1$ and $U_2$ corresponding to unique information from $(r, \theta)$. Plots (d)-(f) are similar to (a)-(c), but repeated for $||\mu||_2 = 1.0$. Legend: ✖($R$), ▲ ($U_1$), ▼ ($U_2$), ● ($S$), ✚(Sum). Observe how PID changes with the change of variable from Cartesian (b and e) to Polar (c and f), as well as how a change in $||\mu||_2$ can lead to a disproportionate change across PID (b vs e).

gradient step involves computing $\tilde{q} = \textsc{Sinkhorn}_{\hat{p}}(A)$, and updating $\phi$ to maximize (1) under $\tilde{q}$. Since Sinkhorn's algorithm is differentiable, gradients can be backpropagated end-to-end.

**Approximation with small subsampled batches**: Finally, to scale this up to large multimodal datasets where the full $\tilde{q}$ may be too large to store, we approximate $\tilde{q}$ with small subsampled batches: for each gradient iteration $t$, the network $f_\phi$ now takes in a batch of $m \ll n$ datapoints sampled from $\mathcal{D}$ and returns $A \in \mathbb{R}^{m \times m \times |\mathcal{Y}|}$ for the subsampled points. We perform Sinkhorn's algorithm on $A$ and a gradient step on $\phi$ as above, *as if* $\mathcal{D}_t$ was the full dataset (i.e., mini-batch gradient descent). While it is challenging to obtain full-batch gradients since computing the full $A$ is intractable, we found our approach to work in practice for large $m$. Our approach can also be informally viewed as performing amortized optimization [3] by using $\phi$ to implicitly share information about the full batch using subsampled batches. Upon convergence of $\phi$, we extract PID by plugging $\tilde{q}$ into (1)-(3).

**Implementation details** such as the network architecture of $f$, approximation of objective (1) via sampling from $\tilde{q}$, and estimation of $I_{\tilde{q}}(\{X_1, X_2\}; Y)$ from learned $\tilde{q}$ are in Appendix B.3.

## 4 Evaluation and Applications of PID in Multimodal Learning

We design experiments to (1) understand PID on synthetic data, (2) quantify real-world multimodal benchmarks, (3) understand the interactions captured by multimodal models, (4) perform model selection across different model families, and (5) applications on novel real-world tasks.

### 4.1 Validating PID Estimates on Synthetic Data

Our first goal is to evaluate the accuracy of our proposed estimators with respect to the ground truth (if it can be computed) or human judgment (for cases where the ground truth cannot be readily obtained). We start with a suite of datasets spanning both synthetic and real-world distributions.

**Synthetic bitwise features**: We sample from a binary bitwise distribution: $x_1, x_2 \sim \{0, 1\}, y = x_1 \wedge x_2, y = x_1 \vee x_2, y = x_1 \oplus x_2,$. Each bitwise operator's PID can be solved exactly when the $x_i$'s and labels are discrete and low-dimensional [13]. Compared to the ground truth in Bertschinger et al. [13], both our estimators exactly recover the correct PID values (Table 1).

Table 1: Results on estimating PID on synthetic bitwise datasets. Both our estimators exactly recover the correct PID values as reported in Bertschinger et al. [13].

| Task | OR | | | | AND | | | | XOR | | | |
|---|---|---|---|---|---|---|---|---|---|---|---|---|
| PID | $R$ | $U_1$ | $U_2$ | $S$ | $R$ | $U_1$ | $U_2$ | $S$ | $R$ | $U_1$ | $U_2$ | $S$ |
| Exact | 0.31 | 0 | 0 | 0.5 | 0.31 | 0 | 0 | 0.5 | 0 | 0 | 0 | 1 |
| CVX | 0.31 | 0 | 0 | 0.5 | 0.31 | 0 | 0 | 0.5 | 0 | 0 | 0 | 1 |
| Batch | 0.31 | 0 | 0 | 0.5 | 0.31 | 0 | 0 | 0.5 | 0 | 0 | 0 | 1 |

**Gaussian Mixture Models (GMM)**: Consider a GMM, where $X_1$, $X_2 \in \mathbb{R}$ and the label $Y \in \{-1, +1\}$, comprising two equally weighted standard multivariate Gaussians centered at $\pm\mu$, where $\mu \in \mathbb{R}^2$, i.e., $Y \sim \text{Bernoulli}(1/2)$, $(X_1, X_2)|Y = y \sim \mathcal{N}(y \cdot \mu, I)$. PID was estimated by sampling $1e6$ points, histogramming them into 50 bins spanning $[-5, +5]$ to give $p$, and then applying the CVX estimator. We term this PID-Cartesian. We also compute PID-Polar, which are PID computed using *polar coordinates*, $(r, \theta)$. We use a variant where the angle $\theta$ is given by the arctangent with principal values $[0, \pi]$ and the length $r \in \mathbb{R}$ could be negative. $\theta$ specifies a line (through the origin), and $r$ tells us where along the line the datapoint lies on.

**Results:** We consider $||\mu||_2 \in \{1.0, 2.0\}$, where for each $||\mu||_2$, we vary the angle $\angle \mu$ that $\mu$ makes with the horizontal axis. Our computed PID is presented in Figure 3. Overall, we find that the PID matches what we expect from intuition. For Cartesian, unique information dominates when

Table 2: Estimating PID on synthetic generative model datasets. Both CVX and BATCH measures agree with each other on relative values and are consistent with ground truth interactions.

| Task | $\mathcal{D}_R$ | | | | $\mathcal{D}_{U_1}$ | | | | $\mathcal{D}_{U_2}$ | | | | $\mathcal{D}_S$ | | | | $y = f(z_1^*, z_2^*, z_c^*)$ | | | |
|---|---|---|---|---|---|---|---|---|---|---|---|---|---|---|---|---|---|---|---|---|
| PID | $R$ | $U_1$ | $U_2$ | $S$ | $R$ | $U_1$ | $U_2$ | $S$ | $R$ | $U_1$ | $U_2$ | $S$ | $R$ | $U_1$ | $U_2$ | $S$ | $R$ | $U_1$ | $U_2$ | $S$ |
| CVX | **0.16** | 0 | 0 | 0.05 | 0 | **0.16** | 0 | 0.05 | 0 | 0 | **0.17** | 0.05 | 0.07 | 0 | 0.01 | **0.14** | **0.04** | 0.01 | 0 | **0.07** |
| BATCH | **0.29** | 0.02 | 0.02 | 0 | 0 | **0.30** | 0 | 0 | 0 | 0 | **0.30** | 0 | 0.11 | 0.02 | 0.02 | **0.15** | **0.06** | 0.01 | 0.01 | **0.06** |
| Truth | 0.58 | 0 | 0 | 0 | 0 | 0.56 | 0 | 0 | 0 | 0 | 0.54 | 0 | 0 | 0 | 0 | 0.56 | 0.13 | 0 | 0 | 0.27 |

| Task | $y = f(z_1, z_2^*, z_c^*)$ | | | | $y = f(z_1, z_2, z_c^*)$ | | | | $y = f(z_1^*, z_2^*, z_c)$ | | | | $y = f(z_2^*, z_c^*)$ | | | | $y = f(z_2^*, z_c)$ | | | |
|---|---|---|---|---|---|---|---|---|---|---|---|---|---|---|---|---|---|---|---|---|
| PID | $R$ | $U_1$ | $U_2$ | $S$ | $R$ | $U_1$ | $U_2$ | $S$ | $R$ | $U_1$ | $U_2$ | $S$ | $R$ | $U_1$ | $U_2$ | $S$ | $R$ | $U_1$ | $U_2$ | $S$ |
| CVX | 0.04 | **0.06** | 0 | **0.07** | 0.07 | 0 | 0 | **0.12** | 0.1 | 0 | 0.01 | **0.07** | 0.03 | 0 | **0.04** | 0.05 | 0.1 | 0 | 0.04 | 0.05 |
| BATCH | 0.04 | **0.09** | 0 | **0.06** | 0.11 | 0.02 | 0.02 | **0.10** | 0.11 | 0.02 | 0.02 | **0.05** | 0.07 | 0 | **0.06** | 0 | 0.19 | 0 | 0.06 | 0 |
| Truth | 0 | 0.25 | 0 | 0.25 | 0.18 | 0 | 0 | 0.36 | 0.22 | 0 | 0 | 0.22 | 0.21 | 0 | 0.21 | 0 | 0.34 | 0 | 0.17 | 0 |

the angle goes to 0 or $\pi/2$ — if centroids share a coordinate, then observing that coordinate yields no information about $y$. Conversely, synergy and redundancy peak at $\pi/4$. Interestingly, synergy seems to be independent of $\|\mu\|_2$. For Polar, redundancy is 0. Furthermore, $\theta$ contains no unique information, since $\theta$ shows nothing about $y$ unless we know $r$ (in particular, its sign). When the angle goes to $\pi/2$, almost all information is unique in $r$. The distinctions between Cartesian and Polar highlight how different representations of data can exhibit wildly different PID values, even if total information is the same. More thorough results and visualizations of $q^*$ are in Appendix C.2.

**Synthetic generative model**: We begin with a set of latent vectors $z_1, z_2, z_c \sim \mathcal{N}(0_d, \Sigma_d^2), d = 50$ representing information unique to $X_1, X_2$ and common to both respectively. $[z_1, z_c]$ is transformed into high-dimensional $x_1$ using a fixed transformation $T_1$ and likewise $[z_2, z_c]$ to $x_2$ via $T_2$. The label $y$ is generated as a function of (1) only $z_c$, in which case we expect complete redundancy, (2) only $z_1$ or $z_2$ which represents complete uniqueness, (3) a combination of $z_1$ and $z_2$ representing complete synergy, or (4) arbitrary ratios of each of the above with $z_i^*$ representing half of the dimensions from $z_i$ and therefore half of each interaction. In total, Table 2 shows the 10 synthetic datasets we generated: 4 specialized datasets $\mathcal{D}_I, I \in \{R, U_1, U_2, S\}$ where $y$ only depends on one interaction, and 6 mixed datasets with varying interaction ratios. We also report the ground-truth interactions as defined by the label-generating process and the total capturable information using the bound in Feder and Merhav [29], which relates the accuracy of the best model on these datasets with the mutual information between the inputs to the label. Since the test accuracies for Table 2 datasets range from 67-75%, this corresponds to total MI of $0.42 - 0.59$ bits.

**Results**: From Table 2, both CVX and BATCH agree in relative PID values, correctly assigning the predominant interaction type and interactions with minimal presence consistent with the ground-truth based on data generation. For example, $\mathcal{D}_R$ has the highest $R$ value, and when the ratio of $z_1$ increases, $U_1$ increases from 0.01 on $y = f(z_1^*, z_2^*, z_c^*)$ to 0.06 on $y = f(z_1, z_2^*, z_c^*)$. We also note some interesting observations due to the random noise in label generation, such as the non-zero synergy measure of datasets such as $\mathcal{D}_R, \mathcal{D}_{U_1}, \mathcal{D}_{U_2}$ whose labels do not depend on synergy.

### 4.2 Quantifying Real-world Multimodal Benchmarks

We now apply these estimators to quantify the interactions in real-world multimodal datasets.

**Real-world multimodal data setup**: We use a large collection of real-world datasets in Multi-Bench [60] which test *multimodal fusion* of different input signals (including images, video, audio, text, time-series, sets, and tables) for different tasks (predicting humor, sentiment, emotions, mortality rate, ICD-9 codes, image-captions, human activities, digits, and design interfaces). We also include experiments on *question-answering* (Visual Question Answering 2.0 [5, 39] and CLEVR [53]) which test grounding of language into the visual domain. For the 4 datasets (top row of Table 3) involving images and text where modality features are available and readily clustered, we apply the CVX estimator on top of discrete clusters. For the remaining 4 datasets (bottom row of Table 3) with video, audio, and medical time-series modalities, clustering is not easy, so we use the end-to-end BATCH estimator (see Appendix C.4 for full dataset and estimator details).

**Human judgment of interactions**: Real-world multimodal datasets do not have reference PID values, and exact PID computation is impossible due to continuous data. We therefore use human judgment as a reference. We design a new annotation scheme where we show both modalities and the label and ask each annotator to annotate the degree of redundancy, uniqueness, and synergy on a scale of 0-5, alongside their confidence in their answers on a scale of 0-5. We show a sample

Table 3: Estimating PID on real-world MultiBench [60] datasets. Many of the estimated interactions align well with human judgement as well as unimodal performance.

| Task | AV-MNIST | | | | ENRICO | | | | VQA 2.0 | | | | CLEVR | | | |
|---|---|---|---|---|---|---|---|---|---|---|---|---|---|---|---|---|
| PID | $R$ | $U_1$ | $U_2$ | $S$ | $R$ | $U_1$ | $U_2$ | $S$ | $R$ | $U_1$ | $U_2$ | $S$ | $R$ | $U_1$ | $U_2$ | $S$ |
| CVX | 0.10 | **0.97** | 0.03 | 0.08 | **0.73** | 0.38 | 0.53 | 0.34 | 0.79 | 0.87 | 0 | **4.92** | 0.55 | 0.48 | 0 | **5.16** |
| Human | **0.57** | **0.61** | 0 | 0 | - | - | - | - | 0 | 0 | 0 | **6.58** | 0 | 0 | 0 | **6.19** |

| Task | MOSEI | | | | UR-FUNNY | | | | MUSTARD | | | | MIMIC | | | |
|---|---|---|---|---|---|---|---|---|---|---|---|---|---|---|---|---|
| PID | $R$ | $U_1$ | $U_2$ | $S$ | $R$ | $U_1$ | $U_2$ | $S$ | $R$ | $U_1$ | $U_2$ | $S$ | $R$ | $U_1$ | $U_2$ | $S$ |
| BATCH | **0.26** | **0.49** | 0.03 | 0.04 | 0.03 | 0.04 | 0.01 | **0.08** | 0.14 | 0.01 | 0.01 | **0.30** | 0.05 | **0.17** | 0 | 0.01 |
| Human | **0.32** | **0.20** | 0.15 | 0.15 | 0.04 | **0.05** | 0.03 | **0.04** | 0.13 | **0.17** | 0.04 | **0.16** | - | - | - | - |

user interface and annotation procedures in Appendix C.5. We give 50 datapoints from each dataset (except MIMIC and ENRICO which require specialized knowledge) to 3 annotators each.

**Results on multimodal fusion**: From Table 3, we find that different datasets do require different interactions. Some interesting observations: (1) all pairs of modalities on MUSTARD sarcasm detection show high synergy values, which aligns with intuition since sarcasm is often due to a contradiction between what is expressed in language and speech, (2) uniqueness values are strongly correlated with unimodal performance (e.g., modality 1 in AV-MNIST and MIMIC), (3) datasets with high synergy do indeed benefit from interaction modeling as also seen in prior work (e.g., MUSTARD, UR-FUNNY) [17, 43], and (4) conversely datasets with low synergy are those where unimodal performance is relatively strong (e.g., MIMIC) [60].

**Results on QA**: We observe very high synergy values as shown in Table 3 consistent with prior work studying how these datasets were balanced (e.g., VQA 2.0 having different images for the same question such that the answer can only be obtained through synergy) [39] and that models trained on these datasets require non-additive interactions [46]. CLEVR has a higher proportion of synergy than VQA 2.0 (83% versus 75%): indeed, CLEVR is a more balanced dataset where the answer strictly depends on both the question and image with a lower likelihood of unimodal biases.

**Comparisons with human judgment**: For human judgment, we cannot ask humans to give a score in bits, so it is on a completely different scale (0-5 scale). To put them on the same scale, we normalize the human ratings such that the sum of human interactions is equal to the sum of PID estimates. The resulting comparisons are in Table 3, and we find that the human-annotated interactions overall align with estimated PID: the highest values are the same for 4 datasets: both explain highest synergy on VQA and CLEVR, image ($U_1$) being the dominant modality in AV-MNIST, and language ($U_1$) being the dominant modality in MOSEI. Overall, the Krippendorff's alpha for inter-annotator agreement is high (0.72 for $R$, 0.68 for $U_1$, 0.70 for $U_2$, 0.72 for $S$) and the average confidence scores are also high (4.36/5 for $R$, 4.35/5 for $U_1$, 4.27/5 for $U_2$, 4.46/5 for $S$), indicating that the human-annotated results are reliable. For the remaining two datasets (UR-FUNNY and MUSTARD), estimated PID matches the second-highest human-annotated interaction. We believe this is because there is some annotator subjectivity in interpreting whether sentiment, humor, and sarcasm are present in language only ($U_1$) or when contextualizing both language and video ($S$), resulting in cases of low annotator agreement in $U_1$ and $S$: $-0.14$, $-0.03$ for UR-FUNNY and $-0.08$, $-0.04$ for MUSTARD.

**Comparisons with other interaction measures**: Our framework allows for easy generalization to other interaction definitions: we also implemented 3 information theoretic measures **I-min** [108], **WMS** [19], and **CI** [74]. These results are in Table 10 in the Appendix, where we explain the limitations of these methods as compared to PID, such as over- and under-estimation, and potential negative estimation [41]. These are critical problems with the application of information theory for shared $I(X_1; X_2; Y)$ and unique information $I(X_1; Y|X_2)$, $I(X_2; Y|X_1)$ often quoted in the co-training [8, 14] and multi-view learning [89, 94, 98] literature. We also tried 3 non-info theory measures: Shapley values [67], Integrated gradients (IG) [91], and CCA [4], which are based on quantifying interactions captured by a multimodal model. Our work is fundamentally different in that interactions are properties of data before training any models (see Appendix C.6).

### 4.3 Quantifying Multimodal Model Predictions

We now shift our focus to quantifying multimodal models. *Do different multimodal models learn different interactions?* A better understanding of the types of interactions that our current models struggle to capture can provide new insights into improving these models.

Table 4: Average interactions ($R/U/S$) learned by models alongside their average performance on interaction-specialized datasets ($\mathcal{D}_R/\mathcal{D}_U/\mathcal{D}_S$). Synergy is the hardest to capture and redundancy is relatively easier to capture by existing models.

| Model | EF | ADDITIVE | AGREE | ALIGN | ELEM | TENSOR | MI | MULT | LOWER | REC | AVERAGE |
|---|---|---|---|---|---|---|---|---|---|---|---|
| $R$ | 0.35 | **0.48** | **0.44** | **0.47** | 0.27 | **0.55** | 0.20 | 0.40 | **0.47** | **0.53** | **0.41 $\pm$ 0.11** |
| Acc($\mathcal{D}_R$) | 0.71 | **0.74** | **0.73** | **0.74** | 0.70 | **0.75** | 0.67 | 0.73 | **0.74** | **0.75** | 0.73 $\pm$ 0.02 |
| $U$ | 0.29 | 0.31 | 0.19 | 0.44 | 0.20 | 0.52 | 0.18 | 0.45 | **0.55** | **0.55** | **0.37 $\pm$ 0.14** |
| Acc($\mathcal{D}_U$) | 0.66 | 0.55 | 0.60 | 0.73 | 0.66 | 0.73 | 0.66 | 0.72 | **0.73** | **0.73** | 0.68 $\pm$ 0.06 |
| $S$ | 0.13 | 0.09 | 0.08 | 0.29 | 0.14 | **0.33** | 0.12 | **0.29** | 0.31 | **0.32** | **0.21 $\pm$ 0.10** |
| Acc($\mathcal{D}_S$) | 0.56 | 0.66 | 0.63 | 0.72 | 0.66 | **0.74** | 0.65 | **0.72** | 0.73 | **0.74** | 0.68 $\pm$ 0.06 |

**Setup**: For each dataset, we train a suite of models on the train set $\mathcal{D}_{\text{train}}$ and apply it to the validation set $\mathcal{D}_{\text{val}}$, yielding a predicted dataset $\mathcal{D}_{\text{pred}} = \{(x_1, x_2, \hat{y}) \in \mathcal{D}_{\text{val}}\}$. Running PID on $\mathcal{D}_{\text{pred}}$ summarizes the interactions that the model captures. We categorize and implement a comprehensive suite of models (spanning representation fusion at different feature levels, types of interaction inductive biases, and training objectives) that have been previously motivated to capture redundant, unique, and synergistic interactions (see Appendix C.7 for full model descriptions).

**Results**: We show results in Table 4 and highlight the following observations:

*General observations*: We first observe that model PID values are consistently higher than dataset PID. The sum of model PID is also a good indicator of test performance, which agrees with their formal definition since their sum is equal to $I(\{X_1, X_2\}; Y)$, the total task-relevant information.

*On redundancy*: Several methods succeed in capturing redundancy, with an overall average of $R = 0.41 \pm 0.11$ and accuracy of $73.0 \pm 2.0\%$ on redundancy-specialized datasets. Additive, agreement, and alignment-based methods are particularly strong, and we do expect them to capture redundant shared information [26, 82]. Methods based on tensor fusion (synergy-based), including lower-order interactions, and adding reconstruction objectives (unique-based) also capture redundancy.

*On uniqueness*: Uniqueness is harder to capture than redundancy, with an average of $U = 0.37 \pm 0.14$. Redundancy-based methods like additive and agreement do poorly on uniqueness, while those designed for uniqueness (lower-order interactions [115] and modality reconstruction objectives [100]) do well, with on average $U = 0.55$ and 73.0% accuracy on uniqueness datasets.

*On synergy*: Synergy is the hardest to capture, with an average score of only $S = 0.21 \pm 0.10$. Some of the strong methods are tensor fusion [33], tensors with lower-order interactions [115], modality reconstruction [100], and multimodal transformer [112], which achieve around $S = 0.30, \text{acc} = 73.0\%$. Additive, agreement, and element-wise interactions do not seem to capture synergy well.

*On robustness*: Finally, we also show connections between PID and model performance in the presence of missing modalities. We find high correlation ($\rho = 0.8$) between the performance drop when $X_i$ is missing and the model's $U_i$ value. Inspecting Figure 4, we find that the implication only holds in one direction: high $U_i$ coincides with large performance drops (in red), but low $U_i$ can also lead to performance drops (in green). The latter can be further explained by the presence of large $S$ values:

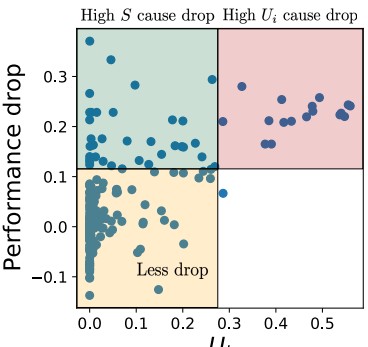

Figure 4: We find high correlation ($\rho = 0.8$) between the performance drop when $X_i$ is missing and the model's $U_i$ value: high $U_i$ coincides with large performance drops (red), but low $U_i$ can also lead to performance drops. The latter can be further explained by large $S$ so $X_i$ is necessary (green).

when $X_i$ is missing, synergy can no longer be learned which affects performance. For the subset of points when $U_i \leq 0.05$, the correlation between $S$ and performance drop is $\rho = 0.73$ (in contrast, the correlation for $R$ is $\rho = 0.01$).

### 4.4 PID Agreement and Model Selection

Now that we have quantified datasets and models individually, the natural next question unifies both: *what does the agreement between dataset and model* PID *measures tell us about model performance?* We hypothesize that models able to capture the interactions necessary in a given dataset should also

Table 5: **Model selection** results on unseen synthetic and real-world datasets. Given a new dataset $\mathcal{D}$, finding the closest synthetic dataset $\mathcal{D}'$ with similar PID values and recommending the best models on $\mathcal{D}'$ consistently achieves $95\% - 100\%$ of the best-performing model on $\mathcal{D}$.

| Dataset | 5 Synthetic Datasets | MIMIC | ENRICO | UR-FUNNY | MOSEI | MUSTARD | MAPS |
|---|---|---|---|---|---|---|---|
| % Performance | 99.91% | 99.78% | 100% | 98.58% | 99.35% | 95.15% | 100% |

achieve high performance. Given estimated interactions on dataset $\mathcal{D}$ and model $f(\mathcal{D})$ trained on $\mathcal{D}$, we define the agreement for each interaction $I \in \{R, U_1, U_2, S\}$ as:

$$\alpha_I(f, \mathcal{D}) = \hat{I}_{\mathcal{D}} I_{f(\mathcal{D})}, \quad \hat{I}_{\mathcal{D}} = \frac{I_{\mathcal{D}}}{\sum_{I' \in \{R, U_1, U_2, S\}} I'_{\mathcal{D}}}, \tag{6}$$

which summarizes the quantity of an interaction captured by a model ($I_{f(\mathcal{D})}$) weighted by its normalized importance in the dataset ($\hat{I}_{\mathcal{D}}$). The total agreement sums over $\alpha(f, \mathcal{D}) = \sum_I \alpha_I(f, \mathcal{D})$.

**Results**: Our key finding is that PID agreement scores $\alpha(f, \mathcal{D})$ correlate ($\rho = 0.81$) with model accuracy across all 10 synthetic datasets as illustrated in Figure 5. This shows that PID agreement can be a useful proxy for model performance. For the specialized datasets, we find that the correlation between $\alpha_I$ and $\mathcal{D}_I$ is 0.96 for $R$, 0.86 for $U$, and 0.91 for $S$, and negatively correlated with other specialized datasets. For mixed datasets with roughly equal ratios of each interaction, the measures that correlate most with performance are $\alpha_R$ ($\rho = 0.82$) and $\alpha_S$ ($\rho = 0.89$); datasets with relatively higher redundancy see $\rho = 0.89$ for $\alpha_R$; those with higher uniqueness have $\alpha_{U_1}$ and $\alpha_{U_2}$ correlate $\rho = 0.92$ and $\rho = 0.85$; those with higher synergy increases the correlation of $\alpha_S$ to $\rho = 0.97$.

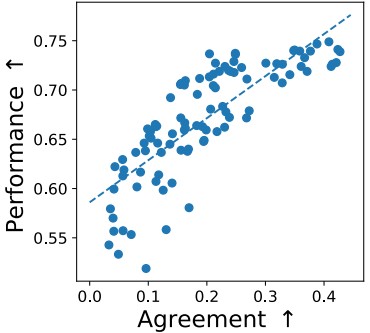

Figure 5: PID agreement $\alpha(f, \mathcal{D})$ between datasets and models strongly correlate with model accuracy ($\rho = 0.81$).

Using these observations, our final experiment is model selection: *can we choose the most appropriate model to tackle the interactions required for a dataset?*

**Setup**: Given a new dataset $\mathcal{D}$, we first compute its difference in normalized PID values with respect to $\mathcal{D}'$ among our suite of 10 synthetic datasets, $s(\mathcal{D}, \mathcal{D}') = \sum_{I \in \{R, U_1, U_2, S\}} |\hat{I}_{\mathcal{D}} - \hat{I}_{\mathcal{D}'}|$, to rank the dataset $\mathcal{D}^*$ with the most similar interactions, and return the top-3 performing models on $\mathcal{D}^*$. In other words, we select models that best capture interactions that are of similar nature and degree as those in $\mathcal{D}$. We emphasize that even though we restrict dataset and model search to *synthetic datasets*, we evaluate model selection on real-world datasets and find that it *generalizes to the real world*.

**Results**: We test our selected models on 5 new synthetic datasets with different PID ratios and 6 real-world datasets, summarizing results in Table 5. We find that the top 3 chosen models achieve $95\% - 100\%$ of the best-performing model accuracy, and $> 98.5\%$ for all datasets except $95.2\%$ on MUSTARD. For example, UR-FUNNY and MUSTARD have the highest synergy ($S = 0.13$, $S = 0.3$) and indeed transformers and higher-order interactions are helpful (MULT: 65%, MI: 61%, TENSOR: 60%). ENRICO has the highest $R = 0.73$ and $U_2 = 0.53$, and methods for redundant and unique interactions perform best (LOWER: 52%, ALIGN: 52%, AGREE: 51%). MIMIC has the highest $U_1 = 0.17$, and unimodal models are mostly sufficient [60].

### 4.5 Real-world Applications

Finally, we apply PID to 3 real-world case studies: pathology, mental health, and robotic perception (see full details and results in Appendix C.9-C.11).

**Case Study 1: Computational pathology.** Cancer prognostication is a challenging task in anatomic pathology that requires integration of whole-slide imaging (WSI) and molecular features for patient stratification [21, 63, 71]. We use The Cancer Genome Atlas (TCGA), a large public data consortium of paired WSI, molecular, and survival information [107, 96], including modalities: (1) pre-extracted histology image features from diagnostic WSIs and (2) bulk gene mutation status, copy number variation, and RNA-Seq abundance values. We evaluate on two cancer datasets in TCGA, lower-grade glioma (LGG [73], $n = 479$) and pancreatic adenocarcinoma (PAAD [83], $n = 209$).

**Results**: In TCGA-LGG, most PID measures were near-zero except $U_2 = 0.06$ for genomic features, which indicates that genomics is the only modality containing task-relevant information. This conclusion corroborates with the high performance of unimodal-genomic and multimodal models in Chen et al. [22], while unimodal-pathology performance was low. In TCGA-PAAD, the uniqueness in pathology and genomic features was less than synergy ($U_1 = 0.06$, and $U_2 = 0.08$ and $S = 0.15$), which also match the improvement of using multimodal models that capture synergy.

**Case Study 2: Mental health.** Suicide is the second leading cause of death among adolescents [18]. Intensive monitoring of behaviors via adolescents' frequent use of smartphones may shed new light on the early risk of suicidal ideations [37, 72], since smartphones provide rich behavioral markers [59]. We used a dataset, MAPS, of mobile behaviors from high-risk consenting adolescent populations (approved by IRB). Passive sensing data is collected from each participant's smartphone across 6 months. The modalities include (1) *text* entered by the user represented as a bag of top 1000 words, (2) *keystrokes* that record the exact timing and duration of each typed character, and (3) *mobile applications* used per day as a bag of 137 apps. Every morning, users self-report their daily mood, which we discretized into $-1, 0, +1$. In total, MAPS has 844 samples from 17 participants.

**Results**: We first experiment with $\text{MAPS}_{T,K}$ using text and keystroke features. PID measures show that $\text{MAPS}_{T,K}$ has high synergy ($0.40$), some redundancy ($0.12$), and low uniqueness ($0.04$). We found the purely synergistic dataset $\mathcal{D}_S$ has the most similar interactions and the suggested models LOWER, REC, and TENSOR that work best on $\mathcal{D}_S$ were indeed the top 3 best-performing models on $\text{MAPS}_{T,K}$, indicating that model selection is effective. Model selection also retrieves the best-performing model on $\text{MAPS}_{T,A}$ using text and app usage features.

**Case Study 3: Robotic Perception.** MuJoCo PUSH [57] is a contact-rich planar pushing task in MuJoCo [95], where a 7-DoF Panda Franka robot is pushing a circular puck with its end-effector in simulation. The dataset consists of 1000 trajectories with 250 steps sampled at 10Hertz. The multimodal inputs are gray-scaled images from an RGB camera, force and binary contact information from a force/torque sensor, and the 3D position of the robot end-effector. We estimate the 2D position of the unknown object on a table surface while the robot intermittently interacts with it.

**Results**: We find that BATCH predicts $U_1 = 1.79$ as the highest PID value, which aligns with our observation that image is the best unimodal predictor. Comparing both estimators, CVX underestimates $U_1$ and $R$ since the high-dimensional time-series modality cannot be easily described by clusters without losing information. In addition, both estimators predict a low $U_2$ value but attribute high $R$, implying that a multimodal model with higher-order interactions would not be much better than unimodal models. Indeed, we observe no difference in performance between these two.

## 5    Conclusion

Our work aims to quantify the nature and degree of feature interactions by proposing scalable estimators for redundancy, uniqueness, and synergy suitable for high-dimensional heterogeneous datasets. Through comprehensive experiments and real-world applications, we demonstrate the utility of our proposed framework in dataset quantification, model quantification, and model selection. We are aware of some potential **limitations**:

1. These estimators only approximate real interactions due to cluster preprocessing or unimodal models, which naturally introduce optimization and generalization errors. We expect progress in density estimators, generative models, and unimodal classifiers to address these problems.
2. It is harder to quantify interactions for certain datasets, such as ENRICO which displays all interactions which makes it difficult to distinguish between $R$ and $S$ or $U$ and $S$.
3. Finally, there exist challenges in quantifying interactions since the data generation process is never known for real-world datasets, so we have to resort to human judgment, other automatic measures, and downstream tasks such as estimating model performance and model selection.

**Future work** can leverage PID for targeted dataset creation, representation learning optimized for PID values, and applications of information theory to higher-dimensional data. More broadly, there are several exciting directions in investigating more applications of multivariate information theory in modeling feature interactions, predicting multimodal performance, and other tasks involving feature interactions such as privacy-preserving and fair representation learning from high-dimensional data [28, 42]. Being able to provide guarantees for fairness and privacy-preserving learning can be particularly impactful.

## Acknowledgements

This material is based upon work partially supported by Meta, National Science Foundation awards 1722822 and 1750439, and National Institutes of Health awards R01MH125740, R01MH132225, R01MH096951 and R21MH130767. PPL is supported in part by a Siebel Scholarship and a Waibel Presidential Fellowship. RS is supported in part by ONR grant N000142312368 and DARPA FA87502321015. One of the aims of this project is to understand the comfort zone of people for better privacy and integrity. Any opinions, findings, conclusions, or recommendations expressed in this material are those of the author(s) and do not necessarily reflect the views of the sponsors, and no official endorsement should be inferred. Finally, we would also like to acknowledge feedback from the anonymous reviewers who significantly improved the paper and NVIDIA's GPU support.

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

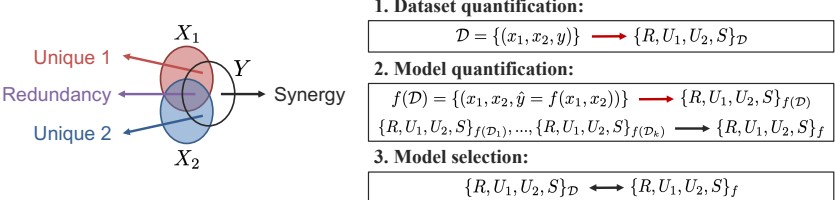

Figure 6: The PID framework formalizes the nature of feature interactions for a task. Through our proposed scalable estimators for *redundancy*, *uniqueness*, and *synergy*, we demonstrate their usefulness towards quantifying multimodal datasets and multimodal model predictions, which together yield principles for model selection.

## Appendix

## A    Partial Information Decomposition

Information theory formalizes the amount of information that a single variable $(X)$ provides about another $(Y)$, and is quantified by Shannon's mutual information [86] $I(X;Y)$ which measures the amount of uncertainty reduced from $H(Y)$ to $H(Y|X)$ when given $X$ as input. However, the direct extension of information theory to 3 or more variables, such as through total correlation [106, 34, 90, 97] or interaction information [70, 93, 12, 35], both have significant shortcomings. In particular, the three-way mutual information $I(X_1;X_2;Y)$ can be both negative and positive, leading to considerable difficulty in its interpretation by theoreticians and practitioners.

Partial information decomposition (PID) [108] was proposed as a method to generalize information theory to multiple variables, by positing a decomposition of the total information 2 variables provide about a task $I(X_1, X_2; Y)$ into 4 quantities: redundancy $R$ between $X_1$ and $X_2$, unique information $U_1$ in $X_1$ and $U_2$ in $X_2$, and synergy $S$ that should together satisfy the following consistency equations:

$$R + U_1 = I(X_1; Y), \tag{7}$$
$$R + U_2 = I(X_2; Y) \tag{8}$$
$$U_1 + S = I(X_1; Y|X_2), \tag{9}$$
$$U_2 + S = I(X_2; Y|X_1) \tag{10}$$
$$R - S = I(X_1; X_2; Y) \tag{11}$$

A definition of $R$ was first proposed by Williams and Beer [108] and was subsequently improved by Bertschinger et al. [13], Griffith and Koch [41] giving:

$$R = \max_{q \in \Delta_p} I_q(X_1; X_2; Y) \tag{12}$$

$$U_1 = \min_{q \in \Delta_p} I_q(X_1; Y|X_2) \tag{13}$$

$$U_2 = \min_{q \in \Delta_p} I_q(X_2; Y|X_1) \tag{14}$$

$$S = I_p(X_1, X_2; Y) - \min_{q \in \Delta_p} I_q(X_1, X_2; Y) \tag{15}$$

where $\Delta_p = \{q \in \Delta : q(x_i, y) = p(x_i, y) \ \forall y, x_i, i \in \{1, 2\}\}$ and $\Delta$ is the set of all joint distributions over $X_1, X_2, Y$. We call this the set of marginal-matching distributions. The core intuition of these definitions and how they enable PID lies in optimization over $\Delta_p$ instead of simply estimating information-theoretic measures for the observed distribution $p$. Specifically, synergy $S$ is the difference between total multimodal information in $p$ and total multimodal information in $q^* \in \Delta_p$, the non-synergistic distribution (see Figure 7 for an illustration). While there have been other definitions of PID that also satisfy the consistency equations, Bertschinger et al. [13] showed that their proposed measure satisfies several desirable properties. Optimizing over $\Delta_p$ can be seen as going beyond purely observed data and discovering latent redundancy, uniqueness, and synergy via representation learning.

According to Bertschinger et al. [13], it suffices to solve for $q$ using the following max-entropy optimization problem, the same $q$ equivalently solves any of the remaining problems defined for

Unimodal marginal-matching distributions:

$$\Delta_p = \{q(x_1, x_2, y) : q(x_1, y) = p(x_1, y), q(x_2, y) = p(x_2, y)\}$$

$$p(x_1, x_2, y)$$

$$S = I_p(X_1, X_2; Y) - \min_{q \in \Delta_p} I_q(X_1, X_2; Y)$$

Task-relevant multimodal info

Task-relevant multimodal info without synergy:

$$S_{q^*} = I_{q^*}(X_1, X_2; Y) - \min_{q \in \Delta_p} I_q(X_1, X_2; Y) = 0$$

Figure 7: We illustrate the set of marginal-matching distributions $q \in \Delta_p$, and an intuition behind the definition of synergy $S$ as the difference between total multimodal information in $p$ and total multimodal information in $q^* \in \Delta_p$, the non-synergistic distribution.

redundancy, uniqueness, and synergy.

$$q^* = \arg\max_{q \in \Delta_p} H_q(Y | X_1, X_2) \qquad \text{(max-entropy)} \qquad (16)$$

$$q^* = \arg\max_{q \in \Delta_p} I_q(X_1; X_2; Y) \qquad \text{(redundancy)} \qquad (17)$$

$$q^* = \arg\min_{q \in \Delta_p} I_q(\{X_1, X_2\}; Y) \qquad \text{(synergy)} \qquad (18)$$

$$q^* = \arg\min_{q \in \Delta_p} I_q(X_1; Y | X_2) \qquad \text{(unique in } X_1) \qquad (19)$$

$$q^* = \arg\min_{q \in \Delta_p} I_q(X_2; Y | X_1) \qquad \text{(unique in } X_2) \qquad (20)$$

# B  Scalable PID Estimators

This section provides algorithmic and implementation details on the estimation of PID values for high-dimensional continuous datasets.

## B.1  Estimator 1: CVX for Discrete Representations

Our first proposed estimator, CVX, is exact, based on convex optimization, and is able to scale to problems where $|\mathcal{X}_i|$ and $|\mathcal{Y}|$ are around 100. We show a high-level diagram of CVX in Figure 8 and provide several implementation details here:

**Implementation Details**: We implemented (5) using CVXPY [25, 1]. The transformation from the max-entropy objective (16) to (5) ensures adherence to disciplined convex programs [40], thus allowing CVXPY to recognize it as a convex program. All 3 conic solvers, ECOS [27], SCS [75], and MOSEK [6] were used, with ECOS and SCS being default solvers packaged with CVXPY. Our experience is that MOSEK is the fastest and most stable solver, working out of the box without any tuning. However, it comes with the downside of being commercial. For smaller problems, ECOS and SCS work just fine.

**Potential Numerical Issues**: Most conic solvers will suffer from numerical instability when dealing with ill-conditioned problems, typically resulting from coefficients that are too large or small. It is fairly common, especially in the GMM experiment that a single bin contains just one sample. Since $P$ is estimated from an empirical frequency matrix, it will contain probabilities of the order $1/\text{num-samples}$. When the number of samples is too large (e.g., $\geq 1e8$), this frequency is too small and causes all 3 solvers to crash.

**Approximating Continuous Representation using Histogramming**: The number and width of bins can affect quality of PID estimation [77]. For example, it is known that with a fixed number of samples, making the width of bins arbitrarily small will cause KL estimates to diverge. It is known that the number of bins should grow sub-linearly with the number of samples. Rice [84] suggest setting the number of bins to be the cubed-root of the number of samples. In our synthetic experiments, we sampled at least $1e6$ samples and used no more than 100 bins. Note that increasing the number of samples ad infinitum is not an option due to potential numerical issues described above.

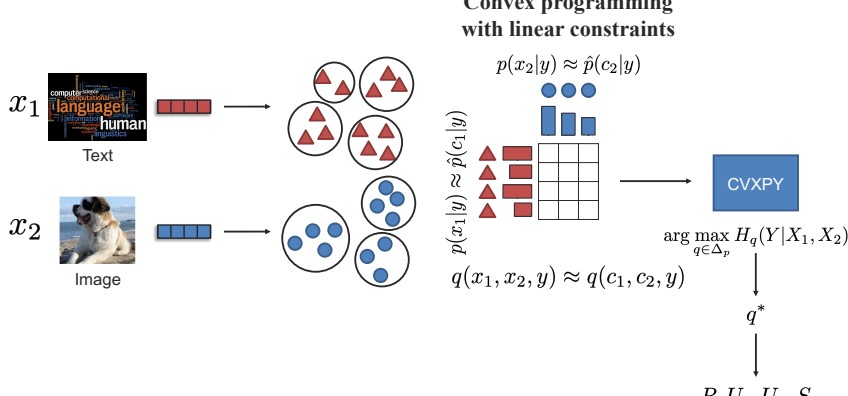

Figure 8: Overview of CVX estimator. Appropriate histogramming of each $X_i$ or clustering in preprocessed feature space is first performed to constrain $|\mathcal{X}_i|$. By rewriting the entropy objective as a KL divergence (with auxiliary variables) which is recognized as convex, this allows us to use conic solvers such as SCS [75], ECOS [27], and MOSEK [6] in CVXPY to solve for $q^*$ and obtain PID values.

## B.2 Alternate Methods for Working with Continuous $X_1$, $X_2$:

Another method for discretization is based on nearest neighbors [80, 11, 104], which has the advantage of performing density estimation as an intermediate step, but is ill-suited for our purposes of approximating $q^*$. Pakman et al. [76] attempt to optimize $q^*$ directly in the space of continuous distribution by approximating an optimal *copula*, which canonically captures dependencies between $X_1$, $X_2$ in $q$. However, this is nontrivial due to the space of possible joint distributions, and the authors restrict themselves to Gaussian copulas.

## B.3 Estimator 2: BATCH for Continuous Representations

Our next estimator, BATCH, is suitable for large datasets where the support of $X_1$ and $X_2$ are high-dimensional continuous vectors and not enumerable. Instead, we propose an end-to-end model parameterizing the joint distribution and a training objective whose solution allows us to obtain PID. Here $|\mathcal{X}_i| = |\mathcal{Y}| = \infty$. However, we wish to estimate the PID values given a sampled dataset $\mathcal{D} = \{(x_1^{(j)}, x_2^{(j)}, y^{(j)})\}$ of size $n$.

In this appendix section, we provide full details deriving our BATCH estimator, see Figure 9 for a high-level overview. We first rewrite the optimization objective in (18) before defining neural estimators to approximate high-dimensional variables that we cannot compute exactly, before optimizing the (18) using learned models and obtaining the PID values.

**Objective function**: First, we observe that the optimization objective in (18) can be written as:

$$q^* = \operatorname*{arg\,min}_{q \in \Delta_p} I_q(\{X_1, X_2\}; Y) \tag{21}$$

$$= \operatorname*{arg\,min}_{q \in \Delta_p} \mathbb{E}_{x_1, x_2, y \sim q} \left[ \log \frac{q(x_1, x_2, y)}{q(x_1, x_2) q(y)} \right] \tag{22}$$

$$= \operatorname*{arg\,min}_{q \in \Delta_p} \mathbb{E}_{x_1, x_2, y \sim q} \left[ \log \frac{q(x_1, x_2 | y)}{q(x_1, x_2)} \right] \tag{23}$$

$$= \operatorname*{arg\,min}_{q \in \Delta_p} \mathbb{E}_{x_1, x_2, y \sim q} \left[ \log \frac{q(x_2 | x_1, y) q(x_1 | y)}{q(x_2 | x_1) q(x_1)} \right] \tag{24}$$

$$= \operatorname*{arg\,min}_{q \in \Delta_p} \mathbb{E}_{\substack{x_1, y \sim q(x_1, y) \\ x_2 \sim q(x_2 | x_1, y)}} \left[ \log \frac{q(x_2 | x_1, y) q(x_1 | y)}{\sum_{y' \in Y} q(x_2 | x_1, y') q(y' | x_1) q(x_1)} \right] \tag{25}$$

Our goal is to approximate the optimal $\tilde{q}$ that satisfies this problem. Note that by the marginal constraints, $q(x_1, y) = p(x_1, y)$ and $q(y' | x_1) = p(y' | x_1)$. The former can be obtained by sampling from the dataset and the latter is an unimodal model independent of $q$.

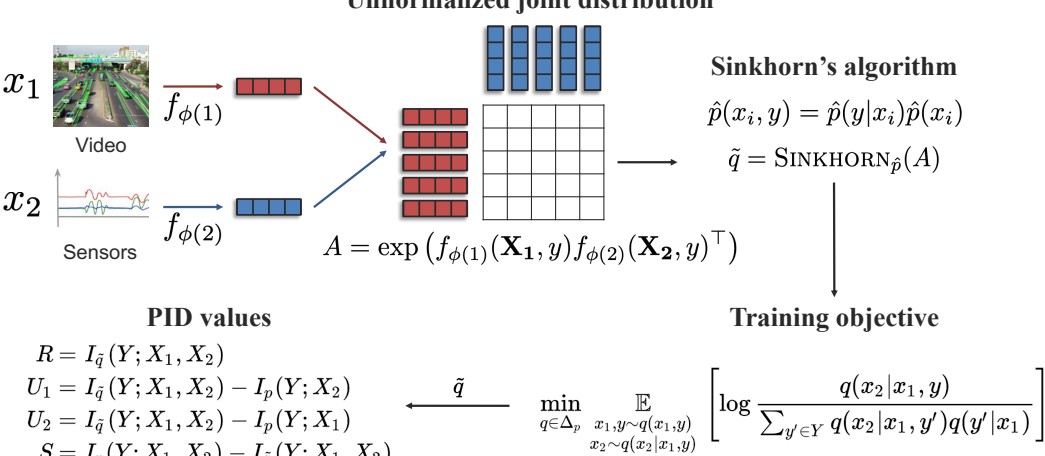

Figure 9: Overview of BATCH estimator: We define an end-to-end model that takes in batches of features $\mathbf{X_1} \in \widetilde{\mathcal{X}}_1^m, \mathbf{X_2} \in \widetilde{\mathcal{X}}_2^m, \mathbf{Y} \in \mathcal{Y}^m$ and learns a matrix $A \in \mathbb{R}^{m \times m \times |\mathcal{Y}|}$ to represent the unnormalized joint distribution $\tilde{q}$. Following ideas in multimodal alignment [99] and attention methods [103] that learn approximate distributions over input data, we parameterize $A$ by computing the outer-product similarity matrix over learned features. To ensure that $\tilde{q} \in \Delta_p$, we use an unrolled version of Sinkhorn's algorithm [24] which projects $A$ onto $\Delta_p$ by iteratively normalizing all rows and columns to sum to 1 and rescaling to satisfy the marginals $\hat{p}$. We plug in the learned $\tilde{q}$ into our objective in (25) and update trainable neural network parameters $\phi$ using projected gradient descent. Upon convergence, we extract PID by approximating $I_{\tilde{q}}(\{X_1, X_2\}; Y)$ by sampling and plugging into (1)-(3). This use of a mini-batch of size $m$ can be seen as an approximation of full-batch gradient descent over the entire dataset of size $n \gg m$. While it is challenging to obtain an unbiased estimator of the full-batch gradient since computing the full $A$ is intractable, we found our approach to work in practice for large $m$. Our approach can also be informally viewed as performing amortized optimization [3] by using $\phi$ to implicitly share information about the full batch using subsampled batches.

**Neural architecture**: To scale to continuous or unseen $x_i$'s, we define a function approximator that takes in $x_1$, $x_2$ and $y$ and outputs an unnormalized joint density $\tilde{q}(x_1, x_2, y)$. Given batches $\mathbf{X_1} \in \widetilde{\mathcal{X}}_1^m, \mathbf{X_2} \in \widetilde{\mathcal{X}}_2^m, \mathbf{Y} \in \mathcal{Y}^m$, we define learnable encoders $f_{\phi(1)}$ and $f_{\phi(2)}$ that output a matrix $A \in \mathbb{R}^{m \times m \times |\mathcal{Y}|}$ to represent the unnormalized joint distribution $\tilde{q}$, i.e., we want $A[i][j][y] = \tilde{q}(\mathbf{X_1}[i], \mathbf{X_2}[j], y)$. Following ideas in multimodal alignment [99] and attention methods [103] in deep learning that learn approximate distributions over input data, we also parameterize our neural architecture by computing the outer-product similarity matrix over learned features:

$$A = \exp\left(f_{\phi(1)}(\mathbf{X_1}, y) f_{\phi(2)}(\mathbf{X_2}, y)^\top\right) \tag{26}$$

**Constraining $A$ using Sinkhorn's algorithm**: To ensure that the learned $A$ is a valid pdf and respects the marginal constraints $\tilde{q}(x_i, y) = p(x_i, y)$, we use the Sinkhorn-Knopp algorithm [24], which projects $A$ into the space of non-negative square matrices that are row and column-normalized to satisfy the desired marginals $p(x_1, y)$ and $p(x_2, y)$. Sinkhorn's algorithm enables us to perform this projection by iteratively normalizing all rows and columns of $A$ to sum to 1 and rescaling the rows and columns to satisfy the marginals. Since $p(x_i, y)$ is not easy to estimate for continuous $x_i$, we compute the marginals using Bayes' rule by expanding it as $p(x_i)p(y|x_i)$. Note that by sampling $x_i$ from the dataset, the rows and columns of $A$ are already distributed as $p(x_i)$. The only remaining term needed is $p(y|x_i)$, for which we use unimodal models $\hat{p}(y|x_i)$ trained before running the estimator and subsequently frozen. Finally, Sinkhorn's algorithm is performed by normalizing each row to $\hat{p}(y|x_1)$ and normalizing each column to $\hat{p}(y|x_2)$.

**Optimization of the objective**: We obtain the value of $\tilde{q}(x_2|x_1, y)$ by marginalizing from the above distribution $\tilde{q}(x_1, x_2|y)$. Given matrix $A$ representing $\tilde{q}(x_1, x_2, y)$, the remaining terms can all be computed in closed form through appropriate summation across dimensions in $A$ to obtain $\tilde{q}(x_i)$,

$\tilde{q}(x_1, x_2)$, $\tilde{q}(x_i|y)$, and $\tilde{q}(x_1, x_2|y)$. We now have all the terms for the objective in (25). We optimize the objective via gradient descent.

Our final algorithm for training BATCH is outlined in Algorithm 1.

**Extracting PID values from the learned** $\tilde{q}$: To extract the PID values from a learned distribution of $\tilde{q}$, we first expand the definition of mutual information under $p$ and $\tilde{q}$, and calculate the PID values from their definitions in (1)-(3):

$$I_p(Y; X_i) = \mathop{\mathbb{E}}_{x_i, y \sim p}\left[\log\left(\frac{\hat{p}(y|x_i)}{\hat{p}(y)}\right)\right] \tag{27}$$

$$I_p(Y; X_1, X_2) = \mathop{\mathbb{E}}_{x_1, x_2, y \sim p}\left[\log\left(\frac{\hat{p}(y|x_1, x_2)}{\hat{p}(y)}\right)\right] \tag{28}$$

$$I_{\tilde{q}}(Y; X_1, X_2) = \mathop{\mathbb{E}}_{x_1, x_2, y \sim \tilde{q}}\left[\log\left(\frac{\tilde{q}(x_2|x_1, y)\hat{p}(y|x_1)}{\hat{p}(y)\sum_{y'}\tilde{q}(x_2|x_1, y')\hat{p}(y'|x_1)}\right)\right] \tag{29}$$

$$R = I_{\tilde{q}}(Y; X_1, X_2) \tag{30}$$

$$U_1 = I_{\tilde{q}}(Y; X_1, X_2) - I_p(Y; X_2) \tag{31}$$

$$U_2 = I_{\tilde{q}}(Y; X_1, X_2) - I_p(Y; X_1) \tag{32}$$

$$S = I_p(Y; X_1, X_2) - I_{\tilde{q}}(Y; X_1, X_2) \tag{33}$$

---

**Algorithm 1** BATCH algorithm.

---

**Require:** Multimodal dataset $\mathbf{X_1} \in \widetilde{\mathcal{X}}_1^n, \mathbf{X_2} \in \widetilde{\mathcal{X}}_2^n, \mathbf{Y} \in \mathcal{Y}^n$.

---

Initialize feed-forward networks $f_{\phi(1)}, f_{\phi(2)}$.
**while** not converged **do**
    **for** sampled batch $\mathbf{X_1} \in \widetilde{\mathcal{X}}_1^m, \mathbf{X_2} \in \widetilde{\mathcal{X}}_2^m, \mathbf{Y} \in \mathcal{Y}^m$ **do**
        $h_1[:, y] \leftarrow f_{\phi(1)}(x_1, y), h_2[:, y] \leftarrow f_{\phi(2)}(x_2, y)$
        $A[:, :, y] \leftarrow \exp(h_1[:, y]h_2[:, y]^\top)$
        $\tilde{p}(y|x_i) \leftarrow$ unimodal predictions.
        $A \leftarrow \text{SINKHORN}_{\hat{p}}(A)$
        Marginalize and sample $(x_1, x_2, y)$ from $A$.
        Calculate the loss from (25) using $(x_1, x_2, y)$.
        Perform a gradient step on the loss.
    **end for**
**end while**
**return** network weights in $f_{1,j}, f_{2,j}$ for $j \in [C]$

---

**Important assumptions and caveats**: As a note, the above formulation assumes the existence of unimodal models $\hat{p}(y|x_i)$. We train these models independently and freeze them when training the above estimator. After obtaining $\tilde{q}$, we extract the PID values. To do so, we need an additional multimodal model to estimate $\hat{p}(y|x_1, x_2)$ due to (28). Because part of our goal is multimodal model selection, we must assume that the existing multimodal model might not be expressive enough and could underestimate (28). However, this value is only subsequently used to calculate $S$, leading to a possibly underestimated $S$, but leaving all other PID values unaffected by the choice of the multimodal model. In practice, the possibility of $S$ being larger than estimated should be considered, and more complex models should be investigated until $S$ stops increasing. Empirically, we have observed that measured value of $S$ by BATCH is generally larger than that measured by CVX; the underestimation is not as large as in the case of clustering in CVX.

In additional to the above, we note that the learned joined distribution $\tilde{q}(x_1, x_2, y)$ is limited by the expressivity of the neural network and similarity matrix method used in our model. Empirically, the PID values are not sensitive to hyperparameters, as long as the network is trained to convergence. Finally, our approximation using batch sampling limits the possible joined distributions to alignments within a batch, which we alleviates by using large batch size (256 in our experiments).

**Implementation Details**: We separately train the unimodal and multimodal models from the dataset using the best model from Liang et al. [60] and freeze them when running Algorithm 1. For the

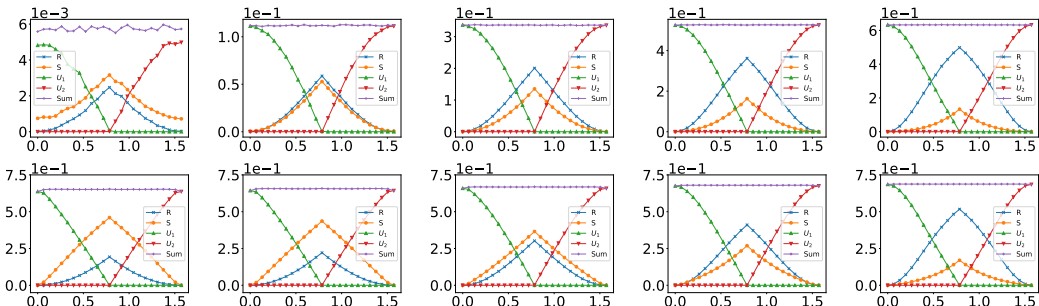

Figure 10: Results for PID decomposition under Cartesian coordinates. Within each plot: PID values for GMMs for $\angle\mu$ in $\{\frac{i}{40} \cdot \frac{\pi}{2} | i \in \{0, \ldots, 40\}\}$. ✖, ▲, ▼, ●, ✚ represent the $R$, $U_1$, $U_2$, $S$ and their sum respectively. $(r, \theta)$ corresspond to the unique information $(U_1, U_2)$ respectively. Across columns: As $||\mu||_2$ varies from $\{0.1, 0.5, 1.0, 1.5, 2.0\}$. Top-Bottom: Soft and Hard labels respectively. Note the scale of the $y$-axis for Soft. The overall trend for each PID component as $\angle\mu$ varies is similar across $||\mu||_2$. However, the *relative contribution* of Synergy and Redundancy changes significantly as $||\mu||_2$ varies, particularly for Hard.

neural network in Algorithm 1, we use a 3-layer feedforward neural network with a hidden dimension of 32. We train the network for 10 epochs using the Adam optimizer with a batch size of 256 and learning rate of 0.001.

**To summarize**, we propose 2 scalable estimators for PID, each with their approximations for the parameterization of $q$ and procedure for the estimation of PID.

1. Parameterizing $q$ explicitly via a joint probability matrix over $\Delta_p$, and optimizing convex objective (5) with linear constraints. This can only be used for finite $\mathcal{X}_1, \mathcal{X}_2, Y$, or after preprocessing by histogramming continuous domains.

2. Parameterizing $q$ via neural networks and optimize their parameters $\phi$ by repeatedly subsampling from data; for each subsample, we perform one step of projected gradient descent (using Sinkhorn's method) as if this subsample was the full dataset.

## C  Experiments

### C.1  Synthetic Bitwise Features

We show results on estimating PID values for OR, AND, and XOR interactions between $X_1$ and $X_2$ in Table 1 and find that both our estimators exactly recover the ground truth results in Bertschinger et al. [13].

### C.2  Gaussian Mixture Models (GMM)

**Extended Results:** For each of Cartesian and Polar we consider two cases, Soft-labels, when we observe $Y$, and Hard-labels, where we use $\hat{Y}$ predicted by the optimal linear classifier $\hat{y} = \text{sign}\left(\langle(x_1, x_2), \mu\rangle\right)$. We hope this will provide additional intuition to the bitwise modality experiments. This results are given in Figures 10, 11. Most of the trends are aligned with what we would expect from Redundancy, Uniqueness, and Synergy. We list the most interesting observations here.

- In all cases, total information (sum of PID values) is independent of $\angle\mu$. This is a natural consequence of rotational symmetry. It is also higher for higher $||\mu||_2$, e.g., 2.0, reflecting the fact that $X_1$, $X_2$ contain less information about labels when their centroids are close. Lastly, observe, that when $||\mu||_2$ increases, the total information approaches $\log(2)$, which is all the label information. This perfectly aligned with our intuition: if the cluster centroids are far away, then knowing a datapoint's cartesian coordinates will let us predict its label accurately (which is one bit of information).
- For Cartesian, unique information dominates when angle goes to 0 or $\pi/2$. This is sensible, if centroids share a coordinate, then observing that coordinate yields no information about $y$. Conversely, synergy and redundancy peak at $\pi/4$.

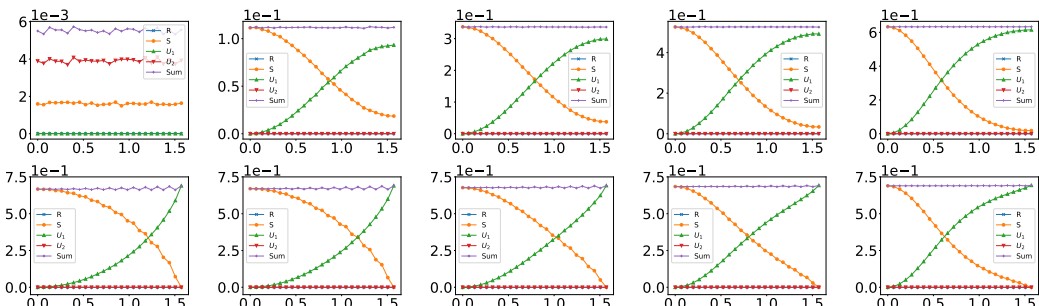

Figure 11: Results for PID decomposition under Polar coordinates where $\theta \in [0, \pi]$ and $r \in \mathbb{R}$, i.e., angle is restricted to the first two quadrants but distance could be negative. Within each plot: PID values for GMMs for $\angle\mu$ in $\{\frac{i}{40} \cdot \frac{\pi}{2} | i \in \{0, \ldots, 40\}\}$. ✖, ▲, ▼, ●, ✚ represent the $R$, $U_1$, $U_2$, $S$ and their sum respectively. Across columns: As $||\mu||_2$ varies from $\{0.1, 0.5, 1.0, 1.5, 2.0\}$. Top-Bottom: Soft and Hard labels respectively. Note the scale of the $y$-axis for Soft. The overall trend for each PID component as $\angle\mu$ varies is similar across $||\mu||_2$, except for when $||\mu||_2$ is small in Soft. However, the *relative contribution* of Synergy and Redundancy changes significantly as $||\mu||_2$ varies, particularly for Hard.

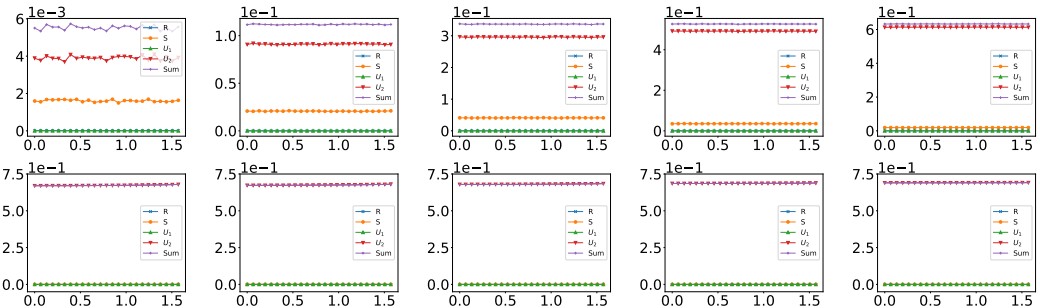

Figure 12: Results for PID decomposition under Polar coordinates, using the normal parameterization where $\theta \in [-\pi, \pi]$ and $r \geq 0$. Within each plot: PID values for GMMs for $\angle\mu$ in $\{\frac{i}{40} \cdot \frac{\pi}{2} | i \in \{0, \ldots, 40\}\}$. ✖, ▲, ▼, ●, ✚ represent the $R$, $U_1$, $U_2$, $S$ and their sum respectively. $(r, \theta)$ corresspond to the unique information $(U_1, U_2)$ respectively. Across columns: As $||\mu||_2$ varies from $\{0.1, 0.5, 1.0, 1.5, 2.0\}$. Top-Bottom: Soft and Hard labels respectively. Note the scale of the $y$-axis for Soft. Observe that PID values are independent of $\angle\mu$, with unique information from $\theta$ dominating.

- Redundancy dominates Synergy as $||\mu||_2$ increases. Interestingly, synergy seems to be independent of $||\mu||_2$ (one has to account for the scale in the vertical axis to recognize this).
- Hard contains $0.65 \approx \log(2)$ nats of total information, since $Y|X_1, X_2$ is deterministic. The deficit of $0.04$ nats is due to discretization near the decision boundary, and is expected to vanish if we increase the number of bins (though this ought to be accompanied by an increase in samples used to approximate PID).
- However, for Soft, total information is nearly 0 for small $||\mu||_2$ (and approaches $\log(2)$ when large). When the two cluster centroids are close, it is difficult to glean any information about the (soft) label, while we can predict $y$ accurately from both $x_1, x_2$ when centroids are far apart.
- For Polar, redundancy is 0. Furthermore, $\theta$ contains no unique information, since $\theta$ shows nothing about $y$ unless we know $r$ (in particular, its sign). When angle goes to $\pi/2$, almost all information is unique in $r$.

Lastly, we also experimented on the more common definition of polar coordinates, with $\theta \in [-\pi, \pi]$ and $r \geq 0$ (that is, angle is allowed to be negative but lengths have to be nonnegative). The results are in Figure 12. As one may expect, there is nothing exciting going on for both Softand Hard. For Hard, the optimal classifier is precisely expressed in terms of $\theta$ only, so all the information belongs to $U_2$, i.e., $\theta$. For Soft, the same trend remains: virtually all information is in $U_2$, i.e., $\theta$, with the trend becoming even sharper as $||\mu||_2$ increases. Intuitively, when the 2 centroids are extremely far

apart, then the optimal linear classifier is extremely accurate; which may be expressed by $\theta$ alone without any knowledge of $r$. Conversely, when the centroids are close, $r$ can provide some synergy with $\theta$—the larger $r$ is, the more "confident" we are in predictions based on $\theta$.

**Visualization of $q^*$:** The following plots of $q^*$ are provided to provide some intuition behind the objectives in (16). We focus on Cartesian for ease of visualization. For each $||\mu||_2 \in \{0.5, 1.0, 2.0\}$, we vary the angle $\angle\mu$ and sample $1e7$ labelled datapoints. We then compare $q^*$ (for both Soft and Hard) with the empirical distribution $p$. To compute PID, we use 50 bins evenly spaced between $[-5, 5]$ for both dimensions. The results for varying $||\mu||_2$ are shown in Figures 13-15.

When $\angle\mu$ is 0, there are few differences, barring a slightly "blockier" feel for $q^*$ over $p$. However, as $\angle\mu$ increases, $q^*$ gets skewed rather than rotated, becoming "thinner" in the process. The limiting case is when $\angle\mu_2 = \pi/4$, i.e., 45 degrees. Here, the objective diverges to infinity, with what appears to be a solution with nonnegative entries only along the diagonal—this observation appears to hold even as we increase the number of bins. We hypothesize that in the truly continuous case (recall we are performing discretization for this set of experiments), the optimum is a degenerate "one dimensional" GMM with mass located only at $q(x, x)$.

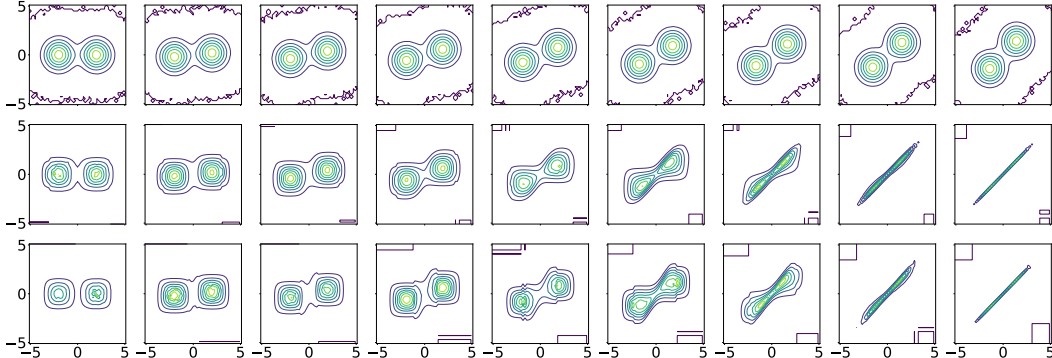

Figure 13: Densities for GMMs where $||\mu||_2 = 2.0$. Top to Bottom: Densities for $p$, $q^*$ (Soft), and $q^*$ (Hard). Left to right, $\angle\mu$ in $\{\frac{i}{8} \cdot \frac{\pi}{4} | i \in \{0, \ldots, 8\}\}$

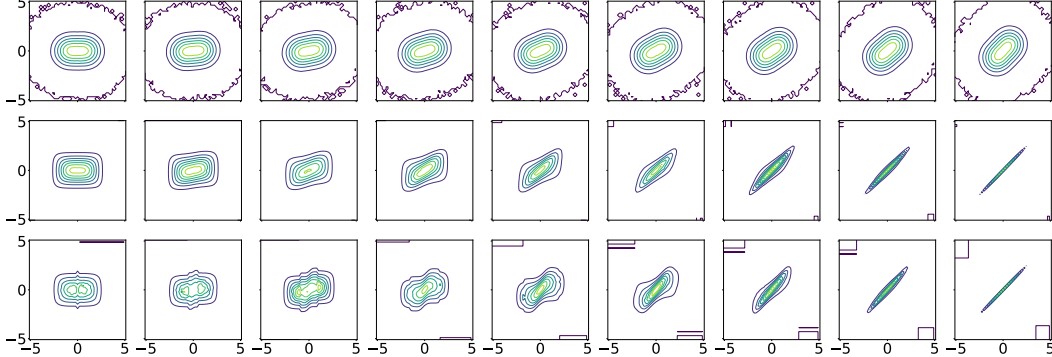

Figure 14: Densities for GMMs where $||\mu||_2 = 1.0$. Top to Bottom: Densities for $p$, $q^*$ (Soft), and $q^*$ (Hard). Left to right, $\angle\mu$ in $\{\frac{i}{8} \cdot \frac{\pi}{4} | i \in \{0, \ldots, 8\}\}$

### C.3 Synthetic Generative Model

Our third setting simulates a data generation process where there exists a set of latent concepts that either redundantly, uniquely, or synergistically determine the label.

**Setup**: We begin with a fixed set of latent variables $z_1, z_2, z_c \in \mathbb{R}^{50}$ from $\mathcal{N}(0, \sigma^2)$ with $\sigma = 0.5$ representing latent concepts for modality 1, modality 2, and common information respectively. The concatenated variables $[z_1, z_c]$ are transformed into high-dimensional $x_1 \in \mathbb{R}^{100}$ data space using a fixed transformation matrix $T_1 \in \mathbb{R}^{50 \times 100}$ and likewise $[z_2, z_c]$ to $x_2$ via $T_2$. The label $y$ is generated

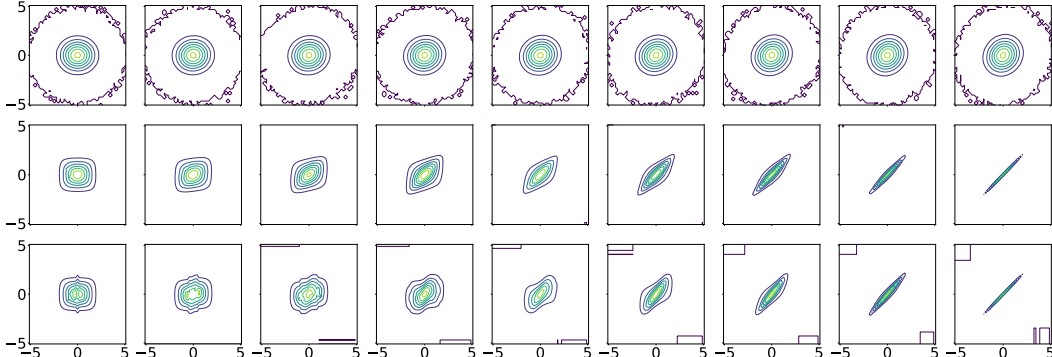

Figure 15: Densities for GMMs where $||\mu||_2 = 0.5$. Top to Bottom: Densities for $p$, $q^*$ (Soft), and $q^*$ (Hard). Left to right, $\angle\mu$ in $\{\frac{i}{8} \cdot \frac{\pi}{4} | i \in \{0, \ldots, 8\}\}$

as a transformation function of (1) only $z_c$, in which case we expect redundancy in $x_1$ and $x_2$ with respect to $y$, (2) only $z_1$ or $z_2$ which represents uniqueness in $x_1$ or $x_2$ respectively, (3) the concatenated $[z_1, z_2]$ representing synergy, or (4) arbitrary ratios of each of the above. We also introduce nonlinearity and random noises to increase data complexity. In formal notation,

$$y = \left[\text{sigmoid}\left(\frac{\sum_{i=0}^{n} f(\mathbf{z})_i}{n}\right) \geq 0.5\right]$$

where $f$ is a fixed nonlinear transformation with dropout ($p = 0.1$) and $\mathbf{z}$ can be $z_1, z_2, z_c$ or any combination according to the cases described above. These datasets are designed to test a combination of feature learning with predefined interactions between the modalities and the label.

We generated 10 synthetic datasets as shown in Table 2. The first four are specialized datasets denoted as $\mathcal{D}_m$, $m \in \{R, U_1, U_2, S\}$ for which the label $y$ only depends on redundancy, uniqueness, or synergy features respectively. The rest are mixed datasets with $y$ generated from $z_1, z_2, z_c$ of varying dimensions. We use $z_i^* \in \mathbb{R}^{50}$ to denote a randomly sub-sampled feature from $z_i$. For example, to generate the label of one data point $(x_1^{(j)}, x_2^{(j)})$ in the mixed dataset $y = (z_1^*, z_2^*, z_c)$, we sub-sampled $(z_1^*)^{(j)}, (z_2^*)^{(j)}$ from $z_1, z_2$ and generate $y$ from $[(z_1^*)^{(j)}, (z_2^*)^{(j)}, z_c]$ following the process above.

To report the ground-truth interactions for these datasets, we first train multimodal models and obtain an estimate of test performance on them. We then convert $P_{\text{acc}}$, the test accuracy of the best model on these datasets into the mutual information between the inputs to the label using the bound in Feder and Merhav [29]:

$$I(X_1, X_2; Y) \leq \log P_{\text{acc}} + H(Y) \tag{34}$$

Since the test accuracies for Table 2 datasets range from 67-75%, this corresponds to a total MI of $0.42 - 0.59$ bits (i.e., from $\log_2 0.67 + 1$ to $\log_2 0.75 + 1$). The information in each interaction is then computed by dividing this total mutual information by the interactions involved in the data generation process: if the total MI is $0.6$ bits and the label depends on half from the common information between modalities and half from the unique information in $x_1$, then the ground truth $R = 0.3$ and $U_1 = 0.3$.

**Results**: In the main paper, we note the agreement between CVX and BATCH and their consistency with the dataset (Table 2). In addition, we note that the sum of PID values (which sum to $I_p(Y; X_1, X_2)$) is generally larger in BATCH than in CVX, which accounts for the information loss during CVX's clustering. Moreover, in the last two cases ($y = f(z_2^*, z_c^*)$ and $y = f(z_2^*, z_c)$), CVX predicts non-zero amount of $S$ despite the dataset being synergy-free by construction. Because the total mutual information is small, the spurious synergy becomes a large portion of the total PID values. This potentially demonstrates that when the total mutual information is small, spurious information gain could have a large impact on the results.

### C.4 Quantifying Multimodal Datasets

**Setup**: We use a large collection of real-world datasets in MultiBench [60] which test *multimodal fusion* of different input signals and requires representation learning of complex real-world interactions

Table 6: Full results on estimating PID on real-world MultiBench [60] datasets. Many of the estimated interactions align well with dataset construction (e.g., MUSTARD for sarcasm) and unimodal performance (e.g., MIMIC and AV-MNIST). On QA [5] datasets, synergy is consistently the highest.

| Task | AV-MNIST | | | | ENRICO | | | | VQA 2.0 | | | | CLEVR | | | |
|---|---|---|---|---|---|---|---|---|---|---|---|---|---|---|---|---|
| Measure | $R$ | $U_1$ | $U_2$ | $S$ | $R$ | $U_1$ | $U_2$ | $S$ | $R$ | $U_1$ | $U_2$ | $S$ | $R$ | $U_1$ | $U_2$ | $S$ |
| CVX | 0.10 | **0.97** | 0.03 | 0.08 | **0.73** | 0.38 | 0.53 | 0.34 | 0.79 | 0.87 | 0 | **4.92** | 0.55 | 0.48 | 0 | **5.16** |

| Task | MIMIC | | | | UR-FUNNY A,T | | | | UR-FUNNY V,T | | | | UR-FUNNY V,A | | | |
|---|---|---|---|---|---|---|---|---|---|---|---|---|---|---|---|---|
| Measure | $R$ | $U_1$ | $U_2$ | $S$ | $R$ | $U_1$ | $U_2$ | $S$ | $R$ | $U_1$ | $U_2$ | $S$ | $R$ | $U_1$ | $U_2$ | $S$ |
| BATCH | 0.05 | **0.17** | 0 | 0.01 | 0.02 | 0 | **0.08** | 0.07 | 0.06 | 0 | 0.04 | **0.11** | 0.02 | 0.04 | 0 | **0.06** |

| Task | MOSEI A,T | | | | MOSEI V,T | | | | MOSEI V,A | | | |
|---|---|---|---|---|---|---|---|---|---|---|---|---|
| Measure | $R$ | $U_1$ | $U_2$ | $S$ | $R$ | $U_1$ | $U_2$ | $S$ | $R$ | $U_1$ | $U_2$ | $S$ |
| BATCH | 0.22 | 0.04 | 0.09 | **0.13** | 0.30 | **0.70** | 0 | 0 | 0.26 | **0.74** | 0 | 0 |

| Task | MUSTARD A,T | | | | MUSTARD V,T | | | | MUSTARD V,A | | | |
|---|---|---|---|---|---|---|---|---|---|---|---|---|
| Measure | $R$ | $U_1$ | $U_2$ | $S$ | $R$ | $U_1$ | $U_2$ | $S$ | $R$ | $U_1$ | $U_2$ | $S$ |
| BATCH | 0.14 | 0.01 | 0.01 | **0.37** | 0.14 | 0.02 | 0.01 | **0.34** | 0.14 | 0.01 | 0.01 | **0.20** |

for different tasks. MultiBench spans 10 diverse modalities (images, video, audio, text, time-series, various robotics sensors, sets, and tables), 15 prediction tasks (humor, sentiment, emotions, mortality rate, ICD-9 codes, image-captions, human activities, digits, robot pose, object pose, robot contact, and design interfaces), and 5 research areas (affective computing, healthcare, multimedia, robotics, and HCI). These datasets are designed to test a combination of feature learning and arbitrary complex interactions between the modalities and the label in the real world. We also include experiments on *question-answering* (Visual Question Answering [5, 39] and CLEVR [53]) which test grounding of language into the visual domain.

**Details of applying CVX**: For quantifying multimodal datasets with CVX, we first apply PCA to reduce the high dimension of multimodal data. For each of the train, validation, and test split, we then use unsupervised clustering to generate 20 clusters. We obtain a clustered version of the original dataset $\mathcal{D} = \{(x_1, x_2, y)\}$ as $\mathcal{D}_{\text{cluster}} = \{(c_1, c_2, y)\}$ where $c_i \in \{1, \ldots, 20\}$ is the ID of the cluster that $x_i$ belongs to. For datasets with 3 modalities we estimate PID separately for each of the 3 modality pairs.

**Details of applying BATCH**: For quantifying multimodal datasets with BATCH, we first use pretrained encoders to obtain the image and text features, and then train 2 unimodal and 1 multimodal classifiers on those features to predict the original answer. The final PID values are computed using the encoded features and the 3 classifiers. In our experiments we used the pretrained image and text encoders from the CLIP-VIT-B/32 [82] model, and the unimodal and multimodal classifiers are 2-layer MLPs.

**Results on multimodal fusion**: We show full results on quantifying datasets in Table 6. For multimodal fusion, we find that different datasets do require different interactions. Among interesting observations are that (1) all pairs of modalities on sarcasm detection shows high synergy values, which aligns with intuition on sarcasm in human communication, (2) uniqueness values are strongly correlated with unimodal performance (e.g., AV-MNIST and MIMIC first modality), (3) datasets with high synergy benefit from interaction modeling (e.g., sarcasm), and (4) conversely datasets with low synergy are also those where modeling in higher-order interactions do not help (e.g., MIMIC).

To test grounding of language into the visual domain, we also run experiments on the VQA v2.0 [5, 39] dataset, which contains 443757 training and 214354 validation data with real-life images, and the CLEVR [53] dataset, which contains 699960 training and 149991 validation data with rendered images and generated questions. The test spilt for both datasets are not used because the labels are not publicly available. Note that for both validation datasets, we first filter out all data instances where the label is neither "yes" nor "no". On the remaining data, we use the outputs from the last feature layer of a VGG19 [87] model as image embeddings and use RoBERTa [64] to obtain the text embeddings for questions, and then apply K-means clustering to both embeddings to obtain a clustered version

Table 7: Estimating PID on QA [5] datasets with both CVX and BATCH estimators. Synergy is consistently the highest and both estimators return PID values that are consistent relative to each other.

| Task | VQA 2.0 | | | | CLEVR | | | |
|---|---|---|---|---|---|---|---|---|
| Measure | $R$ | $U_1$ | $U_2$ | $S$ | $R$ | $U_1$ | $U_2$ | $S$ |
| CVX | 0.79 | 0.87 | 0 | **4.92** | 0.55 | 0.48 | 0 | **5.16** |
| BATCH | 0.10 | 0.14 | 0.09 | **0.16** | 0.01 | 0.01 | 0.01 | **0.03** |
| Human | 0.23 | 0 | 0 | **4.80** | 0 | 0 | 0 | **5.00** |

| Instruction:
Look at the data and provide your rating for the following questions on a scale of 0 (none at all) - 5 (large extent)

and score your confidence in your answer from a scale of 0 (no confidence) - 5 (high confidence): | | | 1. The extent to which both modalities enable you to make the same predictions about the task. | | 2. The extent to which modality 1 enables you to make a prediction about the task that you would not if using modality 2. | | 3. The extent to which modality 2 enables you to make a prediction about the task that you would not if using modality 1. | | 4. The extent to which both modalities enable you to make a prediction about the task that you would not otherwise make using either modality individually. | |
|---|---|---|---|---|---|---|---|---|---|---|
| ID | Modality 1 | Modality 2 | 1 Rating | 1 Confidence | 2 Rating | 2 Confidence | 3 Rating | 3 Confidence | 4 Rating | 4 Confidence |
| mosei1 | video | The devil deceives false Christians into thinking they are already safe and secure and on their way to heaven. | | | | | | | | |
| mosei2 | video | Currently, over four-hundred thousand Americans with disabilities are being paid less than the minimum wage, some of them mere pennies per hour. | | | | | | | | |

Figure 16: We show a sample user interface for annotating information decomposition into redundancy, uniqueness, and synergy, which we give to 3 annotators each to obtain agreeing and reliable judgments of each interaction for real-world multimodal datasets.

of the original dataset $\mathcal{D} = \{(x_1, x_2, y)\}$ as $\mathcal{D}_{\text{cluster}} = \{(c_1, c_2, y)\}$, where $c_i \in \{1, \ldots, 20\}$ are the cluster IDs. We then calculate the PID values on the clustered dataset $\mathcal{D}_{\text{cluster}}$.

**Results on QA**: As a whole, we observe consistently high synergy values on QA datasets as shown in Table 6. This is consistent with prior work studying how these datasets were balanced (e.g., VQA 2.0 having different images for the same question such that both yes/no answers are observed in the dataset so that the answer can only be obtained through synergy) [39] and models trained on these datasets [46].

**Comparing CVX and BATCH**: We carefully compare CVX and BATCH on the 2 question answering datasets VQA and CLEVR, showing these results in Table 7. We choose these two datasets since their input modalities are images and text, which make them suitable for both clustering of representations extracted from pre-trained before input into CVX, and end-to-end encoding and PID estimation with BATCH. We also include human judgment of PID on these 2 datasets for comparison. From Table 7, we find that all three measures agree on the largest interaction, as well as the relative order of interaction values. We find that the absolute values of each interaction can be quite different due to the unavoidable information loss of both input modalities to the label during clustering in CVX or representation learning in BATCH. Hence, we emphasize that these estimated interactions are most useful through relative comparison to judge which interaction is most dominant in a particular dataset.

### C.5 Comparisons with Human Judgment

To quantify information decomposition for real-world tasks, we investigate whether human judgment can be used as a reliable estimator. We propose a new annotation scheme where we show both modalities and the label and ask each annotator to annotate the degree of redundancy, uniqueness, and synergy on a scale of 0-5 using the following definitions inspired by the formal definitions in information decomposition:

1. $R$: The extent to which using the modalities individually and together gives the same predictions on the task,
2. $U_1$: The extent to which $x_1$ enables you to make a prediction about the task that you would not if using $x_2$,
3. $U_2$: The extent to which $x_2$ enables you to make a prediction about the task that you would not if using $x_1$,
4. $S$: The extent to which only both modalities enable you to make a prediction about the task that you would not otherwise make using either modality individually,

alongside their confidence in their answers on a scale of 0-5. We show a sample user interface for the annotations in Figure 16.

Table 8: Collection of datasets used for our study of multimodal fusion interactions covering diverse modalities, tasks, and research areas in multimedia and affective computing.

| Datasets | Modalities | Size | Prediction task | Research Area |
|---|---|---|---|---|
| MUSTARD [17] | {text, video, audio} | 690 | sarcasm | Affective Computing |
| UR-FUNNY [43] | {text, video, audio} | 16,514 | humor | Affective Computing |
| MOSEI [116] | {text, video, audio} | 22,777 | sentiment, emotions | Affective Computing |
| CLEVR [53] | {image, question} | 853,554 | QA | Multimedia |
| VQA 2.0 [39] | {image, question} | 1,100,000 | QA | Multimedia |

Table 9: Annotator agreement and average confidence scores of human annotations of PID values for real-world multimodal datasets.

| Task | Agreement | | | | Confidence | | | |
|---|---|---|---|---|---|---|---|---|
| Measure | $R$ | $U_1$ | $U_2$ | $S$ | $R$ | $U_1$ | $U_2$ | $S$ |
| Human | 0.72 | 0.68 | 0.70 | 0.72 | 4.36 | 4.35 | 4.27 | 4.46 |

Based on direct communication with our institution's IRB office, this line of user-study research is aligned with similar annotation studies at our institution that are exempt from IRB. The information obtained during our study is recorded in such a manner that the identity of the human subjects cannot readily be ascertained, directly or through identifiers linked to the subjects. We do not collect any identifiable information from annotators. Participation in all annotations was fully voluntary and we obtained consent from all participants prior to annotations. Participants were not the authors nor in the same research groups as the authors. They all hold or are working towards a graduate degree in a STEM field and have knowledge of machine learning. None of the participants knew about this project before their session and each participant only interacted with the setting they were involved in.

We sample 10 datapoints with class balance from each of the 5 datasets in Table 8 and give them to 3 annotators. All the data were manually selected and do not contain offensive content. We clarified all the confusion that the annotators may have about the instructions and the definitions above at the beginning of the annotation process. We summarize the results and key findings:

1. **General observations on interactions, agreement, and confidence**: The annotated interactions align with prior intuitions on these multimodal datasets and do indeed explain the interactions between modalities. For example, VQA and CLEVR are annotated with significantly high synergy, and language is annotated as the dominant modality in sentiment, humor, and sarcasm (high $U_1$ values). Overall, Table 9 shows that the Krippendorff's alpha for inter-annotator agreement in human annotations of the interactions is quite high and the average confidence scores are also quite high. This indicates that the human-annotated results are reliable.

2. **Annotator subjectivity in interpreting presence vs absence of an attribute and contextualizing both language and video**: We observe that there is some confusion between uniqueness in the language modality and synergy in the video datasets, resulting in cases of low agreement in annotating $U_1$ and $S$: $-0.09, -0.07$ for MOSEI, $-0.14, -0.03$ for UR-FUNNY and $-0.08, -0.04$ for MUSTARD respectively. We believe this is due to annotator subjectivity in interpreting whether sentiment, humor, and sarcasm are present only in the language modality ($U_1$) or present when contextualizing both language and video ($S$). We further investigated the agreement and confidence in the presence or absence of an attribute (e.g., humor or sarcasm). We examined videos that show and do not show an attribute separately and indeed found in general, humans reached higher agreement on annotating attribute-present videos. The agreement of annotating $S$ is 0.13 when the attribute is present, compared to $-0.10$ when absent.

## C.6 Comparisons with Other Interaction Measures

It is worth noting that other valid information-theoretic definitions of multimodal interactions also exist, but are known to suffer from issues regarding over- and under-estimation, and may even be negative [41]; these are critical problems with the application of information theory for shared

Table 10: Estimating PID on synthetic generative model datasets. Ground truth total information is computed based on an upper bound from the best multimodal test accuracy. Both CVX and BATCH measures agree with each other on relative values and are consistent with ground truth interactions. We also implement 3 other definitions: **I-min** can sometimes overestimate synergy and uniqueness; **WMS** is actually synergy minus redundancy, so can be negative and when R & S are of equal magnitude WMS cancels out to be 0; **CI** can also be negative and sometimes incorrectly concludes highest uniqueness for S-only data.

| Task | $R$-only data | | | | $U_1$-only data | | | | $U_2$-only data | | | | $S$-only data | | | |
|---|---|---|---|---|---|---|---|---|---|---|---|---|---|---|---|---|
| PID | $R$ | $U_1$ | $U_2$ | $S$ | $R$ | $U_1$ | $U_2$ | $S$ | $R$ | $U_1$ | $U_2$ | $S$ | $R$ | $U_1$ | $U_2$ | $S$ |
| I-MIN | 0.17 | 0.08 | 0.07 | 0 | 0 | 0.23 | 0 | 0.06 | 0 | 0 | 0.25 | 0.08 | 0.07 | 0.03 | 0.04 | **0.17** |
| WMS | 0 | 0.20 | 0.20 | −0.11 | 0 | 0.25 | 0.02 | 0.05 | 0 | 0.03 | **0.27** | 0.05 | 0 | 0.14 | 0.15 | 0.07 |
| CI | **0.34** | −0.09 | −0.10 | 0.17 | 0 | 0.23 | 0 | 0.06 | 0 | 0.01 | 0.25 | 0.07 | −0.02 | 0.13 | 0.14 | 0.08 |
| CVX | 0.16 | 0 | 0 | 0.05 | 0 | 0.16 | 0 | 0.05 | 0 | 0 | 0.17 | 0.05 | 0.07 | 0 | 0.01 | **0.14** |
| BATCH | **0.29** | 0.02 | 0.02 | 0 | 0 | **0.30** | 0 | 0 | 0 | 0 | **0.30** | 0 | 0.11 | 0.02 | 0.02 | **0.15** |
| Truth | 0.58 | 0 | 0 | 0 | 0 | 0.56 | 0 | 0 | 0 | 0 | 0.54 | 0 | 0 | 0 | 0 | 0.56 |

$I(X_1; X_2; Y)$ and unique information $I(X_1; Y|X_2)$, $I(X_2; Y|X_1)$ often quoted in the co-training [14, 8] and multi-view learning [98, 94, 89] literature. We additionally compare PID with other previously used measures for feature interactions on the synthetic generative model datasets. Both CVX and BATCH measures agree with each other on relative values and are consistent with ground truth interactions. We also implement 3 other definitions: **I-min** can sometimes overestimate synergy and uniqueness; **WMS** is actually synergy minus redundancy, so can be negative and when R & S are of equal magnitude WMS cancels out to be 0; **CI** can also be negative and sometimes incorrectly concludes highest uniqueness for $S$-only data. These results are in Table 10. We also tried 3 non-info theory measures: Shapley values, Integrated gradients (IG), and CCA, which are based on quantifying interactions captured by a multimodal model. Our work is fundamentally different in that interactions are properties of data before training any models.

### C.7 Quantifying Model Predictions

**Setup**: For each dataset, we first train a suite of models on the train set $\mathcal{D}_{\text{train}}$ and apply it to the validation set $\mathcal{D}_{\text{val}}$, yielding a predicted dataset $\mathcal{D}_{\text{pred}} = \{(x_1, x_2, \hat{y}) \in \mathcal{D}_{\text{val}}\}$. Running PID on $\mathcal{D}_{\text{pred}}$ summarizes the interactions that the model captures. We categorize and implement a comprehensive suite of models (spanning representation fusion at different feature levels, types of interaction inductive biases, and training objectives) that have been previously motivated in the literature to capture certain types of interactions [62]. In the following, we use $\mathbf{x}_i$ to denote data or extracted features from unimodal models where appropriate. We include all implementation details and hyperparameters in Table 11, 12, 13, 14, 15.

*General*: We first implement the most general model, which we call early fusion:

1. EF: Concatenating data at the earliest input level, essentially treating it as a single modality, and defining a suitable prediction model $y = f([\mathbf{x}_1, \mathbf{x}_2])$ [62].

*Redundancy*: Fusion at the decision level (i.e., *late fusion*) is often motivated to capture redundancy between unimodal predictors.

1. ADDITIVE: Suitable unimodal models are first applied to each modality before aggregating the outputs using an additive average: $y = 1/2(f_1(\mathbf{x}_1) + f_2(\mathbf{x}_2))$ [44].

2. AGREE: Prediction agreement $(+\lambda(f_1(\mathbf{x}_1) - f_2(\mathbf{x}_2))^2$ [26]) can be added for redundancy.

3. ALIGN: Feature alignment $(+\lambda \text{sim}(\mathbf{x}_1, \mathbf{x}_2)$ like contrastive learning [82]) can also be added for redundancy.

*Synergy*: Exemplified by methods such as bilinear pooling [33] and tensor fusion [115], researchers have long sought to learn higher-order interactions across modalities to capture synergistic interactions. We implement the following models:

1. ELEM: For static interactions (i.e., without trainable interaction parameters), we implement element-wise interactions: $y = f(\mathbf{x}_1 \odot \mathbf{x}_2)$ [2, 50].

2. TENSOR: We also implement outer-product interactions (i.e., higher-order tensors) $y = f\left(\mathbf{x}_1 \mathbf{x}_2^\top\right)$ [33, 115, 47, 58, 65].

3. MI: Dynamic interactions with learnable weights include multiplicative interactions $\mathbb{W}$: $y = f(\mathbf{x}_1 \mathbb{W} \mathbf{x}_2)$ [52].

4. MULT: Another example of dynamic interactions with learnable weights is Crossmodal self-attention typically used in multimodal Transformers: $y = f(\text{softmax}(\mathbf{x}_1 \mathbf{x}_2^\top)\mathbf{x}_1)$ [103, 99, 113].

*Unique*: There have been relatively fewer methods that explicitly aim to capture uniqueness, since label information is often assumed to be shared across modalities (i.e., redundancy) [98, 94], two that we implement are:

1. LOWER: Some approaches explicitly include lower-order terms in higher-order interactions to capture unique information [65, 115].

2. REC: Finally, several methods include reconstruction objectives to encourage maximization of unique information (i.e., adding an objective $\mathcal{L}_{\text{rec}} = \|g_1(\mathbf{z}_{\text{mm}}) - \mathbf{x}_1\|_2 + \|g_2(\mathbf{z}_{\text{mm}}) - \mathbf{x}_2\|_2$ where $g_1, g_2$ are auxiliary decoders mapping $\mathbf{z}_{\text{mm}}$ to each raw input modality [92, 100, 111].

**Results**: We show the full results on estimating PID for all models trained on all datasets in Table 18. To better understand the models' ability in capturing each type of interaction, we present a concise overview of the full results in Table 4, focusing on the PID values of each type of interaction $I$ for each model on $I$-specialized dataset $\mathcal{D}_I$. We also compute the distribution of $I_{\mathcal{D}_I}$ over all models. We highlight the following observations:

*General observations*: We first observe that model PID measures are consistently higher than dataset measures. The sum of model measures is also a good indicator of test performance, which agrees with their formal definition since their sum is equal to $I(\{X_1, X_2\}; Y)$, the total explained mutual information between multimodal data and $Y$.

*On redundancy*: Overall, several methods succeed in capturing redundancy, with an overall average of $0.41 \pm 0.11$ and accuracy of $73.0 \pm 2.0\%$ on redundancy-specialized datasets. Additive, agreement, and alignment-based methods are particularly strong, but other methods based on tensor fusion (synergy-based), including lower-order interactions, and adding reconstruction objectives (unique-based) also capture redundancy well.

*On uniqueness*: Uniqueness is harder to capture than redundancy, with an average of $0.37 \pm 0.14$. Certain methods like additive and agreement do poorly on uniqueness, while those designed for uniqueness (lower-order interactions and reconstruction objectives) do well, with $U = 0.55$ and $73.0\%$ accuracy on uniqueness datasets.

*On synergy*: On average, synergy is the hardest to capture, with an average score of only $0.21 \pm 0.10$. The best-performing methods are tensor fusion $S = 0.33, \text{acc} = 74.0\%$ and multimodal transformer $S = 0.29, \text{acc} = 73.0\%$. Additive, agreement, and element-wise interactions do not seem to capture synergy well.

*On robustness*: Finally, we also show empirical connections between estimated PID values with model performance in the presence of noisy or missing modalities. Specifically, with the original performance $Acc$ on perfect bimodal data, we define the performance drop when $X_i$ is missing as $Acc_{-i} = Acc - Acc_i$ where $Acc_i$ is the unimodal performance when $X_i$ is missing. We find high correlation ($\rho = 0.8$) between $Acc_{-i}$ and the model's $U_i$ value. Inspecting the graph closely in Figure 5, we find that the correlation is not perfect because the implication only holds in one direction: high $U_i$ coincides with large performance drops, but low $U_i$ can also lead to performance drops. We find that the latter can be further explained by the presence of large $S$ values: when $X_i$ is missing, these interactions can no longer be discovered by the model which decreases robustness. For the subset of points when $U_i \leq 0.05$, the correlation between $S$ and performance drop is $\rho = 0.73$ while the correlation between $R$ and performance drop is only $\rho = 0.01$.

Table 11: Table of hyperparameters for prediction on synthetic data.

| Component | Model | Parameter | Value |
|---|---|---|---|
| Encoder | Identity | / | / |
| | Linear | Input size
Hidden dim | [200, 200]
512 |
| Decoder | Linear | Input dim
Hidden dim | [200, 200]
512 |
| Fusion | Concat | / | / |
| | ElemMultWithLinear
MI-Matrix [52] | Output dim | 512 |
| | LRTF [65] | Output dim
rank | 512
32 |
| | MULT [99] | Embed dim
Num heads | 512
8 |
| Classification head | Identity | / | / |
| | 2-Layer MLP | Hidden size
Activation
Dropout | 512
LeakyReLU(0.2)
0.1 |
| Training | EF & ADDITIVE & ELEM & TENSOR
MI & MULT & LOWER | Loss
Batch size
Num epochs
Optimizer/Learning rate | Cross Entropy
128
100
Adam/0.0001 |
| | AGREE & ALIGN | Loss

Batch size
Num epochs
Optimizer/Learning rate
Cross Entropy Weight
Agree/Align Weight | Cross Entropy
+ Agree/Align Weight
128
200
Adam/0.0001
2.0
1.0 |
| | REC [100] | Loss

Batch size
Num epochs
Optimizer
Learning rate
Recon Loss Modality Weight
Cross Entropy Weight

Intermediate Modules | Cross Entropy
+ Reconstruction (MSE)
128
100
Adam
0.0001
[1, 1]
2.0

MLP [512, 256, 256]
MLP [512, 256, 256] |

Table 12: Table of hyperparameters for prediction on affective computing datasets.

| Component | Model | Parameter | Value |
|---|---|---|---|
| Encoder | Identity | / | / |
| | GRU | Input size
Hidden dim | $[5, 20, 35, 74, 300, 704]$
$[32, 64, 128, 512, 1024]$ |
| Decoder | GRU | Input size
Hidden dim | $[5, 20, 35, 74, 300, 704]$
$[32, 64, 128, 512, 1024]$ |
| Fusion | Concat | / | / |
| | ElemMultWithLinear
MI-Matrix [52] | Output dim | $[400, 512]$ |
| | Tensor Fusion [115] | Output dim | 512 |
| | MULT [99] | Embed dim
Num heads | 40
8 |
| Classification head | Identity | / | / |
| | 2-Layer MLP | Hidden size
Activation
Dropout | 512
LeakyReLU(0.2)
0.1 |
| Training | EF & ADDITIVE & ELEM & TENSOR
MI & MULT & LOWER | Loss
Batch size
Num epochs
Optimizer/Learning rate | L1 Loss
32
40
Adam/0.0001 |
| | AGREE & ALIGN | Loss

Batch size
Num epochs
Optimizer/Learning rate
Agree/Align Weight | L1 Loss
+ Agree/Align Weight
32
30
Adam/0.0001
0.1 |
| | REC [100] | Loss

Batch size
Num epochs
Optimizer
Learning rate
Recon Loss Modality Weight

Intermediate Modules | L1 Loss
+ Reconstruction (MSE)
128
50
Adam
0.001
$[1, 1]$
MLP $[600, 300, 300]$
MLP $[600, 300, 300]$ |

Table 13: Table of hyperparameters for prediction on ENRICO dataset in the HCI domain.

| Model | Parameter | Value |
|---|---|---|
| Unimodal | Hidden dim | 16 |
| MI-Matrix [52] | Hidden dim
Input dims | 32
16, 16 |
| MI | Hidden dim
Input dims | 32
16, 16 |
| LRTF [65] | Hidden dim
Input dims
Rank | 32
16, 16
20 |
| Training | Loss
Batch size
Activation
Dropout
Optimizer
Learning Rate
Num epochs | Class-weighted Cross Entropy
32
ReLU
0.2
Adam
$10^{-5}$
30 |

Table 14: Table of hyperparameters for prediction on MIMIC dataset in the healthcare domain.

| Component | Model | Parameter | Value |
|---|---|---|---|
| Static Encoder | 2-layer MLP | Hidden sizes | $[10, 10]$ |
| | | Activation | LeakyReLU(0.2) |
| Static Decoder | 2-layer MLP | Layer sizes | $[200, 40, 5]$ |
| | | Activation | LeakyReLU(0.2) |
| Time Series Encoder | GRU | Hidden dim | 30 |
| Time Series Decoder | GRU | Hidden dim | 30 |
| Classification Head | 2-Layer MLP | Hidden size | 40 |
| | | Activation | LeakyReLU(0.2) |
| Fusion | LRTF [65] | Output dim | 100 |
| | | Ranks | 40 |
| | MI-Matrix [52] | output dim | 100 |
| Training | TENSOR, MI | Loss | Cross Entropy |
| | | Batch size | 40 |
| | | Num epochs | 20 |
| | | Optimizer | RMSprop |
| | | Learning rate | 0.001 |
| | REC [100] | Loss | Cross Entropy + Reconstruction(MSE) |
| | | Batch size | 40 |
| | | Num epochs | 30 |
| | | Optimizer | Adam |
| | | Learning Rate | 0.001 |
| | | Recon Loss Modality Weights | $[1, 1]$ |
| | | Cross Entropy Weight | 2.0 |
| | | Intermediate Modules | MLPs $[200, 100, 100]$, $[200, 100, 100], [400, 100, 100]$ |

Table 15: Table of hyperparameters for prediction on MAPS.

| Component | Model | Parameter | Value |
|---|---|---|---|
| Encoder | Identity | / | / |
| | Linear | Input size
Hidden dim | [1000, 18, 137]
512 |
| Decoder | Linear | Input dim
Hidden dim | [1000, 18, 137]
512 |
| Fusion | Concat | / | / |
| | ElemMultWithLinear
MI-Matrix [52] | Output dim | 512 |
| | LRTF [65] | Output dim
rank | 512
32 |
| | MULT [99] | Embed dim
Num heads | 512
8 |
| Classification head | Identity | / | / |
| | 2-Layer MLP | Hidden size
Activation
Dropout | 512
LeakyReLU(0.2)
0.1 |
| Training | EF & ADDITIVE & ELEM & TENSOR MI & MULT & LOWER | Loss
Batch size
Num epochs
Optimizer/Learning rate | Cross Entropy
128
100
Adam/0.0001 |
| | AGREE & ALIGN | Loss

Batch size
Num epochs
Optimizer/Learning rate
Cross Entropy Weight
Agree/Align Weight | Cross Entropy
+ Agree/Align Weight
128
200
Adam/0.0001
2.0
1.0 |
| | REC [100] | Loss

Batch size
Num epochs
Optimizer
Learning rate
Recon Loss Modality Weight
Cross Entropy Weight
Intermediate Modules | Cross Entropy
+ Reconstruction (MSE)
128
100
Adam
0.0001
[1, 1]
2.0
MLP [512, 256, 256]
MLP [512, 256, 256] |

Table 16: Table of hyperparameters for prediction on Push dataset.

| Component | Model | Parameter | Value |
|---|---|---|---|
| Pos Encoder | Linear | Hidden sizes | [64, 64, 64 (residual)] |
| Sensors Encoder | Linear | Hidden sizes | [64, 64, 64 (residual)] |
| Image Encoder | CNN | Filter sizes
Num filters
Filter strides
Filter padding | [5, 3, 3, 3, 3]
[32, 32, 32, 16, 8]
1
[2, 1, 1, 1, 1] |
| Control Encoder | Linear | Hidden sizes | [64, 64, 64 (residual)] |
| Fusion | Unimodal LSTM | Hidden size
Num layers | 512
2 |
| | Late Fusion LSTM | Hidden size
Num layers | 256
1 |
| Classification Head | Linear | Hidden size | 64 |
| Training | | Loss
Batch size
Num epochs
Activation
Optimizer
Learning rate | Mean Squared Error
32
20
ReLU
Adam
$10^{-5}$ |

Table 17: Table of hyperparameters for prediction on AV-MNIST dataset.

| Component | Model | Parameter | Value |
|---|---|---|---|
| Image Encoder | LeNet-3 | Filter Sizes | $[5, 3, 3, 3]$ |
| | | Num Filters | $[6, 12, 24, 48]$ |
| | | Filter Strides / Filter Paddings | $[1, 1, 1, 1]$ /$[2, 1, 1, 1]$ |
| | | Max Pooling | $[2, 2, 2, 2]$ |
| Image Decoder | DeLeNet-3 | Filter Sizes | $[4, 4, 4, 8]$ |
| | | Num Filters | $[24, 12, 6, 3]$ |
| | | Filter Strides / Filter Paddings | $[2, 2, 2, 4]/[1, 1, 1, 1]$ |
| Audio Encoder | LeNet-5 | Filter Sizes | $[5, 3, 3, 3, 3, 3]$ |
| | | Num Filters | $[6, 12, 24, 48, 96, 192]$ |
| | | Filter Strides / Filter Paddings | $[1, 1, 1, 1, 1, 1]/[2, 1, 1, 1, 1, 1]$ |
| | | Max Pooling | $[2, 2, 2, 2, 2, 2]$ |
| Audio Decoder | DeLeNet-5 | Filter Sizes | $[4, 4, 4, 4, 4, 8]$ |
| | | Num Filters | $[96, 48, 24, 12, 6, 3]$ |
| | | Filter Strides / Filter Paddings | $[2, 2, 2, 2, 2, 4]/[1, 1, 1, 1, 1, 1]$ |
| Classification Head | 2-Layer MLP | Hidden size | 100 |
| | | Activation | LeakyReLU(0.2) |
| Training | Unimodal, LF | Loss | Cross Entropy |
| | | Batch size | 40 |
| | | Num epochs | 25 |
| | | Optimizer/Learning rate/weight decay | SGD/0.05/0.0001 |

Table 18: Results on estimating PID on model families trained on synthetic datasets with controlled interactions.

| Category | GENERAL | | | | | REDUNDANCY | | | | | | | | | |
|---|---|---|---|---|---|---|---|---|---|---|---|---|---|---|---|
| Model | EF | | | | | ADDITIVE | | | | | AGREE | | | | |
| Measure | $R$ | $U_1$ | $U_2$ | $S$ | Acc | $R$ | $U_1$ | $U_2$ | $S$ | Acc | $R$ | $U_1$ | $U_2$ | $S$ | Acc |
| $(z_c)$ | 0.47 | 0.01 | 0.01 | 0.04 | 0.74 | 0.35 | 0.2 | 0 | 0.04 | 0.71 | 0.48 | 0.01 | 0.01 | 0.06 | 0.74 |
| $(z_1)$ | 0 | 0.41 | 0 | 0.04 | 0.73 | 0.02 | 0.14 | 0 | 0.03 | 0.61 | 0.02 | 0.15 | 0 | 0.04 | 0.38 |
| $(z_2)$ | 0.01 | 0 | 0.47 | 0.04 | 0.72 | 0.05 | 0 | 0.43 | 0.06 | 0.72 | 0.01 | 0 | 0.48 | 0.04 | 0.72 |
| $(z_1, z_2)$ | 0.15 | 0.02 | 0 | 0.29 | 0.72 | 0.15 | 0 | 0.2 | 0.13 | 0.56 | 0.14 | 0 | 0.26 | 0.09 | 0.66 |
| $(z_1^*, z_2^*, z_c^*)$ | 0.22 | 0 | 0.07 | 0.15 | 0.64 | 0.31 | 0.16 | 0 | 0.08 | 0.64 | 0.21 | 0 | 0.11 | 0.15 | 0.58 |
| $(z_1, z_2^*, z_c^*)$ | 0.16 | 0.15 | 0 | 0.15 | 0.71 | 0.06 | 0 | 0.18 | 0.06 | 0.6 | 0.06 | 0 | 0.08 | 0.08 | 0.53 |
| $(z_1, z_2, z_c^*)$ | 0.16 | 0.05 | 0 | 0.25 | 0.74 | 0.09 | 0 | 0.23 | 0.11 | 0.52 | 0.02 | 0 | 0.33 | 0.06 | 0.62 |
| $(z_1^*, z_2^*, z_c)$ | 0.25 | 0.03 | 0 | 0.14 | 0.66 | 0.39 | 0.11 | 0 | 0.08 | 0.67 | 0.44 | 0.04 | 0 | 0.09 | 0.67 |
| $(z_2^*, z_c^*)$ | 0.19 | 0 | 0.12 | 0.06 | 0.65 | 0.15 | 0 | 0.07 | 0.18 | 0.61 | 0.13 | 0 | 0.24 | 0.04 | 0.64 |
| $(z_2^*, z_c)$ | 0.26 | 0 | 0.12 | 0.05 | 0.7 | 0.27 | 0 | 0.18 | 0.08 | 0.71 | 0.1 | 0 | 0.1 | 0.16 | 0.61 |

| Category | REDUNDANCY | | | | | SYNERGY | | | | | | | | | |
|---|---|---|---|---|---|---|---|---|---|---|---|---|---|---|---|
| Model | ALIGN | | | | | ELEM | | | | | TENSOR | | | | |
| Measure | $R$ | $U_1$ | $U_2$ | $S$ | Acc | $R$ | $U_1$ | $U_2$ | $S$ | Acc | $R$ | $U_1$ | $U_2$ | $S$ | Acc |
| $(z_c)$ | 0.44 | 0.02 | 0 | 0.05 | 0.73 | 0.27 | 0 | 0.01 | 0.07 | 0.7 | 0.47 | 0.01 | 0.01 | 0.04 | 0.74 |
| $(z_1)$ | 0.03 | 0 | 0 | 0.05 | 0.48 | 0 | 0.2 | 0 | 0.05 | 0.66 | 0 | 0.56 | 0 | 0.02 | 0.74 |
| $(z_2)$ | 0.05 | 0 | 0.39 | 0.1 | 0.71 | 0.01 | 0 | 0.2 | 0.05 | 0.66 | 0.01 | 0 | 0.54 | 0.02 | 0.73 |
| $(z_1, z_2)$ | 0.02 | 0 | 0.38 | 0.08 | 0.63 | 0.07 | 0.02 | 0 | 0.14 | 0.66 | 0.15 | 0.03 | 0 | 0.31 | 0.73 |
| $(z_1^*, z_2^*, z_c^*)$ | 0.33 | 0.06 | 0 | 0.15 | 0.66 | 0.05 | 0 | 0.01 | 0.07 | 0.56 | 0.2 | 0.01 | 0.01 | 0.21 | 0.65 |
| $(z_1, z_2^*, z_c^*)$ | 0.21 | 0 | 0.05 | 0.14 | 0.66 | 0.04 | 0.05 | 0 | 0.08 | 0.61 | 0.09 | 0.24 | 0 | 0.13 | 0.71 |
| $(z_1, z_2, z_c^*)$ | 0.14 | 0 | 0.26 | 0.04 | 0.64 | 0.08 | 0.02 | 0 | 0.12 | 0.66 | 0.17 | 0.06 | 0 | 0.27 | 0.72 |
| $(z_1^*, z_2^*, z_c)$ | 0.23 | 0 | 0.09 | 0.08 | 0.64 | 0.1 | 0 | 0.01 | 0.06 | 0.6 | 0.27 | 0.05 | 0.02 | 0.15 | 0.68 |
| $(z_2^*, z_c^*)$ | 0.18 | 0.1 | 0 | 0.17 | 0.6 | 0.04 | 0 | 0.03 | 0.05 | 0.57 | 0.05 | 0 | 0.01 | 0.06 | 0.56 |
| $(z_2^*, z_c)$ | 0.16 | 0 | 0.2 | 0.06 | 0.69 | 0.15 | 0 | 0.03 | 0.06 | 0.65 | 0.14 | 0 | 0.03 | 0.05 | 0.65 |

| Category | SYNERGY | | | | | | | | | |
|---|---|---|---|---|---|---|---|---|---|---|
| Model | MI | | | | | MULT | | | | |
| Measure | $R$ | $U_1$ | $U_2$ | $S$ | Acc | $R$ | $U_1$ | $U_2$ | $S$ | Acc |
| $(z_c)$ | 0.2 | 0 | 0.01 | 0.05 | 0.67 | 0.4 | 0.02 | 0 | 0.06 | 0.73 |
| $(z_1)$ | 0.01 | 0.13 | 0 | 0.05 | 0.66 | 0 | 0.48 | 0 | 0.02 | 0.73 |
| $(z_2)$ | 0.01 | 0 | 0.23 | 0.05 | 0.66 | 0.02 | 0 | 0.42 | 0.06 | 0.71 |
| $(z_1, z_2)$ | 0.05 | 0.04 | 0 | 0.12 | 0.65 | 0.16 | 0.05 | 0 | 0.29 | 0.72 |
| $(z_1^*, z_2^*, z_c^*)$ | 0.07 | 0.01 | 0.01 | 0.08 | 0.55 | 0.21 | 0.11 | 0 | 0.2 | 0.66 |
| $(z_1, z_2^*, z_c^*)$ | 0.02 | 0.01 | 0 | 0.06 | 0.58 | 0.1 | 0.29 | 0 | 0.11 | 0.7 |
| $(z_1, z_2, z_c^*)$ | 0.04 | 0.01 | 0 | 0.08 | 0.62 | 0.18 | 0.02 | 0 | 0.3 | 0.72 |
| $(z_1^*, z_2^*, z_c)$ | 0.12 | 0.01 | 0.01 | 0.06 | 0.62 | 0.33 | 0 | 0.03 | 0.14 | 0.68 |
| $(z_2^*, z_c^*)$ | 0.02 | 0 | 0.01 | 0.06 | 0.54 | 0.13 | 0 | 0.39 | 0.04 | 0.67 |
| $(z_2^*, z_c)$ | 0.13 | 0 | 0.01 | 0.06 | 0.64 | 0.36 | 0 | 0.18 | 0.04 | 0.72 |

| Category | UNIQUE | | | | | | | | | |
|---|---|---|---|---|---|---|---|---|---|---|
| Model | LOWER | | | | | REC | | | | |
| Measure | $R$ | $U_1$ | $U_2$ | $S$ | Acc | $R$ | $U_1$ | $U_2$ | $S$ | Acc |
| $(z_c)$ | 0.53 | 0 | 0.01 | 0.03 | 0.75 | 0.55 | 0.02 | 0.01 | 0.02 | 0.75 |
| $(z_1)$ | 0 | 0.56 | 0 | 0.02 | 0.74 | 0 | 0.53 | 0 | 0.03 | 0.74 |
| $(z_2)$ | 0.01 | 0 | 0.54 | 0.02 | 0.72 | 0.01 | 0 | 0.55 | 0.02 | 0.73 |
| $(z_1, z_2)$ | 0.15 | 0.03 | 0 | 0.32 | 0.74 | 0.14 | 0.06 | 0 | 0.34 | 0.74 |
| $(z_1^*, z_2^*, z_c^*)$ | 0.21 | 0.01 | 0.01 | 0.2 | 0.65 | 0.19 | 0.03 | 0 | 0.26 | 0.66 |
| $(z_1, z_2^*, z_c^*)$ | 0.09 | 0.27 | 0 | 0.13 | 0.71 | 0.08 | 0.29 | 0 | 0.16 | 0.71 |
| $(z_1, z_2, z_c^*)$ | 0.16 | 0.06 | 0 | 0.27 | 0.73 | 0.21 | 0.05 | 0 | 0.31 | 0.72 |
| $(z_1^*, z_2^*, z_c)$ | 0.31 | 0.01 | 0.01 | 0.16 | 0.68 | 0.32 | 0 | 0.06 | 0.21 | 0.68 |
| $(z_2^*, z_c^*)$ | 0.13 | 0 | 0.29 | 0.04 | 0.64 | 0.19 | 0 | 0.36 | 0.04 | 0.66 |
| $(z_2^*, z_c)$ | 0.31 | 0 | 0.13 | 0.04 | 0.7 | 0.42 | 0 | 0.15 | 0.03 | 0.72 |

Table 19: Results on estimating PID on model families trained on real-world multimodal datasets.

| Category | GENERAL | | | | | REDUNDANCY | | | | | | | | | |
|---|---|---|---|---|---|---|---|---|---|---|---|---|---|---|---|
| Model | EF | | | | | ADDITIVE | | | | | AGREE | | | | |
| Measure | $R$ | $U_1$ | $U_2$ | $S$ | Acc | $R$ | $U_1$ | $U_2$ | $S$ | Acc | $R$ | $U_1$ | $U_2$ | $S$ | Acc |
| UR-FUNNY A,T | 0.02 | 0 | 0.02 | 0.06 | 0.59 | 0.02 | 0 | 0.02 | 0.15 | 0.60 | 0.02 | 0 | 0.02 | 0.05 | 0.59 |
| UR-FUNNY V,T | 0.05 | 0.01 | 0 | 0.17 | 0.62 | 0.11 | 0.08 | 0 | 0.17 | 0.60 | 0.02 | 0.04 | 0 | 0.24 | 0.59 |
| UR-FUNNY V,A | 0.01 | 0.27 | 0 | 0.19 | 0.59 | 0.03 | 0 | 0.02 | 0.15 | 0.56 | 0.02 | 0 | 0.02 | 0.15 | 0.53 |
| MOSEI A,T | 0.16 | 0.27 | 0.03 | 0 | 0.79 | 0.15 | 0.19 | 0.02 | 0.13 | 0.80 | 0.14 | 0.11 | 0.05 | 0.06 | 0.80 |
| MOSEI V,T | 0.01 | 0 | 0 | 0 | 0.63 | 0.11 | 1.21 | 0 | 0 | 0.80 | 0.1 | 0.84 | 0 | 0 | 0.80 |
| MOSEI V,A | 0 | 0 | 0 | 0.01 | 0.63 | 0.59 | 0.62 | 0 | 0.12 | 0.65 | 0.10 | 1.46 | 0 | 0.24 | 0.61 |
| MUSTARD A,T | 0.06 | 0 | 0.03 | 0.04 | 0.42 | 0.17 | 0 | 0.04 | 0.33 | 0.70 | 0.18 | 0 | 0.05 | 0.31 | 0.68 |
| MUSTARD V,T | 0 | 0 | 0 | 0 | 0.57 | 0.14 | 0 | 0.07 | 0.33 | 0.69 | 0.19 | 0 | 0.07 | 0.27 | 0.67 |
| MUSTARD V,A | 0 | 0 | 0 | 0 | 0.57 | 0.25 | 0.12 | 0 | 0.15 | 0.61 | 0.33 | 0.10 | 0 | 0.16 | 0.58 |
| MIMIC | 0.01 | 0.27 | 0 | 0 | 0.91 | 0.01 | 0.27 | 0 | 0 | 0.92 | 0.01 | 0.27 | 0 | 0 | 0.92 |
| ENRICO | 0.71 | 0.34 | 0.44 | 0.38 | 0.50 | 0.69 | 0.28 | 0.39 | 0.35 | 0.30 | 0.40 | 0.30 | 0.52 | 0.75 | 0.51 |

| Category | REDUNDANCY | | | | | SYNERGY | | | | | | | | | |
|---|---|---|---|---|---|---|---|---|---|---|---|---|---|---|---|
| Model | ALIGN | | | | | ELEM | | | | | TENSOR | | | | |
| Measure | $R$ | $U_1$ | $U_2$ | $S$ | Acc | $R$ | $U_1$ | $U_2$ | $S$ | Acc | $R$ | $U_1$ | $U_2$ | $S$ | Acc |
| UR-FUNNY A,T | 0.02 | 0 | 0.02 | 0.06 | 0.58 | 0.01 | 0 | 0.03 | 0.05 | 0.64 | 0.01 | 0 | 0.02 | 0.05 | 0.62 |
| UR-FUNNY V,T | 0.02 | 0 | 0.02 | 0.15 | 0.60 | 0.02 | 0.01 | 0 | 0.16 | 0.61 | 0.03 | 0.01 | 0 | 0.14 | 0.62 |
| UR-FUNNY V,A | 0.12 | 0.05 | 0.01 | 0.19 | 0.53 | 0.03 | 0.13 | 0 | 0.16 | 0.59 | 0.02 | 0.15 | 0 | 0.15 | 0.60 |
| MOSEI A,T | 0.12 | 0.26 | 0.14 | 0.16 | 0.65 | 0.09 | 1.07 | 0 | 0 | 0.80 | 0.18 | 0.17 | 0.06 | 0 | 0.80 |
| MOSEI V,T | 0.21 | 0.61 | 0 | 0.09 | 0.65 | 0.25 | 0.92 | 0 | 0.15 | 0.81 | 0.12 | 1.23 | 0 | 0 | 0.81 |
| MOSEI V,A | 1.15 | 0 | 0 | 0.79 | 0.40 | 0.14 | 1.21 | 0 | 0 | 0.65 | 0.12 | 1.07 | 0 | 0 | 0.65 |
| MUSTARD A,T | 0.20 | 0 | 0.06 | 0.29 | 0.70 | 0.17 | 0.01 | 0.11 | 0.23 | 0.60 | 0.20 | 0 | 0.13 | 0.25 | 0.59 |
| MUSTARD V,T | 0.16 | 0 | 0.08 | 0.29 | 0.70 | 0.18 | 0.01 | 0.04 | 0.32 | 0.64 | 0.18 | 0.01 | 0.02 | 0.34 | 0.59 |
| MUSTARD V,A | 0.27 | 0.12 | 0 | 0.14 | 0.62 | 0.36 | 0.07 | 0 | 0.16 | 0.55 | 0.33 | 0.05 | 0 | 0.18 | 0.60 |
| MIMIC | 0.01 | 0.28 | 0 | 0 | 0.91 | 0.04 | 0.24 | 0 | 0.01 | 0.91 | 0 | 0.28 | 0 | 0.01 | 0.91 |
| ENRICO | 0.37 | 0.34 | 0.57 | 0.76 | 0.52 | 0.3 | 0.43 | 0.29 | 0.73 | 0.44 | 0.38 | 0.48 | 0.32 | 0.69 | 0.50 |

| Category | SYNERGY | | | | | UNIQUE | | | | |
|---|---|---|---|---|---|---|---|---|---|---|
| Model | MI | | | | | LOWER | | | | |
| Measure | $R$ | $U_1$ | $U_2$ | $S$ | Acc | $R$ | $U_1$ | $U_2$ | $S$ | Acc |
| UR-FUNNY A,T | 0.01 | 0 | 0.03 | 0.04 | 0.62 | 0.02 | 0 | 0.01 | 0.14 | 0.60 |
| UR-FUNNY V,T | 0.04 | 0.02 | 0 | 0.14 | 0.64 | 0.04 | 0 | 0.01 | 0.15 | 0.62 |
| UR-FUNNY V,A | 0.02 | 0.23 | 0 | 0.15 | 0.61 | 0.02 | 0.15 | 0 | 0.19 | 0.58 |
| MOSEI A,T | 0.13 | 0.05 | 0.01 | 0.19 | 0.81 | 0.20 | 0.92 | 0 | 0.54 | 0.79 |
| MOSEI V,T | 0.11 | 0.98 | 0 | 0 | 0.80 | 0.17 | 1.14 | 0 | 0.08 | 0.80 |
| MOSEI V,A | 0.12 | 1.0 | 0 | 0 | 0.65 | 1.54 | 0.64 | 0 | 0.13 | 0.65 |
| MUSTARD A,T | 0.18 | 0 | 0.08 | 0.29 | 0.63 | 0.21 | 0 | 0.05 | 0.26 | 0.59 |
| MUSTARD V,T | 0.20 | 0.01 | 0.03 | 0.25 | 0.63 | 0.17 | 0 | 0.06 | 0.28 | 0.58 |
| MUSTARD V,A | 0.28 | 0.04 | 0.02 | 0.18 | 0.56 | 0.24 | 0.08 | 0 | 0.24 | 0.59 |
| MIMIC | 0.02 | 0.26 | 0 | 0.01 | 0.92 | 0.01 | 0.28 | 0 | 0.01 | 0.91 |
| ENRICO | 0.28 | 0.41 | 0.34 | 0.48 | 0.38 | 0.52 | 0.29 | 0.69 | 0.52 | 0.56 |

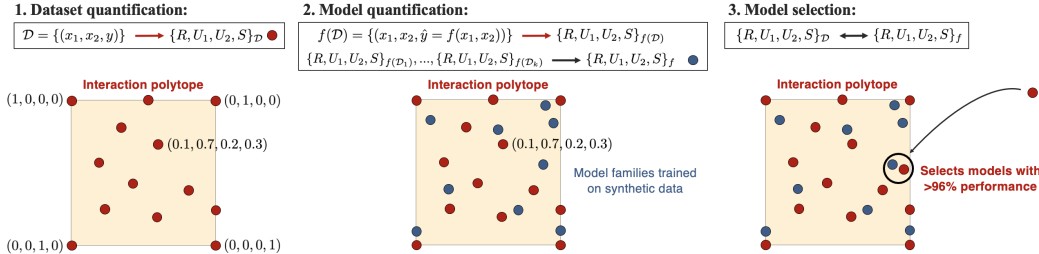

Figure 17: Model selection pipeline: (1) quantifying the interactions in different datasets, (2) quantifying interactions captured by different model families, then (3) selecting the closest model for a given new dataset yields models of $> 96\%$ performance as compared to exhaustively trying all models.

### C.8 Multimodal Model Selection

**Setup**: We show that our findings on quantifying multimodal datasets and model predictions are generalizable, and that the PID estimations are informative for model selection on new datasets *without training all models from scratch*. We simulate the model selection process on $5$ new synthetic datasets and $6$ real-world multimodal datasets. Given a new dataset $\mathcal{D}$, we compute their PID values $I_{\mathcal{D}}$ and normalize it to obtain $\hat{I}_{\mathcal{D}}$:

$$\hat{I}_{\mathcal{D}} = \frac{I_{\mathcal{D}}}{\sum_{I' \in \{R, U_1, U_2, S\}} I'_{\mathcal{D}}}. \tag{35}$$

We select the most similar dataset $\mathcal{D}^*$ from our base datasets (the 10 synthetic datasets presented in Table 2) that has the smallest difference measure

$$s(\mathcal{D}, \mathcal{D}') = \sum_{I \in \{R, U_1, U_2, S\}} |\hat{I}_{\mathcal{D}} - \hat{I}_{\mathcal{D}'}| \tag{36}$$

which adds up the absolute difference between normalized PID values on $\mathcal{D}$ and $\mathcal{D}'$. The model selection pipeline is illustrated in Figure 17. We hypothesize that the top-3 best-performing models on $\mathcal{D}^*$ should also achieve decent performance on $\mathcal{D}$ due to their similar distribution of interactions. We verify the quality of the model selection by computing a percentage of the performance of the selected model $f$ with respect to the performance of the actual best-performing model $f^*$ on $\mathcal{D}$, i.e. $\%\ \mathrm{Performance}(f, f,^*) = Acc(f)/Acc(f^*)$.

**Results**: We summarize our evaluation of the model selection in Table 5, and find that the top 3 chosen models all achieve $95\% - 100\%$ of the best-performing model accuracy, and above $98.5\%$ for all datasets except MUSTARD. For example, UR-FUNNY and MUSTARD have the highest synergy ($S = 0.13$, $S = 0.3$) and indeed transformers and higher-order interactions are helpful (MULT: $0.65\%$, MI: $0.61\%$, TENSOR: $0.6\%$). ENRICO has the highest $R = 0.73$ and $U_2 = 0.53$, and indeed methods for redundant and unique interactions perform best (LOWER: $0.52\%$, ALIGN: $0.52\%$, AGREE: $0.51\%$). MIMIC has the highest $U_1 = 0.17$, and indeed unimodal models are mostly sufficient [60]. We hypothesize that model selection is more difficult on MUSTARD since it has the highest synergy ratio, and it is still an open question regarding how to capture synergistic multimodal interactions. Therefore, PID values provide a strong signal for multimodal model selection.

### C.9 Real-world Case Study 1: Computational Pathology

**Setup**: The current standard of care for cancer prognosis in anatomic pathology is the integration of both diagnostic histological information (via whole-slide imaging (WSI)) as well as molecular information for patient stratifcation and therapeutic decision-making [63]. For cancer types such as diffuse gliomas, the combination of histological subtypes (astrocytoma, oligodendroglioma) and molecular markers (*IDH1* mutation, 1p19q codeletion) determines whether patients should be administered tumor resection, radiation therapy, adjuvant chemotherapy, or combination therapies [66, 16, 20]. Recent applications of multimodal learning in pathology have found that though fusion confers benefit for a majority of cancer types, unimodal models can be as competitive in patient stratification performance, which suggest potential clinical application of prognosticating using only a single modality for certain cancers [22].

Table 20: Results on estimating PID across the TCGA-GBMLGG and TCGA-PAAD datasets. $U_1, U_2$ correspond to pathology and genomics respectively.

| Dataset | TCGA-LGG | | | | TCGA-PAAD | | | |
|---------|----------|-----|-----|-----|-----------|-----|-----|-----|
| Measure | $R$ | $U_1$ | $U_2$ | $S$ | $R$ | $U_1$ | $U_2$ | $S$ |
| CVX | 0 | 0.02 | **0.06** | 0.02 | 0 | 0.06 | **0.08** | **0.15** |

Using The Cancer Genome Atlas (TCGA), PID was used to quantify pathology image-omic interactions in downstream prognostication tasks for two cancer datasets: lower-grade glioma (TCGA-LGG [15] ($n = 479$) and pancreatic adenocarcinoma (TCGA-PAAD [83], ($n = 209$). As mentioned in the main text, the modalities include: (1) a sequence of pre-extracted histology image features from diagnostic WSIs ($N \times 1024$, where $N$ is the number of non-overlapping $256 \times 256$ patches at $20\times$ magnification and 1024 is the extracted feature dimension of a pretrained vision encoder), and (2) feature vector ($1 \times D$) of $D$ bulk gene mutation status, copy number variation, and RNA-Seq abundance values. The label space is overall survival time converted to 4 discrete bins. Following the study design of [22], we trained an Attention-Based Multiple Instance Learning (ABMIL) model and a Self-Normalizing Network (SNN) for the pathology and genomic modalities using a log likelihood loss for survival analysis respectively [48, 55]. For ABMIL, a truncated ResNet-50 encoder pretrained on ImageNet was used for patch feature extraction of WSIs [45]. Unimodal features were extracted before the classification layer, with K-means clustering ($k = 3$) for each modality used to obtain $(x_1, x_2, y)$ pairs where $x_1$ is pathology, $x_2$ is genomics and $y$ is the discrete bins, which can then be used in CVX to compute PID measures.

**Results**: We assess these PID measures (Table 20) in the context of current scientific understanding of pathology and genomic interactions in cancer prognostication in the literature. Specifically, we compare against the findings [22], which previously investigated unimodal and multimodal survival models in TCGA and observed that risk predictions were primarily driven by genomics across most cancer types (such as TCGA-LGG), with exceptions in a few cancer types such as TCGA-PAAD. In the evaluation of TCGA-LGG and TCGA-PAAD in [22], concordance Index performance of unimodal-pathology, unimodal-genomic, and multimodal survival models were (0.668, 0.792, 0.808) and (0.580, 0.593, 0.653) respectively. In [22], one of the original hypotheses for the lack of multimodal improvement for some cancer types was due to high mutual information between both modalities. In our results, we observe the opposite effect in which there is little mutual information between both modalities. This observation coupled with the low and high synergy values in TCGA-LGG ($S = 0.02$) and TCGA-PAAD ($S = 0.15$) suggest that only TCGA-PAAD would benefit from multimodal integration, which was empirically found with both the c-Index improvement and statistical significant patient stratification results in TCGA-PAAD evaluation. The uniqueness values of both tasks also corroborate the high amount of task-specific information in genomics with high c-Index performance, and vice versa with pathology. Overall, we demonstrate we can use PID measures to refine biomedical insights into understanding when multimodal integration confers benefit.

### C.10    Real-world Case Study 2: Mood Prediction from Mobile Data

**Setup**: Suicide is the second leading cause of death among adolescents [18]. Despite these alarming statistics, there is little consensus concerning imminent risk for suicide [31, 56]. Intensive monitoring of behaviors via adolescents' frequent use of smartphones may shed new light on the early risk of suicidal thoughts and ideations [37, 72]. Smartphones provide a valuable and natural data source with rich behavioral markers spanning online communication, keystroke patterns, and application usage [59]. Learning these markers requires large datasets with diversity in participants, variety in features, and accuracy in annotations. As a step towards this goal, we partnered with several hospitals (approved by NIH IRB for central institution and secondary sites) to collect a dataset of mobile behaviors from high-risk adolescent populations with consent from participating groups. This data monitors adolescents spanning (a) recent suicide attempters (past 6 months) with current suicidal ideation, (b) suicide ideators with no past suicide attempts, and (c) psychiatric controls with no history of suicide ideation or attempts. Passive sensing data is collected from each participant's smartphone across a duration of 6 months. The modalities include (1) *text* entered by the user represented as a bag of top 1000 words that contains the daily number of occurrences of each word, (2) *keystroke* features that record the exact timing and duration that each character was typed on a mobile keyboard

Table 21: Results on estimating PID on the MAPS dataset with pairs of modalities text + apps ($\text{MAPS}_{T,A}$) and text + keystrokes ($\text{MAPS}_{T,K}$).

| Dataset | $\text{MAPS}_{T,A}$ | | | | $\text{MAPS}_{T,K}$ | | | |
|---|---|---|---|---|---|---|---|---|
| Measure | $R$ | $U_1$ | $U_2$ | $S$ | $R$ | $U_1$ | $U_2$ | $S$ |
| CVX | 0.09 | 0 | 0.09 | **0.26** | 0.12 | 0 | 0.04 | **0.40** |

Table 22: Results on estimating PID on model families trained on the MAPS dataset.

| Category | GENERAL | | | | | REDUNDANCY | | | | | | | | | |
|---|---|---|---|---|---|---|---|---|---|---|---|---|---|---|---|
| Model | EF | | | | | ADDITIVE | | | | | AGREE | | | | |
| Measure | $R$ | $U_1$ | $U_2$ | $S$ | Acc | $R$ | $U_1$ | $U_2$ | $S$ | Acc | $R$ | $U_1$ | $U_2$ | $S$ | Acc |
| $\text{MAPS}_{T,A}$ | 0.61 | 0.01 | **0.08** | 0.02 | 0.42 | 0.43 | **0** | 0.26 | 0 | 0.39 | 0.45 | **0** | 0.24 | 0 | 0.40 |
| $\text{MAPS}_{T,K}$ | 0.16 | 0.29 | 0 | 0.08 | 0.51 | 0.09 | **0** | 0.18 | 0.21 | 0.47 | 0.20 | **0** | 0.24 | 0.17 | 0.42 |

| Category | REDUNDANCY | | | | | SYNERGY | | | | | | | | | |
|---|---|---|---|---|---|---|---|---|---|---|---|---|---|---|---|
| Model | ALIGN | | | | | ELEM | | | | | TENSOR | | | | |
| Measure | $R$ | $U_1$ | $U_2$ | $S$ | Acc | $R$ | $U_1$ | $U_2$ | $S$ | Acc | $R$ | $U_1$ | $U_2$ | $S$ | Acc |
| $\text{MAPS}_{T,A}$ | 0.69 | **0** | 0.11 | 0.03 | 0.39 | 0.42 | 0.04 | 0.05 | 0.13 | 0.41 | 0.42 | 0.10 | 0.03 | 0.11 | 0.45 |
| $\text{MAPS}_{T,K}$ | 0.41 | 0.01 | 0.28 | 0.34 | 0.33 | 0.17 | 0.37 | **0.02** | 0.41 | 0.27 | 0.24 | 0.15 | 0 | 0.32 | **0.57** |

| Category | SYNERGY | | | | | | | | | |
|---|---|---|---|---|---|---|---|---|---|---|
| Model | MI | | | | | MULT | | | | |
| Measure | $R$ | $U_1$ | $U_2$ | $S$ | Acc | $R$ | $U_1$ | $U_2$ | $S$ | Acc |
| $\text{MAPS}_{T,A}$ | **0.28** | 0.12 | 0.01 | **0.21** | 0.47 | 0.40 | 0.09 | 0 | 0.03 | 0.48 |
| $\text{MAPS}_{T,K}$ | 0.25 | 0.24 | **0.02** | **0.40** | 0.36 | 0.10 | 0.27 | 0 | 0.19 | 0.50 |

| Category | UNIQUE | | | | | | | | | |
|---|---|---|---|---|---|---|---|---|---|---|
| Model | LOWER | | | | | REC | | | | |
| Measure | $R$ | $U_1$ | $U_2$ | $S$ | Acc | $R$ | $U_1$ | $U_2$ | $S$ | Acc |
| $\text{MAPS}_{T,A}$ | 0.53 | 0.04 | 0.04 | 0.09 | 0.41 | **0.28** | 0.06 | 0.07 | 0.10 | **0.56** |
| $\text{MAPS}_{T,K}$ | **0.12** | 0.24 | 0 | 0.22 | 0.51 | 0.23 | 0.17 | 0 | 0.18 | 0.52 |

(including alphanumeric characters, special characters, spaces, backspace, enter, and autocorrect), and (3) *mobile applications* used per day, creating a bag of 137 apps for each day that are used by at least 10% of the participants.

Every day at 8am, users are asked to respond to the following question - "In general, how have you been feeling over the last day?" - with an integer score between 0 and 100, where 0 means very negative and 100 means very positive. To construct our prediction task, we discretized these scores into the following three bins: *negative* $(0 - 33)$, *neutral* $(34 - 66)$, and *positive* $(67 - 100)$, which follow a class distribution of 12.43%, 43.63%, and 43.94% respectively. For our 3-way classification task, participants with fewer than 50 daily self-reports were removed since these participants do not provide enough data to train an effective model. In total, our dataset consists of 1641 samples, consisting of data coming from 17 unique participants.

**Results**: The results are shown in Table 21, 22. We observe that both $\text{MAPS}_{T,A}$ (text + apps) and $\text{MAPS}_{T,K}$ (text + keystrokes) have the highest synergy ($S = 0.26$, $S = 0.4$ respectively) and some redundancy ($R = 0.09$, $R = 0.12$) and uniqueness in the second modality ($U_2 = 0.09$, $U_2 = 0.04$). We find indeed the best-performing models are REC (acc $= 0.56\%$) and MULT (acc $= 0.48\%$) for $\text{MAPS}_{T,A}$, and TENSOR (acc $= 0.57\%$) and REC (acc $= 0.52\%$) for $\text{MAPS}_{T,K}$, which are designed to capture synergy and uniqueness interactions. Our model selection also successfully chooses REC for $\text{MAPS}_{T,A}$ and TENSOR for $\text{MAPS}_{T,K}$ with a high agreement of $\alpha = 0.66$ and $\alpha = 0.74$ (98.3% and 99.85% of the highest agreement respectively). This result further corroborates the utility and generalizability of our PID estimations and model selection.

Table 23: Results on estimating PID on robotic perception task.

| Task | PUSH I,C | | | |
|---|---|---|---|---|
| Measure | $R$ | $U_1$ | $U_2$ | $S$ |
| CVX | 0.24 | 0.03 | 0.06 | 0.04 |
| BATCH | 0.75 | 1.79 | 0.03 | 0.08 |

### C.11 Real-world Case Study 3: Robotic Perception

**Setup**: MuJoCo PUSH [57] is a contact-rich planar pushing task in the MuJoCo simulation framework, in which a 7-DoF Panda Franka robot is pushing a circular puck with its end-effector in simulation. We estimate the 2D position of the unknown object on a table surface, while the robot intermittently interacts with the object. The final dataset consists of 1000 trajectories with 250 steps at 10Hertz, of a simulated Franka Panda robot arm pushing a circular puck in MuJoCo [95]. The pushing actions are generated by a heuristic controller that tries to move the end-effector to the center of the object. The multimodal inputs are gray-scaled images ($1 \times 32 \times 32$) from an RGB camera, forces (and binary contact information) from a force/torque sensor, and the 3D position of the robot end-effector. The task is to predict the 2-D planar object pose.

We use CNN feature extractors and LSTM as a sequence model for both unimodal and multimodal models. We use hyperparameters from Table 16. Because both CVX and BATCH assumes discrete classes of labels, we discretize all continuous labels into 10 bins of equal data points from the training set, and use the same bin cutoffs at test time. The task is to predict the correct bin that contains the original label, and we use cross entropy loss as our loss function.

**Results**: The results are shown in Table 23. As can be seen from the table, BATCH predicts $U_1$ as the highest PID value ($U_1 = 1.79$), which aligns with the fact that image is the best unimodal predictor for this dataset [60]. Comparing both estimators, CVX underestimates $U_1$ and $R$ since CVX clusters the data before processing, and the high-dimensional time-series modalities cannot be easily described by clusters without losing information. In addition, both BATCH and CVX predict a low $U_2$ value but attributes some mutual information to $R$ (redundancy), implying that a multimodal model with both modalities would not be much better compared to an unimodal model on modality 1, since the multimodal model would mostly get redundant information from modality 2. In our experiments, we observe no difference in performance between the multimodal model and the unimodal model with modality 1.

## D Summary of Takeaway Messages and Future Work

From these results, we emphasize the main take-away messages and motivate several directions for future work:

1. Dataset quantification: PID provides reliable indicators for the nature and quantity of interactions present in both synthetic and real-world multimodal datasets, which provides a useful tool summarizing these datasets. In our opinion, creators of new datasets focused on modeling interactions between multiple features or modalities should report estimated PID values on new datasets alongside justification on whether these values are to be expected based on how the features were selected.

2. Model quantification: PID also provides reasonable indicators for the interactions captured by trained models, but naturally there is more noise due to different model parameter counts, training trajectories, and non-convex optimization artifacts that despite our best efforts remain impossible to exactly control. Despite these details, we still find several consistent patterns in the types of interactions different models capture. We suggest that researchers developing new multimodal models explicitly state the assumptions on the underlying multimodal distribution their models require and the interactions their models are designed to capture, before testing exhaustively on datasets predominantly of those interactions.

3. Model selection: Our experiments on model selection for real-world datasets and applications demonstrate potential utility in a rough 'user-guide' for practitioners aiming to tackle real-world multimodal datasets. Given a new dataset, estimate its PID values. If there is high $U_1$ and $U_2$, just using unimodal models in the corresponding modality may be sufficient.

Otherwise, if there is high $R$, methods like agreement, alignment, ensemble, or co-training should be tried. If there is high $S$, it is worth spending the time on multimodal methods that model more complex interactions based on tensors, multiplicative interactions, and self-attention.

We believe that the intersection of PID and machine learning opens the door to several exciting directions of future work:

1. Pointwise interaction measures: A natural extension is to design pointwise measures: how much does a single datapoint contribute to redundancy, uniqueness, or synergy in the context of an entire distribution? Pointwise measures could help for more fine-grained dataset and model quantification, including error analysis of incorrect predictions or active learning of difficult examples.

2. Representation learning for desired interactions: Can we design new model architectures or training objectives that better capture certain interactions? Current well-performing models like Transformers already include some notion of multiplicative interactions, so it does seem that, heuristically, synergy between input features is playing a role in their success. Designing models that better capture synergy, as quantified by our PID estimators, could be a path towards learning better representations.

3. Principled approaches to fairness and invariance: Currently, PID is designed to measure the information that 2 variables contribute towards a task $Y$, but conversely it can also be used to *remove* information that one variable can have about $Y$, in the context of another variable. These could provide formal learning objectives for fairness, privacy, and other feature invariance tasks.

# E    Limitations and Broader Impact

## E.1    Limitations of our Estimators

While our estimators appear to perform well in practice, each suffers from distinct limitations. For CVX, it is rarely able to scale up to domains with more than several thousand classes. Furthermore, if $|\mathcal{X}_i|$ is large—which is frequently the case in real-world applications, then empirical frequencies will often be near $0$, causing much instability in conic solvers (see Appendix B.1 for a more detailed description).

BATCH sidesteps many of the issues with CVX by applying batch gradient descent. However, it may suffer from approximation error depending on whether the network can sufficiently cover the space of joint distributions (i.e., representation capacity), alongside bias resulting from mini-batch gradient descent (as compared to full-batch gradient descent) when approximating $\tilde{q}$ using bootstrapped samples. Lastly, it may suffer from the usual problems with neural estimators, such as local minima, poor or unstable convergence (in both learning $\tilde{q}$ and $\hat{p}$). Therefore, while BATCH scales to high-dimensional continuous distributions, it comes with the challenges involving training and tuning neural networks.

## E.2    Broader Impact

Multimodal data and models are ubiquitous in a range of real-world applications. Our proposed framework based on PID is our attempt to systematically quantify the plethora of datasets and models currently in use. While these contributions will accelerate research towards multimodal datasets and models as well as their real-world deployment, we believe that special care must be taken in the following regard to ensure that these models are safely deployed:

**Care in interpreting PID values**: Just like with any approximate estimator, the returned PID values are only an approximation to the actual interactions and care should be taken to not overfit to these values. Other appropriate forms of dataset visualization and quantification should still be conducted to obtain holistic understanding of multimodal datasets.

**Privacy, security, and biases:** There may be privacy risks associated with making predictions from multimodal data if the datasets include recorded videos of humans or health indicators. In our experiments with real-world data where people are involved (i.e., healthcare and affective computing), the creators of these datasets have taken the appropriate steps to only access public data which participants/content creators have consented for released to the public. We also acknowledge the risk

of exposure bias due to imbalanced datasets that may cause biases towards certain genders, races, and demographic groups. Therefore, care should be taken in understanding the social risks of each multimodal dataset in parallel to understanding its interactions via PID.

**Time & space complexity**: Modern multimodal datasets and models, especially those pretrained on internet-scale data, may cause broader impacts resulting from the cost of hardware, electricity, and computation, as well as environmental impacts resulting from the carbon footprint required to fuel modern hardware. Future work should carefully investigate the role of size on the interactions learned by models through estimated PID values. Our preliminary experiments showed that smaller models could still capture high degrees of each interaction, which may pave away towards designing new inductive biases that enable interaction modeling while using fewer parameters.

Overall, PID offers opportunities to study the potential social and environmental issues in multimodal datasets by obtaining a deeper understanding of the underlying feature interactions, providing a path towards interpretable and lightweight models. We plan to continue expanding our understanding of PID via deeper engagement with domain experts and how they use this framework in their work. Our released datasets, models, and code will also present a step towards scalable quantification of feature interactions for future work.

