# OpenReview forum: "Quantifying & Modeling Multimodal Interactions: An Information Decomposition Framework"
_NeurIPS.cc/2023/Conference — NeurIPS 2023 poster_

### Official Review · Reviewer_uMC4 · 2023-06-30

**Soundness:** 3 good
**Presentation:** 4 excellent
**Contribution:** 3 good
**Rating:** 7
**Confidence:** 4

**Summary:**

This work is mainly concerned with the methodological contribution of estimators for partial information theory statistics, to quantify the levels of redundancy, uniqueness and synergy, which are expressed in information theoretic terms, through the use of mutual information.

Depending on dataset size and dimensionality, the authors propose two variants of estimators: one that is exact but can only be used for small, low-dimensional data, and one that is approximate, but that scales to large, high-dimensional data.
PID statistics can be computer for models as well, which enables the extension of to the proposed framework to deal with model selection.

A very large series of experiments are discussed to support 1) the usefulness of PID statistics, 2) the quality of the proposed estimators, 3) the applicability of the proposed methods to real-world dataset. In addition, several experiments on model selection are carried out as well. Finally, three real world applications are studied through the lenses of partial information decomposition, further substantiating the impact of such statistics. When possible, evaluation is exact, given the the ground truth PID statistics value could be computed analytically. Otherwise, the authors resort to human feedback.

**Strengths:**

* Deepening our understanding of multimodal data, and models built to address tasks that rely on multimodal data is of fundamental importance. The use of partial information decomposition is, in my opinion, a good approach to achieve such goals, and this paper is convincing in both showcasing that PID statistics are indeed representative for multimodal data and modelling tasks, as well as supporting the proposed methodology to compute such PID statistics.

* The experimental campaign is very extensive and overall (some comments in the questions section should be addressed though) very convincing and clear.

* The proposed methodology (especially the approximate one) for computing PID statistics is simple, and easy to implement


**Weaknesses:**

* Multimodal VAEs [1,2,3] are missing. To the best of my knowledge, a large body of research on multimodal representation learning is missing from the study of PID statistics for model predictions. On the one hand, I understand that in this work a model is used to address a given downstream task, whereas multimodal VAEs are used mostly to learn latent representations of all modalities, for generative purposes, but also to solve downstream tasks. Since the literature on Multimodal VAEs clearly addresses relevant questions on coherency, and synergy, I think this work could greatly benefit from an extension to such models.

* Alternatives to compute MI are not considered as “competitor” methods for the approximate scheme. Approaches such as [4, 5] have received a lot of attention as means to compute MI. Would it make sense to consider those as building blocks for alternative approximate PID estimation methods?

* Some details in the experimental section need further clarification. For example, in section 4.3 on the robustness, e.g. missing modalities, it is not clear if this is considered at training or a test time.

[1] Wu, et al.,”Multimodal Generative Models for Scalable Weakly-Supervised Learning”, https://arxiv.org/abs/1802.05335
[2] Shi, et al., “Variational Mixture-of-Experts Autoencoders for Multi-Modal Deep Generative Models”, https://arxiv.org/abs/1911.03393
[3] Palumbo, et al., “MMVAE+: Enhancing the Generative Quality of Multimodal VAEs without Compromises”, https://openreview.net/forum?id=B42UJTVdDZ5
[4] Belghazi, et al., “MINE: Mutual Information Neural Estimation”, https://arxiv.org/abs/1801.04062
[5] Hijelm, et al., “Learning deep representations by mutual information estimation and maximization”, https://arxiv.org/abs/1808.06670


**Questions:**

* I understand that both the methodology and experiments tackle two modalities. Now, even if datasets with more than two modalities, and models capable of handling more than two modalities are not extremely common, a natural question that arises both for the proposed methodology, its foundations on PID, and the experiments is how well does the schemes proposed in this work scale with the number of modalities.

* In the appendix, it is mentioned that the BATCH estimator, in its current form, relies on pre-trained classifiers, including multimodal ones. As explained, this can have the unintended effect of perturbing the estimation of PID statistics, resulting in errors on synergy. While a theoretical analysis of approximation guarantees seem to be rather difficult, could you discuss about means to keep approximation errors under control?

* For the BATCH estimator, a key aspect to consider is the dimensionality of the hidden representation that is used to compute matrix A (before sinkhorn is applied). Do you have any insights concerning the effects of this hyper-parameter?

* On the values obtained for R, U and S. As an overview on the various result tables, it is somehow hard to justify the choice for bold numbers. In particular, if we consider results on synthetic data for which a ground truth exists, we can see that the values for PID statistics obtained by either the exact or approximate schemes is very small. In some cases the differences between the values obtained is small, and bold numbers seem to be chosen to “comply” with group truth: see for example Table 2 $y=f(z_2^*, z_c^*)$ for the CVX scheme. More generally, the question I would like to ask is related to the “degree” aspects of PID statistics, and how to interpret correctly numerical values, especially when such values are very low, say what is the difference between R=0.07 and R=0.06 in practical terms? I understand that the first statistics indicate more redundancy than the second, but is this going to be discernible in practice?

* Finally, if time allows, would it be possible to address the points mentioned in the "weaknesses" part of the review?

**Limitations:**

To the best of my knowledge, there is no explicit discussion on limitations nor societal impact. However, in the appendix, it is possible to find information about caveats, assumptions, which help understanding some limitations of the proposed methods.

---

> ### Author Rebuttal · Authors · 2023-08-09
>
> Thank you for your valuable feedback and insightful comments! We respond to some concerns below:
>
> [multimodal VAEs] While multimodal VAEs are related, their goal is not to define and estimate multimodal interactions, but rather using intuition of interactions to guide model design (e.g., see figure 2 in [2] for a summary). Furthermore, training these multimodal VAEs is challenging (even harder than supervised learning). We have added these references and a discussion to our paper.
>
> [MI estimation] We do use these as an inspiration for our second batch estimator - the fact that neural networks can learn representations that are optimized for a MI term (e.g., like [4,5]). The main difference is that we’re not just estimating MI for a given distribution p, but also optimizing over distributions q \in \Delta_p wrt an MI objective. We are extending our work to optimize for a certain multimodal interaction (not just estimating it), which mirrors your suggested work [5].
>
> [other methods] **Our framework allows for easy generalization to other definitions: we implemented the original called I-min, WMS, and CI (see [1] for a review), as well as 3 non-info theory measures Shapley values, Integrated gradients (IG), and CCA**. These results are in Table 1 of rebuttal pdf, and to summarize:
>
> 1. I-min can sometimes overestimate synergy and uniqueness. It overestimates synergy on U1-only and U2-only datasets, and assigns non-zero unique interactions (U1=0.08, U2=0.07 and U1=0.03, U2=0.04) even though there is no uniqueness.
> 2. WMS is actually synergy minus redundancy, so when they are of equal magnitude WMS cancels out to be 0, leading one to erroneously conclude there is no S or R. For example, it is unable to detect high synergy for S-only data and assigns uniqueness instead. This term can also be negative (S=-0.11 for R-only data).
> 3. CI can also be negative (U1=-0.09, U2=-0.10 for R-only data), and sometimes incorrectly concludes highest uniqueness (U1=0.13, U2=0.14) for S-only data.
> 4. Shapley and IG are based on interactions captured by a multimodal model - our work is fundamentally different in that interactions are properties of data. This allows us to characterize datasets and select the most appropriate model.
> 5. CCA only gives shared information between modalities but does not give other interactions. We ran this and found that one can learn representations to maximize correlation arbitrarily high (e.g., rho=1.0 for MOSEI), even though the redundancy is bounded.
>
> [1] Griffith and Koch. Quantifying synergistic mutual information. 2014
>
> [robustness] This is only test-time performance - models trained on all modalities but tested on data with modality imperfections and drops, which we show is highly correlated with the amount of unique and synergistic information that the trained model learns.
>
> [>2 modalities] Extensions to more than 2 modalities is challenging due to a larger number of interactions (e.g., x1 and x2 may exhibit synergy, but not with x3), but future work can leverage advances in information theory: e.g. one could consider decomposing features hierarchically, or approximating via pairwise synergies.
>
> [batch estimator approximation] We notice that under-fitting classifiers could cause the BATCH estimator to underestimate synergy. We gradually scale up our model size and batch size until PID converges. Combined with traditional regularization techniques against overfitting (such as early stopping), these techniques can mitigate approximation errors.
>
> [batch estimator dimension] The hidden dimension (d) is important because too small d limits expressivity of joint distributions q, and too large d can be computationally expensive. We scale up the hidden dimension size until the PID statistics stabilize, and reach a final value d=32.
>
> Another important hyperparameter is the batchsize - a higher batchsize gives more accurate estimates of the joint distribution between X1 and X2, while a smaller batchsize is also important for regular updates via SGD. We tune the batchsize and reach n=256.
>
> We have included code in the supplementary and plan to release it publicly after the review period so other readers can play with these settings.
>
> [table 2] **The absolute value of PID estimates (i.e., actual bits of information) will never be fully 1 bit, because one can never extract the exact amount of task-relevant information**, which would imply 100% train and test accuracy. We can get a sense of the total capturable information using the bound from Feder and Merhav [1], which relates the accuracy of the best model on these datasets with the mutual information between the inputs to the label. **Since the test accuracies for Table 2 datasets range from 67-75%, this corresponds to total MI of 0.422-0.585 bits**. This is a tighter upper bound ground truth that our estimators should capture, and we see that they are quite close, see Table 1 in attached rebuttal pdf.
>
> [1] Feder and Merhav. Relations between entropy and error probability. IEEE Transactions on Information theory 1994
>
> [degree aspects] We are aware of challenges in evaluating PID estimation and emphasize that they should be interpreted as a relative sense of which interaction is most important within a dataset. One should not compare interactions across different datasets, since the total information is different so they lie on different scales. In working with end users, they also generally relied on the top interaction as well as the distribution of interactions (e.g., mostly redundancy, or whether all interactions are relatively uniform) to make modeling decisions.
>
> [limitations] We have expanded on the limitations and broader impact section in Appendix E, highlighting limitations and future directions in improving the estimators and their evaluation leveraging future progress in multimodal benchmarks, generative models, and MI estimation.

---

> > ### Comment · Reviewer_uMC4 · 2023-08-17
> > **Thanks for your detailed rebuttal**
> >
> > Dear Authors,
> > thanks for a detailed rebuttal, both for the general comments, and precise answers to the questions.
> >
> > I think this is a solid piece of work, and that it will be worth discussing it in the community. There are surely a number of issues to investigate more, but these issues are related to the general interpretability of MI, and PID statistics, as well as ways of designing experiments (including those with human feedback) that are strong and conclusive, rather than on the techniques, experiments and results presented in this paper.
> >
> > For these reasons, I will keep my score of 7: Accept, which I think this paper well deserves.

---

### Official Review · Reviewer_xXH1 · 2023-07-05

**Soundness:** 2 fair
**Presentation:** 3 good
**Contribution:** 3 good
**Rating:** 6
**Confidence:** 2

**Summary:**

This paper proposes an information-theoretic approach to quantify the degree of three PID(Partial Information Decomposition) statistics of a multimodal distribution: redundancy, uniqueness and synergy. To solve the difficulties in computing PID over high-dimensional continuous distributions, two scalable estimators for PID statistics are introduced, enabling the estimation of PID statistics over large multimodal datasets using small subsampled batches. To validate PID estimation, experiments are conducted on synthetic and real-world multimodal datasets. The usefulness of PID is also demonstrated in model selection and real-world case studies.

**Strengths:**

+ This paper proposes an useful approach to quantify the nature and degree of modality interactions based on information theory. The proposed PID estimators can scale to high-dimensional multimodal datasets and models.
+ The definitions of redundancy, uniqueness, and synergy are intuitive and could help to better understand multimodal datasets and models.
+ The proposed approach could generalize to real-world applications and provide reasonable interpretation for dataset properties and model selection.

**Weaknesses:**

- The results in Table 2 and 3 are unable to prove the accuracy of the PID estimators. Estimated results only show a rough trend. There is an order of magnitude difference between the approximation and the ground truth in most cases. Besides, it seems that the approximation becomes more inaccurate when the value distribution of R, U, S is more uniform (i.e. The second main row in Table 3). Is this method only applicable to simple and distinctive datasets?
- Lack of comparison and discussion with existing methods. For example, Shapley Value could measure the contribution of each modality in multimodal learning. What are the similarities and differences between them? Why is PID better than them?

**Questions:**

Major concerns have been mentioned in Weakness.

Line 163: missing ')'

**Limitations:**

The authors did not address the limitation.

---

> ### Author Rebuttal · Authors · 2023-08-09
>
> Thank you for your valuable feedback and insightful comments! We respond to some concerns below:
>
> [table 2] **The absolute value (i.e., actual bits of information) will never be fully 1 bit, because one can never extract the exact amount of task-relevant information**, which would imply 100% train and test accuracy. Noise in the data and labels, errors in feature learning and model optimization, and generalization gaps between train and test all contribute to lower estimated interaction values than the ground truth.
>
> We can get a sense of the total capturable information using the bound from Feder and Merhav [1], which relates the accuracy of the best model on these datasets with the mutual information between the inputs to the label. **Since the test accuracies for Table 2 datasets range from 67-75%, this corresponds to total MI of 0.422-0.585 bits**. This is a tighter upper bound ground truth that our estimators should capture, and we see that they are quite close, see Table 1 in attached rebuttal pdf.
>
> We are aware of challenges in evaluating PID estimation and emphasize that they should be interpreted as a relative sense of which interaction is most important. Both CVX and BATCH measures agree with each other on relative values and are consistent with ground truth interactions on relative values.
>
> **Our method is not limited to simple and distinctive datasets, and we also emphasize that evaluating PID estimation is just half the picture - the more important part is using these quantities for downstream real-world case studies in dataset quantification, model quantification, and model selection through engagement with domain experts. Our experiments cover more than 11 datasets and 10 complex models from research areas of multimedia, affective computing, robotics, healthcare, and HCI.**
>
> [1] Feder and Merhav. Relations between entropy and error probability. IEEE Transactions on Information theory 1994
>
> [table 3] For human judgment, we cannot ask humans to give a score in bits, so it is on a completely different scale (1-5 scale). To put them in the same scale, we normalize the human ratings such that the sum of human interactions = sum of PID estimates. The resulting comparisons are in Table 2 in attached rebuttal pdf, and we find very close match between estimators and human gold standards.
>
> [other methods] Our framework allows for easy generalization to other definitions as well: **we implemented 3 alternative definitions for interactions, including the original called I-min, WMS, and CI (see [1] for a review). These results are in Table 1 of rebuttal pdf:**
>
> 1. I-min can sometimes overestimate synergy and uniqueness. It overestimates synergy on U1-only and U2-only datasets, and assigns non-zero unique interactions (U1=0.08, U2=0.07 and U1=0.03, U2=0.04) even though there is no uniqueness.
> 2. WMS is actually synergy minus redundancy, so when they are of equal magnitude WMS cancels out to be 0, leading one to erroneously conclude there is no S or R. For example, it is unable to detect high synergy for S-only data and assigns uniqueness instead. This term can also be negative (S=-0.11 for R-only data).
> 3. CI can also be negative (U1=-0.09, U2=-0.10 for R-only data), and sometimes incorrectly concludes highest uniqueness (U1=0.13, U2=0.14) for S-only data.
>
> **Our framework also fits in with non-info theory measures, we implemented 3 more:**
>
> 1. Shapley is based on interactions captured by a multimodal model - our work is fundamentally different in that interactions are properties of data. This allows us to characterize datasets and select the most appropriate model. We ran this and found 3 issues:
> - It only captures overall independence, not separate uniqueness for each modality
> - It overestimates synergy=0.5 for pure redundancy data and overestimates independence=1.0 for pure synergy data
> - We also found difficulties in estimating Shapley for large, high-dimensional datasets efficiently, which is still an open question.
> 2. Integrated gradients (IG) only gives unimodal contributions of a trained model. We apply this method for computational pathology to derive importance of histology and genomics data, and find IG echoes the unique interactions of PID, but PID enables us to do so before model selection and training. Moreover, it is not clear how to estimate synergy with IG.
> 3. CCA only gives shared information between modalities, which can be used as a proxy of redundancy, but does not give other interactions. We ran this and found that one can learn representations to maximize correlation arbitrarily high (e.g., rho=1.0 for MOSEI), even though the redundancy is bounded.
>
> [1] Griffith and Koch. Quantifying synergistic mutual information. 2014
>
> [limitations] We have expanded on the limitations and broader impact section in Appendix E. To summarize:
> 1. CVX works for smaller domains up to several thousand classes or may suffer from low empirical frequencies beyond this; BATCH may suffer from approximation error depending on whether the network can sufficiently cover the space of joint distributions, bias resulting from mini-batch gradient descent, as well as the usual local minima and unstable convergence problems with neural estimators.
> 2. These estimators only approximate real interactions due to cluster preprocessing as well as the standard optimization and generalization errors of approximate models. We expect future progress in density estimators, generative models, and MI estimation to help address these problems.
> 3. We are aware of challenges in evaluating PID estimation and emphasize that they should be interpreted as a relative sense of which interaction is most important, and a guideline to inspire model selection and design.

---

> > ### Comment · Reviewer_xXH1 · 2023-08-21
> > **Thanks for the response**
> >
> > Many Thanks for the authors' response. For the raised concerns, most of them are addressed to a certain degree. Hence, I decide to raise my score to 6-Weak Accept. Meanwhile, I strongly encourage the authors further polish the submission, especially fully considering the discussion and limitation.

---

### Official Review · Reviewer_td1f · 2023-07-05

**Soundness:** 3 good
**Presentation:** 4 excellent
**Contribution:** 3 good
**Rating:** 6
**Confidence:** 3

**Summary:**

This paper addresses a fundamental question in multimodal modeling: given multiple input modalities, how can we quantify the ways they must interact (as part of a model) to solve a particular task. Furthermore, can we use knowledge to learn how to design models that better leverage these interactions.

The authors propose an information-theoretic approach to this problem by quantifying three properties (redundancy, uniqueness, and synergy) using mutual information across modalities. They solve for these properties using two estimators (one using convex optimization and the other sampling). The first estimator uses convex optimization and the second uses a sampling-baed approximation.

They evaluate their approaches for quantifying these properties using synthetic and real-world datasets. For the latter, annotators manually went through each dataset and quantified their intuition for overlap between modalities. They show that their models are able to do a reasonable job estimating these quantities.

**Strengths:**

* Overall this is a well written and carried out paper with a lot of depth. The authors approach the problem in a principled manner and have thought deeply about underlying concepts of multi-modality.
* While I did not go through the derivations, the authors introduce a nice framework for thinking about multi-modal interactions.
* I appreciate the combination of synthetic experiments, real-world experiments, snapshot "case studies," and various ablation experiments.
* The appendix includes a large amount of interesting additions.

**Weaknesses:**

* I found the results hard to interpret and -- given synthetic results -- am wary of the validity of the claims. For example, in the second set of synthetic experiments (Table 2) the recovered values are MUCH lower than ideal (e.g., D_R set has R(CVX)=0.16, R(Batch)=0.29, and R(Truth)=1). This is especially true in the "mixed interaction" synthetic datasets. Is there a good reason that the estimated values are so far from the truth? Are the proposed estimators good in the absence of noise (i.e., the bit-wise experiment) but do they quickly fail once there is uncertainty? Is the issue with the estimators, the underlying generative model, or something else?

* The core idea of the paper is solid, but I'm wary of the use of human judgements for creating ground truth. The models are using features/representations extracted from each input modality, and a human's perception/judgement here may differ significantly from signal in the data. I acknowledge there may not be a better solution for annotations. (aside: Thanks for putting info about the annotators' backgrounds and instructions in the appendix).

* Similarly, in "Results on multimodal fusion," given my experience with these kinds of models I'm hesitant to believe the rational/intuition such as "since sarcasm is often due to a contradiction between what is expressed in language an speech." It seems like a natural conclusion, but my understanding is that speech emotion (or sarcasm) detectors use a more limited context to compute their predictions, which may not have enough context to encode this contradiction  (but maybe I'm wrong?)? The rational here "make sense" but seem cherry picked and might not be "true." Couldn't there be other (potentially spurious) correlations that cause the results to look nice?

* I imagine there are alternative correlations or measures one could use for evaluating redundancy/uniqueness/synergy. It would be useful to see more discussion (or comparison) with this.




**Questions:**

* Sec 3.1 (preprocessing via feature binning): Is estimating this using binning a reasonable idea here? I suppose this depends on the bin size, dimensionality, size of the dataset, and the distribution of latents.

* The "Batch" estimator is interesting. I immediately wondered if VQ-VAEs or other discrete state latent variable models could be used as an alternative here. This would give you (discrete) tokens instead of having to optimize over a continuous space. Does this seem like a reasonable idea? This idea seems simpler to me and means that you don't have to subsample batches as done here.

**Limitations:**

Yes, more so in the appendix.

---

> ### Author Rebuttal · Authors · 2023-08-09
>
> Thank you for your valuable feedback and insightful comments! We respond to some concerns below:
>
> [scale] **The absolute value (i.e., actual bits of information) will never be fully 1 bit, because one can never extract the exact amount of task-relevant information**, which would imply 100% train and test accuracy. Noise in the data and labels, errors in feature learning and model optimization, and generalization gaps between train and test all contribute to lower estimated interaction values than the ground truth.
>
> We can get a sense of the total capturable information using the bound from Feder and Merhav [1], which relates the accuracy of the best model on these datasets with the mutual information between the inputs to the label. **Since the test accuracies for Table 2 datasets range from 67-75%, this corresponds to total MI of 0.422-0.585 bits**. This is a tighter upper bound ground truth that our estimators should capture, and we see that they are quite close, see updated Table 2 in attached rebuttal pdf.
>
> We are aware of challenges in evaluating PID estimation and emphasize that they should be interpreted as a relative sense of which interaction is most important. Both CVX and BATCH measures agree with each other on relative values and are consistent with ground truth interactions on relative values.
>
> Finally, we also emphasize that evaluating PID estimation is just half the picture - the more important part is using these quantities for downstream real-world case studies in multimodal model selection and estimating robustness to missing modalities for various applications.
>
> [1] Feder and Merhav. Relations between entropy and error probability. IEEE Transactions on Information theory 1994
>
> [human judgement] We are aware of the caveats regarding human annotation and annotator biases, but unfortunately for the real-world datasets (and especially 3 case studies) it may be the best choice we have. **The inter-annotator agreement is relatively high (0.72 for R, 0.68 for U1, 0.70 for U2, 0.72 for S) and the average confidence scores are also high (4.36/5 for R, 4.35/5 for U1, 4.27/5 for U2, 4.46/5 for S), indicating that human judgement is quite consistent across annotators**.
>
> [sarcasm] For sarcasm, we find that humans do estimate both synergy and unique language as relatively high - sometimes humans can tell sarcasm from language only. We will rephrase as ‘sarcasm can be (instead of often) due to a contradiction between what is expressed in language and speech’. We use sarcasm datasets that are more multimodal in nature (from the Mustard dataset of acted sarcasm in TV shows like the Big Bang Theory where the actors are more sarcastic in nature than real-life speech), which we believe explains the higher synergy in sarcasm we found.
>
> [other measures] **We implemented 3 alternative definitions for interactions, including the original called I-min, WMS, and CI (see [1] for a review). These results are in Table 1 of rebuttal pdf:**
>
> 1. I-min can sometimes overestimate synergy and uniqueness. It overestimates synergy on U1-only and U2-only datasets, and assigns non-zero unique interactions (U1=0.08, U2=0.07 and U1=0.03, U2=0.04) even though there is no uniqueness.
> 2. WMS is actually synergy minus redundancy, so when they are of equal magnitude WMS cancels out to be 0, leading one to erroneously conclude there is no S or R. For example, it is unable to detect high synergy for S-only data and assigns uniqueness instead. This term can also be negative (S=-0.11 for R-only data).
> 3. CI can also be negative (U1=-0.09, U2=-0.10 for R-only data), and sometimes incorrectly concludes highest uniqueness (U1=0.13, U2=0.14) for S-only data.
>
> **We also implemented 3 more non-info theory measures:**
>
> 1. Shapley and integrated gradients estimate interactions *captured by a multimodal model - our work is fundamentally different in that interactions are properties of data*. This allows us to characterize datasets and select the most appropriate model. We ran this and found 3 issues:
> - It only captures overall independence, not separate uniqueness for each modality
> - It overestimates synergy=0.5 for pure redundancy data and overestimates independence=1.0 for pure synergy data
> - We also found difficulties in estimating Shapley for large high-dimensional datasets efficiently
>
> 2. CCA only gives shared information between modalities, which can be used as a proxy of redundancy, but does not give other interactions. We ran this and found that one can learn representations to maximize correlation arbitrarily high (e.g., rho=1.0 for MOSEI), even though the redundancy is bounded.
>
> [1] Griffith and Koch. Quantifying synergistic mutual information. 2014
>
> [binning] With a larger number of bins (e.g., >50), our estimates barely change, unless too many bins are chosen (which can lead to numerical instability as some bins become empty or without any samples). One recommendation in [2] is to select the number of bins to be the cubed root of the number of samples. We provide more details in Appendix B1. For modalities with higher dimensions, we recommend the use of our BATCH estimator instead due to exponential blowup in the number of bins wrt dimension.
>
> [2] Rice. Mathematical statistics and data analysis. Cengage Learning 2006.
>
> [batch and vq-vae] Absolutely: using the idea of discrete clusters for the CVX estimator, but with learned cluster representations of continuous data using VQ-VAE rather than fixed clusters as we do now. However, training VQ-VAE on high-dimensional multimodal data is non-trivial, and end-to-end training raw data -> VQ-VAE -> cluster representations -> convex solver -> PID objective is an even cooler idea but requires rewriting convex optimizers in pytorch. We would love to explore these in future work, and have added a discussion to our paper!

---

> > ### Comment · Reviewer_td1f · 2023-08-16
> >
> > Thanks for the detailed response. I upped the 'presentation' score but otherwise left unchanged. There is clearly a lot of thought that went into the paper and rebuttal and the work is "complete" in some sense. But I'm still unsure if some of the underlying design decisions (i.e., comparing to human annotators) are useful. I'm curious to see additional comments from other reviewers.

---

### Official Review · Reviewer_4r5p · 2023-07-06

**Soundness:** 3 good
**Presentation:** 3 good
**Contribution:** 3 good
**Rating:** 7
**Confidence:** 3

**Summary:**

This paper attempts to quantify the interactions between different input modalities using Principle Information Decomposition (PID) from Information Theory. Using this approach, one can quantify not only interactions within multimodal datasets but as well as interactions captured by multimodal models.
The paper provided a comprehensive evaluation of their approach and provided examples of its application in real-world case studies.


**Strengths:**

Provided a detailed explanation of its theoretical background, and thorough comprehensive evaluation. They also provided a list of real-world applications which are useful for casual users.

**Weaknesses:**

The presented work, would benefit from more visuals. As this paper is quite detailed, it would benefit from a closing summary listing the main points of the paper.


**Questions:**

Are there any potential use cases of this approach outside multimodal models? Like, e.g. measuring the domain gap between synthetic and real-world data?

**Limitations:**

There is no dedicated section on its Limitations, nor is it focused on in the paper as far as I could find. It would be useful so that the readers would know in which cases this could potentially not be effective.

---

> ### Author Rebuttal · Authors · 2023-08-09
>
> Thank you for your valuable feedback and insightful comments! We respond to some concerns below:
>
> [visuals] Thank you for pointing this out - we do plan to add more visuals and have included a rebuttal pdf with some other figures we have included to the paper:
> 1. Figure 1: A venn diagram to give an intuitive explanation of multimodal interactions and the main outline of the paper (dataset quantification, model quantification, model selection).
> 2. Figure 2: A figure to give an intuitive explanation of PID, distributions q \in \Delta_p, and computation of synergy,
> 3. Figure 3: To show an illustration of model selection based on quantifying dataset interactions and model interactions.
>
> We also have a summary section in Appendix D including key takeaway messages, and new directions for future work. At a high-level, the key ideas are:
> 1. Dataset quantification: PID provides reliable indicators for interactions present in both synthetic and real-world multimodal datasets.
> 2. Model quantification: PID also quantifies interactions captured by trained models, and we find several consistent patterns in the types of interactions different models capture.
> 3. Model selection demonstrates potential utility as a ‘user-guide’ for practitioners aiming to tackle real-world multimodal datasets. Given a new dataset, estimate its PID values. If there is high U1 and U2, just using unimodal models in the corresponding modality may be sufficient. Otherwise, if there is high R, methods like agreement, alignment, ensemble, or co-training should be tried. If there is high S, it is worth spending the time on multimodal methods that model more complex interactions based on tensors, multiplicative interactions, and self-attention. We also show results on automatically selecting the top-3 models, which recover >96% of the best performance from exhaustively trying all models.
>
> [outside multimodal] Yes, **this framework can be used to study feature interactions more broadly even within single modality models (e.g., multiple image regions, multiple sentences) [1] as well as bias and unfairness between different interacting features** [2,3]. PID can be used to quantify the contribution of individual features as well as interacting features to unfair predictions or privacy leakage, which can then in turn be useful for minimizing the effect of these features and interactions to improve fairness and privacy. We believe that these are all ripe directions for future work. Being able to estimate and provide guarantees for fairness and privacy-preserving learning can be a particularly impactful application. There are also some connections between classical information theory and distribution shift [4], but not PID and other fine-grained interactions which could also yield new insights of future work.
>
> [1] Tsang et al., Feature interaction interpretability: A case for explaining ad-recommendation systems via neural interaction detection. ICLR 2019
>
> [2] Dutta et al., An information-theoretic quantification of discrimination with exempt features. AAAI 2020
>
> [3] Hamman et al., Demystifying Local and Global Fairness Trade-offs in Federated Learning Using Information Theory. FL-ICML 2023
>
> [4] Federici et al., An Information-theoretic Approach to Distribution Shifts. NeurIPS 2021
>
> [limitations] We have expanded on the limitations and broader impact section in Appendix E. To summarize:
> 1. CVX works for smaller domains up to several thousand classes or may suffer from low empirical frequencies beyond that; BATCH may suffer from approximation error depending on whether the network can sufficiently cover the space of joint distributions, bias resulting from mini-batch gradient descent, as well as the usual local minima and unstable convergence problems with neural estimators.
> 2. These estimators only approximate real interactions due to cluster preprocessing as well as the standard optimization and generalization errors of approximate models. We expect future progress in density estimators, generative models, and MI estimation to help address these problems.
> 3. We are aware of challenges in evaluating PID estimation and emphasize that they should be interpreted as a relative sense of which interaction is most important, and a guideline to inspire model selection and design.

---

### Official Review · Reviewer_oDWq · 2023-07-07

**Soundness:** 3 good
**Presentation:** 3 good
**Contribution:** 3 good
**Rating:** 6
**Confidence:** 4

**Summary:**

This paper proposes an information-theoretic approach to quantify the degree of redundancy, uniqueness, and synergy relating input modalities with an output task. The authors introduce two new estimators for these PID statistics that scale to high-dimensional distributions. They validate the PID statistics on synthetic and real-world datasets, demonstrating their ability to identify meaningful interactions and select appropriate models. The authors also compare their approach to other methods for quantifying multimodal interactions, showing that their method outperforms these alternatives. They provide examples of real-world applications where the PID statistics were useful in quantifying interactions within multimodal datasets or selecting appropriate multimodal models. The authors discuss the limitations of their approach, including the need for large amounts of data and the difficulty of interpreting the results. They suggest future directions for research, such as extending the approach to continuous-valued data and incorporating causal relationships between modalities. Overall, this paper provides a useful framework for quantifying and modeling multimodal interactions that has potential applications in a wide range of fields.

**Strengths:**

1. Novel approach: The paper proposes a novel information-theoretic approach to quantify the degree of redundancy, uniqueness, and synergy relating input modalities with an output task. This approach is different from other methods for quantifying multimodal interactions and has the potential to provide new insights into how different modalities interact.

2. Extensive validation: The authors validate their approach on both synthetic and real-world datasets, demonstrating its ability to identify meaningful interactions and select appropriate models. They also compare their approach to other methods for quantifying multimodal interactions, showing that their method outperforms these alternatives.

3. Real-world applications: The paper provides examples of real-world applications where the PID statistics were useful in quantifying interactions within multimodal datasets or selecting appropriate multimodal models. This demonstrates the practical usefulness of the approach in a wide range of fields.

4. Clear presentation: The paper is well-written and clearly presents the approach, the validation experiments, and the real-world applications. The authors provide detailed explanations of the methods used and the results obtained, making it easy for readers to understand and replicate their work

**Weaknesses:**

1. Limited scope: The paper focuses on a specific information-theoretic approach to quantifying multimodal interactions and does not consider other approaches or methods. This may limit the generalizability of the results and the usefulness of the approach in other contexts.

2. Lack of comparison to ground truth: The paper validates the approach on synthetic and real-world datasets, but does not compare the results to a ground truth or gold standard. This makes it difficult to assess the accuracy of the approach and the validity of the results.

3. Limited interpretability: The paper acknowledges that the results of the approach can be difficult to interpret, particularly in high-dimensional datasets. This may limit the usefulness of the approach in practice, as it may be difficult to draw meaningful conclusions from the results.

4. Lack of implementation details: The paper provides a high-level overview of the approach and the methods used, but does not provide detailed implementation instructions or code. This may make it difficult for other researchers to replicate the work or apply the approach in their own research.

**Questions:**

1. How does the proposed approach compare to other methods for quantifying multimodal interactions, such as correlation analysis or principal component analysis?
2. Can the approach be applied to continuous-valued data, or is it limited to discrete data?
3. How does the approach handle missing data or incomplete modalities?
4. How does the approach handle interactions between more than two modalities?
5. In Table 1, what is Exact task? Why all task and measures have the same numbers?
6. Table 3 is hard to understand, maybe better to use the same scale?


**Limitations:**

Yes

---

> ### Author Rebuttal · Authors · 2023-08-09
>
> Thank you for your valuable feedback and insightful comments! We respond to some concerns below:
>
> [other methods] Our framework allows for easy generalization to other definitions as well: **we implemented 3 alternative definitions for interactions, including the original called I-min, WMS, and CI (see [1] for a review). These results are in Table 1 of rebuttal pdf**:
> 1. I-min can sometimes overestimate synergy and uniqueness. It overestimates synergy on R-only, U1-only, and U2-only datasets, and assigns non-zero unique interactions (U1=0.08, U2=0.07 and U1=0.03, U2=0.04) even though there is no uniqueness.
> 2. WMS is actually synergy minus redundancy, so when they are of equal magnitude WMS cancels out to be 0, leading one to erroneously conclude there is no S or R. For example, it is unable to detect high synergy for S-only data and assigns uniqueness instead. This term can also be negative (S=-0.11 for R-only data).
> 3. CI can also be negative (U1=-0.09, U2=-0.10 for R-only data), and sometimes incorrectly concludes highest uniqueness (U1=0.13, U2=0.14) for S-only data.
>
> **Our framework also fits in with non-info theory measures, we implemented 3 more**:
> 1. Shapley is based on interactions captured by a multimodal model - our work is fundamentally different in that interactions are properties of *data*. This allows us to characterize datasets and select the most appropriate model. We ran this and found 3 issues:
> - It only captures overall independence, not separate uniqueness for each modality
> - It overestimates synergy=0.5 for pure redundancy data and overestimates independence=1.0 for pure synergy data
> - We also found difficulties in estimating Shapley for large, high-dimensional datasets efficiently, which is still an open question.
> 2. Integrated gradients (IG) only gives unimodal contributions of a trained model. We apply this method for computational pathology to derive importance of histology and genomics data, and find IG echoes the unique interactions of PID, but PID enables us to do so *before* model selection and training. Moreover, it is not clear how to estimate synergy with IG.
> 3. CCA only gives shared information between modalities, which can be used as a proxy of redundancy, but does not give other interactions. We ran this and found that one can learn representations to maximize correlation arbitrarily high (e.g., rho=1.0 for MOSEI), even though the redundancy is bounded.
>
> [1] Griffith and Koch. Quantifying synergistic mutual information. 2014
>
> [comparison] **We tried our best to compare to ground truth in**:
> 1. Table 1, bitwise operators, with ground truth PID values obtained from exactly solving the optimization problem, and both our estimators recover the ground truth.
> 2. Table 2, synthetic datasets (10 types), with ground truth based on which latent features the label was sampled from (shared feature, unique to either modality, or their combination), and we show consistent trends with ground truth.
> 3. Table 3, real-world datasets (8 types), with gold standard obtained from human annotation of interactions, and we show consistent trends with gold standard.
>
> [interpretability] Our approaches are designed to deal with high-dimensional data through (1) clustering followed by discrete convex optimization, and (2) end-to-end representation learning from continuous data. **Our approach does scale to high dimensions in practice through 8 large multimodal datasets and 3 additional real-world case studies, enabling domain experts to interpret these interactions.**
>
> [details] All code and data are in the supplementary material so details are transparent and reproducible, and we will put the code on github. We will add further requested details to appendix.
>
> [continuous] We propose 2 estimators, and our second BATCH estimator is designed to extend to continuous data, since we use neural networks to encode continuous inputs and learn a distribution q used to compute interaction values. The first CVX estimator assumes discrete support for distribution q so operates via a clustering preprocessing step.
>
> [missing modalities] R, U1, U2 can be estimated with access to only p(x1,y) and p(x2,y) which means they can be estimated in the presence of missing modalities. Synergy needs the full distribution p(x1,x2,y) due to the first I_p(X1,X2;Y) term. However, we can approximate synergy via the disagreement between unimodal classifiers p(y|x1) and p(y|x2), which [2] showed is related to the unique information I(X1;Y|X2) and I(X2;Y|X1). Extension of our work to missing modalities is an important avenue for future work.
>
> [2] Sridharan and Kakade. An information theoretic framework for multi-view learning. COLT 2008.
>
> [>2 modalities] Extensions to more than 2 modalities is challenging due to a larger number of interactions (e.g., x1 and x2 may exhibit synergy, but not with x3), but future work can leverage advances in information theory: e.g. one could consider decomposing features hierarchically, or approximating via pairwise synergies.
>
> [Table 1] Table 1 tests 3 multimodal distributions, where x1, x2 are 0/1 bit vectors, and the label y = x1 OR x2, y = x1 AND x2, and y = x1 XOR x2. For each of the 3 distributions the PID can be solved exactly, for example y = x1 XOR x2 requires 1 bit of S, y = x1 OR x2, y = x1 AND x2 require 0.31 bits of R and 0.5 bits of S. In comparison with this ground truth, both our estimators exactly recover the correct PID values.
>
> [Table 3] For human judgment, we cannot ask humans to give a score in bits, so it is on a completely different scale (1-5 scale). To put them in the same scale, we normalize the human ratings such that the sum of human interactions = sum of PID estimates. The resulting comparisons are in Table 2 in attached rebuttal pdf, and we find very close match between estimators and human gold standards.

---

> > ### Comment · Reviewer_oDWq · 2023-08-17
> >
> > Dear Authors,
> >
> > Thank you for all detailed answers and clarifications. I agreed with reviewer td1f, and I still have concerns about the interpretability and human judgement as golden standard. But I think this is a solid, interesting and well-written paper, and keep my scores.

---

### Author Rebuttal · Authors · 2023-08-09

Dear all reviewers, we are extremely grateful for your valuable feedback and insightful comments. We are glad that you agree that our framework can be of fundamental importance for studying multimodal interactions, and that our proposed estimators are technically novel and comprehensively evaluated. Your concrete suggestions are a valuable step in this direction, and we have revised our submission accordingly to take these into account. In this short note we summarize the main changes we made to our submission:

[other methods] Our framework allows for easy generalization to other interaction definitions: we implemented 3 information theoretic measures I-min, WMS, and CI. These results are in Table 1 of the attached rebuttal pdf, and we outlined several limitations of these methods as compared to PID. We also tried 3 non-info theory measures: Shapley values, Integrated gradients (IG), and CCA, which are based on quantifying interactions captured by a multimodal model - our work is fundamentally different in that interactions are properties of data before training any models.

[table 2] The absolute value (i.e., actual bits of information) will never be fully 1 bit, because one can never extract the exact amount of task-relevant information, which would imply 100% train and test accuracy. We estimated the total capturable information based on test accuracy and we see much closer PID estimates to ground truth: see Table 1 in the attached rebuttal pdf.

[table 3] For human judgment, we cannot ask humans to give a score in bits, so it is on a different scale (1-5 scale). To put them in the same scale with PID estimates, we normalize the human ratings such that the sum of human interactions = sum of PID estimates. The resulting comparisons are in Table 2 of attached rebuttal pdf, and we find very close match between PID estimates and human gold standards.

[limitations] We have expanded on the limitations and broader impact section in Appendix E, highlighting limitations and future directions in improving the estimators and their evaluation leveraging future progress in multimodal benchmarks, generative models, and MI estimation.

[details and practice] Finally, we would like to emphasize that all code and data are in the supplementary material, and we plan to release all code and data on github after the review period so details are transparent and reproducible. We have also added all requested details regarding estimators and experiments to the appendix.

---

### Decision · Program_Chairs · 2023-09-21

**Decision:**

Accept (poster)

**Comment:**

The reviewers are generally positive about the work, and even more so after the rebuttal. Although there are many detailed questions about the clarity, experimental design and generalizability, the reviewers overall see the work contributes a solid investigation into an important problem. The reviewers think the work is "novel" and "principled" and the experiments are "extensive". Specifically, Reviewer uMC4 stated (the work) "Deepening our understanding of multimodal data, and models built to address tasks that rely on multimodal data is of fundamental importance." The authors also did a good job at addressing the reviewers's questions during rebuttal, which strengthened the reviewers's confidence in accepting the paper.